

# Aspects of categorical symmetries from branes: SymTFTs and generalized charges

Fabio Apruzzi[1,2], Federico Bonetti[3], Dewi S.W. Gould[4] and Sakura Schäfer-Nameki[4]

**1** Dipartimento di Fisica e Astronomia "Galileo Galilei", Università di Padova,
Via Marzolo 8, 35131 Padova, Italy
**2** INFN, Sezione di Padova Via Marzolo 8, 35131 Padova, Italy
**3** Department of Mathematical Sciences, Durham University, Durham, DH1 3LE, UK
**4** Mathematical Institute, University of Oxford, Andrew-Wiles Building,
Woodstock Road, Oxford, OX2 6GG, UK

## Abstract

Recently it has been observed that branes in geometric engineering and holography have a striking connection with generalized global symmetries. In this paper we argue that branes, in a certain topological limit, not only furnish the symmetry generators, but also encode the so-called Symmetry Topological Field Theory (or SymTFT). For a $d$-dimensional QFT, this is a $(d+1)$-dimensional topological field theory, whose topological defects encode both the symmetry generators (invertible or non-invertible) and the generalized charges. Mathematically, the topological defects form the Drinfeld center of the symmetry category of the QFT. In this paper we derive the SymTFT and the Drinfeld center topological defects directly from branes. Central to the identification of these are Hanany-Witten brane configurations, which encode both topological couplings in the SymTFT and the generalized charges under the symmetries. We exemplify the general analysis with examples of QFTs realized in geometric engineering or holography.

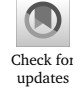

# 1   Introduction

Branes play a central role in string/M-theory: as carriers of gauge degrees of freedom, non-perturbative defects, and as the origin of holographic dualities when back-reacted. Recently it has been observed that in a particular, non-dynamical, limit they give rise to generators of generalized global symmetries (topological defects) in holography [1,2] and in geometric engineering of QFTs [3,4].

Much is known about higher-form and higher-group symmetries in the context of geometric/brane engineering and holography [5–36]. However, most of this work focuses on the (not necessarily topological) extended defect operators, i.e. the generalized (or higher-) charges, which are constructed by wrapping branes on non-compact cycles. These extended objects then mimic infinitely heavy probes in space-time. In turn, relatively little had been known about the symmetry generators in the context of geometric constructions – see however [7,8] for some discussion in terms of flux operators. The recent identification of symmetry generators with branes [1–4] in a topological limit[1] provides a systematic way to study the symmetries of a given theory.

**SymTFTs and Generalized Charges.**   Recent developments in the realm of generalized symmetries have lead to the idea that separating symmetries from physical theories can be insightful. The structure that allows for this is the Symmetry Topological Field Theory (SymTFT) [38–40]. This has several applications, see [1, 39, 41–44]. The SymTFT is invariant under gauging of global symmetries (i.e. symmetries that are related by gauging have the same SymTFT), and perhaps physically most relevant, its topological defects encode the generalized charges [45, 46]. The separation that seems to have emerged in string theory constructions, into symmetry generators and generalized charges is therefore somewhat artificial. There should be a unified prescription that derives from the string theory construction (geometric engineering or holography) of the SymTFT directly.

---

[1]We will equivalently use both "topological limit" or "topological truncation". The procedure we adopt is a truncation to the topological sector [37], which will be explained in more details in section 3.

**SymTFT and Supergravity.**   In string/M-theory the initial constructions of the SymTFT were closely related to various topological sectors of dimensionally reduced supergravity theories: the initial analysis of the SymTFT in [39] derived it in the context of geometric engineering on a non-compact space **X** in M-theory. The SymTFT in that context was derived by dimensional reduction on $\partial$**X**. Likewise in holography it is connected to the topological sector of the bulk supergravity [37,47]. It emerges naturally via anomaly inflow methods for QFTs realized with branes [48–50].

Given the recent proposal [1–4] relating symmetry generators and branes in string theoretic settings, it is natural to ask whether branes also provide a realization of the SymTFT, in particular the topological defects of the SymTFT, as well as the generalized charges. In this paper we propose a general framework for this, and substantiate it in various setups in both geometric engineering and holography. This connects also to the general philosophy, that the symmetry and generalized charges should all have a unified construction in terms of the SymTFT topological defects.

**SymTFT and Drinfeld Center.**   Mathematically the topological defects of the SymTFT correspond to the Drinfeld center of the symmetry category $\mathcal{S}$ of the physical theory $\mathcal{T}$. The symmetry categories in QFT are generically fusion higher-categories, for which it is indeed known that the Drinfeld center is invariant under gauging (Morita equivalence) [51]. The SymTFT for a $d$-dimensional theory $\mathcal{T}$ is a $(d+1)$-dimensional theory. It has two boundaries: the symmetry boundary $\mathcal{B}^{\text{sym}}$, which is gapped, and the physical boundary $\mathcal{B}^{\text{phys}}$. Compactification of this 'sandwich' results back in the theory $\mathcal{T}$, see figure 1. The topological defects that project parallel onto the symmetry boundary (Neumann boundary conditions)[2] result in topological defects in the boundary. In turn extended operators that can end or form junctions with topological defects on the boundary result in non-topological defect operators in the interval compactification. This is relatively well-understood for theories with abelian symmetries (i.e. finite abelian higher-form or higher-group symmetries), but is true more generally for any type of generalized symmetry [45,46], and in 2d in [52,53].

In particular it becomes a key tool to study gauging and generalized charges for noninvertible symmetries in higher dimensions, whose constructions have been abundant in the past 2 years [1–4, 36, 41, 43, 44, 46, 53–105]. For reviews on this topic see [106, 107].

**Summary of Results.**   In this work we will argue that branes (in a certain topological limit) encode the SymTFT of QFTs that are realized in terms of geometric engineering or in holographic dualities. As most geometric engineering and holographic theories mostly admit abelian generalized symmetries, we will focus on these symmetries. Restricting to these symmetries, this amounts to showing that branes give rise to BF-terms and (mixed) 't Hooft anomalies at the level of SymTFT topological couplings. At the level of defects of the SymTFT, we use brane effects to determine generalized charges of higher-form symmetries (which are the topological defects of the SymTFT). In the process we also identify condensation defects in terms of branes.

**BF-terms** for abelian finite higher-form symmetries will be shown to be encoded in the topological sector of 10/11d supergravity once we also include **source terms for wrapped branes**. These have the interpretation of generating the associated symmetries. Terms of this type derive from two origins: either from Chern-Simons terms or kinetic terms in the supergravity action. By including sources, these terms describe how the geometric linking of wrapped branes in the bulk corresponds to the action of symmetry generators, i.e. topological defects, on (extended) charged operators of the QFT.

---

[2]Note that in a topological field theory, such as the SymTFT, Neumann corresponds to freely varying and Dirichlet to fixed boundary conditions.

Including brane sources induces further topological couplings in the SymTFT, which in some global forms can have the interpretation of (mixed) 't Hooft anomalies. We will refer to these topological couplings in the SymTFT as **anomaly couplings**. They are encoded in various linking configurations of branes in the bulk. We first give a general procedure for deriving anomaly couplings from the 10/11d supergravity topological sector and Bianchi identities in terms of background fields. Re-phrasing these relations in terms of brane sources allows us to re-write anomaly couplings as linking configurations of the branes which generate the associated symmetries.

An important aspect of the categorical description of symmetries is the notion of condensation completion [108]: i.e. all symmetries can be condensed on topological defects that are generically defined on submanifolds of spacetime (as opposed to the whole spacetime). So far the conjectured identification of branes with symmetry generators [1,2] does not incorporate condensation defects. In this paper we argue that **condensation defects** can be constructed from a "condensation completed" SymTFT, where in addition to the BF-couplings in the $(d+1)$-dimensional spacetime of the SymTFT, we also include couplings to either lower dimensional discrete gauge theories (possibly with theta angles), which realize standard condensation defects, or more generally lower-dimensional TQFTs which give rise to so-called (twisted) theta defects [46].

In the string theoretic setting we will obtain such couplings by considering **brane-anti-brane pairs** (in a topological limit), where the standard D$p$-brane charge cancels out, but topological couplings on the world-volumes survive, which live in lower than $p+1$ dimensions.

The topological defects of the SymTFT encode the **generalized charges** of a categorical symmetry [46]. In particular the linking in the SymTFT (for abelian symmetries) provides a way to compute the charges. It was already shown in [1] that in a specific 4d $\mathcal{N}=1$ Super-Yang-Mills theory setting, the action of generalized symmetries on branes can be realized in terms of Hanany-Witten moves on brane-configurations. This realizes the action of the non-invertible symmetries on 't Hooft lines in the $PSU(N)$ SYM theory. In this paper we show that more generally, the action of generalized symmetries on generalized charges has as its origin the **Hanany-Witten** configuration and moves for branes.

The general considerations of this paper will be illustrated with numerous examples, both in geometric engineering and holography, including:

1. Holographic and geometric engineering constructions of 4d $\mathcal{N}=1$ Super-Yang-Mills theory (SYM) with gauge algebra $\mathfrak{su}(N)$, considering various global forms of the gauge group.

2. 4d $\mathcal{N}=4$ SYM with gauge algebra $\mathfrak{so}(4N)$.

3. 4d $\mathcal{N}=4$ SYM with gauge algebra $\mathfrak{su}(N)$ with duality/ triality defects.

4. 4d $\mathcal{N}=2$ Argyres-Douglas theories with duality defects.

In the setups 1. and 2. we explain how known SymTFT couplings (BF and anomaly couplings) can be derived from the linking of branes in the bulk, giving a new perspective on these theories. Furthermore we explain how Hanany-Witten configurations signal the presence of mixed 't Hooft anomalies in each theory, and furthermore, how they describe the generalized charges. In particular for the various global forms of the gauge groups for $\mathfrak{so}(4n)$ algebras we identify 2d generalized charges for non-invertible 1-form symmetries using this method. For the gauge group $\mathcal{G} = \mathrm{Spin}(4N)$ global variant we also show how brane effects model the 2-group global symmetry.

In the setup 3. we show how Hanany-Witten configurations can be used to diagnose the intrinsic/ non-intrinsic nature of non-invertible symmetries. The brane mechanism imposes

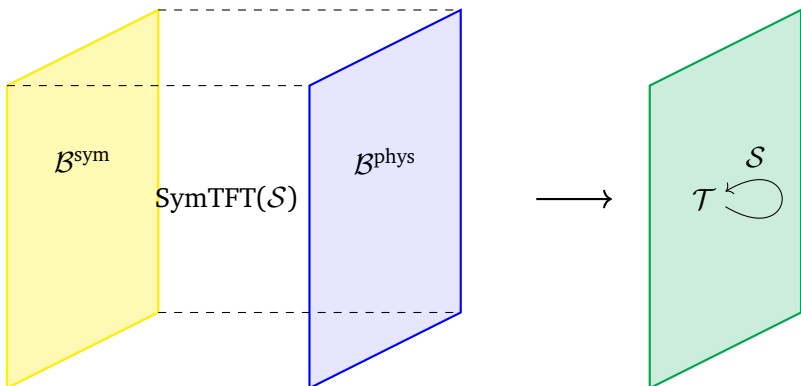

Figure 1: The SymTFT sandwich and interval compactification to a $d$-dimensional physical theory $\mathcal{T}$ with symmetry $\mathcal{S}$. The SymTFT is a $(d + 1)$-dimensional theory, SymTFT($\mathcal{S}$), which is shown on the left: it has two boundaries, the gapped, symmetry boundary $\mathcal{B}^{\text{sym}}$, and the physical boundary $\mathcal{B}^{\text{phys}}$, which is not gapped (unless the theory $\mathcal{T}$ was topological as well).

a simple constraint which allows a classification of the type of these non-invertible symmetries for arbitrary gauge group rank, extending previous results [72]. In the setup 4. we propose new brane origins for the topological defects generating (non-invertible) higher-form symmetries and derive the SymTFT for these theories using brane sources. In particular, the construction relies on world-volume flux on branes which induces lower-dimensional brane charges.

**Notational conventions.** Spacetime dimension for the QFT is $d$, the spacetime for the supergravity theory (string or M-theory) that we start with is $D + 1 = 10, 11$. The dimensional reduction is either on the link $L_n \equiv L_n(\mathbf{X}_{n+1}) = \partial \mathbf{X}_{n+1}$ of dimension $n$ or the cone over the link $\mathbf{X}_{n+1}$ which is $n + 1$ dimensional. Gauge groups will be denoted by $\mathcal{G}$. $p$-dimensional topological defects are $D_p$ and $q$-charges (not necessarily topological $q$-dimensional charged defects) are $\mathcal{O}_q$.

## 2 Symmetry TFTs: Symmetries and generalized charges

### 2.1 Symmetry TFT

The general idea behind the SymTFT is to separate the symmetry aspects from the non-topological degrees of freedom of a QFT [38–40]. Throughout this paper, we consider generalized symmetries that are abelian, i.e. they have a formulation in terms of abelian background fields, and their SymTFT has an action formulation in terms of these backgrounds. Most known symmetries that arise in the holographic or geometric engineering context seem to be of this type (or gauge-related to such symmetries), rather than more general categorical symmetries (e.g. non-abelian group or higher-representations for non-abelian groups). For a discussion of the SymTFT in this more general setting in particular in view of the generalized charges see [46, 82].

Consider a $d$-dimensional theory $\mathcal{T}$ with symmetry

$$\mathcal{S} = \prod_p G^{(p)} , \tag{1}$$

which in the present case is a product of higher-form symmetry groups. Here we assume that

$\mathcal{T}$ is an absolute theory (so we have chosen a specific polarization in the defect group). Denote the background fields for individual components of the $p$-form symmetry group $G^{(p)}$ by

$$B_{p+1}^i \in H^p\left(M_d, \mathbb{Z}_{n_i^p}\right), \tag{2}$$

where $G^{(p)} = \prod_i \mathbb{Z}_{n_i^p}$ for some $n_i^p \in \mathbb{Z}_+$. Furthermore, these higher-form symmetries can have non-trivial (mixed) 't Hooft anomalies which we summarize as $\mathcal{A}(\{B_{p+1}\})$.

The SymTFT will be a $(d+1)$-dimensional theory, which can be thought of as a gauging of the $p$-form symmetries in $(d+1)$-dimensions, i.e. coupling the theory to a Dijkgraaf-Witten type dynamical discrete gauge theory. This contains then BF-couplings for the now dynamical fields $b_{p+1}^i$ and the dual fields $\widehat{b}_{d-p-1}^i$, as well as the anomaly term

$$S_{\text{SymTFT}} = \int_{M_{d+1}} \sum_p \sum_{i,j} n_{ij}^p \, b_{p+1}^i \wedge d\widehat{b}_{d-p-1}^j + \mathcal{A}\left(\{b_{p+1}^i\}\right). \tag{3}$$

Here we use a continuum field formulation. The fields $b$ are $U(1)$-valued, with equation of motion $n \, db = 0$. These are related to the finite group cocycles by $b^{\text{continuum}} = \frac{2\pi}{N} b^{\text{discrete}}$. Here we have chosen a normalization which removes any explicit $2\pi$ factors in the action. We will later on consider generalization to twisted cocycles in specific examples.

## 2.2 Topological defects of the SymTFT

The SymTFT has two $d$-dimensional boundaries: one is the physical boundary $\mathcal{B}^{\text{phys}}$ which is generically not gapped, and the second is the symmetry boundary $\mathcal{B}^{\text{sym}}$ which is gapped. See figure 1. Upon interval compactification back to $d$-dimensions, we obtain the theory $\mathcal{T}$ with symmetry $\mathcal{S}$.

The topological defects of the SymTFT will be denoted by $\mathbf{Q}$, and in this setup are given in terms of the generalized Wilson lines for the gauge fields.

In general we can determine the charges from a SymTFT of type (3) by determining the Gauss law constraints and then exponentiating (see e.g. [1, 109]). To illustrate the setup lets start in the absence of any anomaly couplings $\mathcal{A} = 0$, then the topological defects are generated by

$$\mathbf{Q}_{p+1}^{(b^i)}(M_{p+1}) = \exp\left(2\pi i \int_{M_{d+1}} b_{p+1}^i\right),$$

$$\mathbf{Q}_{d-p-1}^{(\widehat{b}^i)}(M_{d-p-1}) = \exp\left(2\pi i \int_{M_{d-p-1}} \widehat{b}_{d-p-1}^i\right). \tag{4}$$

Mathematically, these are the elements of the Drinfeld center of the SymTFT. These have a non-trivial commutation relation

$$\mathbf{Q}_{p+1}^{(b^i)}(M)\mathbf{Q}_{d-p-1}^{(\widehat{b}^i)}(M') = \exp\left(2\pi i \frac{L(M_{p+1}, M'_{d-p-1})}{n_i^p}\right)\mathbf{Q}_{d-p-1}^{(\widehat{b}^i)}(M')\mathbf{Q}_{p+1}^{(b^i)}(M). \tag{5}$$

Here $L(M, M')$ is the linking of the two manifolds in the $(d+1)$-dimensional spacetime.

However in the presence of non-trivial couplings between the fields $b_{p+1}^i$ in $\mathcal{A}(\{b_{p+1}^i\})$, there will be additional terms in the above expressions for the topological defects. We will study those in detail in subsequent sections, rather than present a general analysis here.

**SymTFT from Theta-Defects.** A useful perspective is to think of the SymTFT in this case as a theta-defect construction [46, 67]: we consider the trivial SymTFT and gauge the $G^{(p)}$-symmetry in $(d + 1)$-dimensions. Such a gauging always allows for the inclusion of a class of $(q + 1)$-dimensional topological defects, the theta defects, that correspond to $(q + 1)$-representations of the higher-form symmetry group $G^{(p)}$ (these are the $(q + 1)$-dimensional TQFTS that are $G^{(p)}$-symmetric). In particular, these TQFTS give rise to the the condensation defects for the dual symmetry. When constructing the SymTFT we should also include these additional defects. We will refer to this as "condensation completion" of the SymTFT,[3] in analogy to the condensation completion of higher fusion categories in [108].

**Condensation Completion of SymTFTs.** In addition to the defects in (4) there will be also condensation- or theta-like defects that need to be included into the construction of the SymTFT. The condensation defects arise by condensing the defects of the dual symmetry, generated by $\mathbf{Q}_{p+1}^{(b)}$ on the defect $\mathbf{Q}_{d-p-1}^{(\hat{b})}$ that generates the symmetry $G^{(p)}$:

$$
\mathbf{C}\left(\mathbf{Q}_{d-p-1}^{(\hat{b})}\left(M_{d-p-1}\right), \mathbf{Q}_{p+1}^{(b)}\right) = \frac{1}{\left|H_{p+1}\left(M_{d-p-1}, \mathbb{Z}_n\right)\right|} \sum_{M_{p+1} \in H_{p+1}\left(M_{d-p-1}, \mathbb{Z}_n\right)} \mathbf{Q}_{p+1}^{(b)}\left(M_{p+1}\right) \mathbf{Q}_{d-p-1}^{(\hat{b})}\left(M_{d-p-1}\right). \quad (6)
$$

We can also condense these on other higher-form symmetry generators, as long as this is dimensionally consistent. Including this into the symTFT guarantees that all the topological defects (symmetry generators) and generalized charges will be realized.

These additional defects can be realized also by introducing localized couplings in the SymTFT, which correspond to coupling lower-dimensional DW type theories to the SymTFT. Taking into account all possible condensations this is

$$
S_{\text{SymTFT}} \supset n_p \int_{M_{d+1}} b_{p+1} \wedge d\,\widehat{b}_{d-p-1} + \sum_{k \geq 1} \int_{M_{d-k}} \left(b_{p+1} \wedge a_{d-k-p-1} + n_p a_{d-k-p-1} \wedge d\,\widehat{a}_p\right). \quad (7)
$$

**Example.** Consider an example of a QFT in $d = 4$ with $p = 0$ and $p' = 1$, corresponding to $\mathbb{Z}_N$ 0-form symmetry and a $\mathbb{Z}_M$ 1-form symmetries, respectively. The couplings are then

$$
\begin{aligned}
S_{\text{SymTFT}} = & \int_{M_5} N b_1 \wedge \widehat{b}_3 + M b_2 \wedge d\widehat{b}_2 + \int_{M_3} b_1 \wedge a_2 + N a_2 \wedge d\widehat{a}_0 \\
& + \int_{M_2} b_1 \wedge a_1 + N a_1 \wedge d\widehat{a}_0 + \int_{M_1} b_1 \wedge a_0 + N a_1 \wedge d\widehat{a}_0 \\
& + \int_{M_3} b_2 \wedge a_1 + M a_1 \wedge d\widehat{a}_1 + \int_{M_2} b_2 \wedge a_0 + M a_0 \wedge d\widehat{a}_1 \\
& + \int_{M_3} \widehat{b}_2 \wedge a_1' + M a_1' \wedge d\widehat{a}_1' + \int_{M_2} \widehat{b}_2 \wedge a_0' + M a_0' \wedge d\widehat{a}_1'. \quad (8)
\end{aligned}
$$

The topological defects are the standard ones for the 0-form and 1-form symmetries and their Pontryagin dual symmetries $\widehat{G}^{(2)}$ (generated by topological lines $\mathbf{Q}_1^{(\hat{b}_1)}$) and $\widehat{G}^{(1)}$ (generated by topological surfaces $\mathbf{Q}_2^{(\hat{b}_2)}$) as in (4) but in addition we also have condensation/theta-defects:

- $\mathbf{Q}_1^{(b_1)}$ on the defects $\mathbf{Q}_3^{(\hat{b}_3)}$ and $\mathbf{Q}_2^{(b_2)}$, as well as the trivial 1d defect.

- $\mathbf{Q}_2^{(b_2)}$ on the defects $\mathbf{Q}_3^{(\hat{b}_3)}$ and $\mathbf{Q}_2^{(b_2)}$.

---

[3]Mathematically we would always consider the SymTFT including all such condensations, however in the physics-literature, often, only the standard topological defects that are associated with the $p$-form symmetry and its dual are manifestly included.

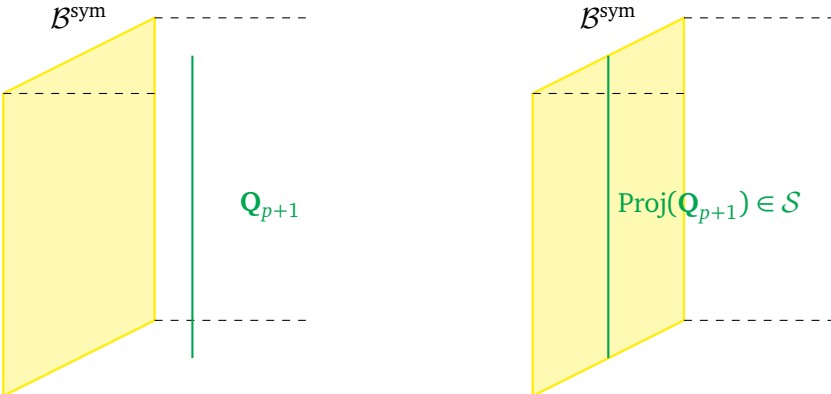

Figure 2: Symmetries from the SymTFT: the parallel projection of topological defects $\mathbf{Q}_{p+1}$ gives rise to topological defects on the symmetry boundary $\mathcal{B}^{\text{sym}}$. Put differently, the associated background fields, have Neumann boundary conditions. In general this is not the complete set of symmetry operators, but for abelian higher-form fields, for which we consider the SymTFT, this is the case.

**Generalizations.** The above corresponds to stacking with TQFTs without any cocycles. More generally we can of course include these. For example, for surface defects with $G^{(0)}$ symmetry we would construct 2d TQFTs with additional $\omega \in H^2(G^{(0)}, U(1))$. We can have additional discrete theta angles, for instance in the case of a 2d theory

$$b_1 \wedge a_1 + n a_1 \wedge d\hat{a}_0 \quad \rightarrow \quad b_1 \wedge a_1 + n a_1 \wedge d\hat{a}_0 + a_1 \wedge \theta(a_1), \tag{9}$$

where $\theta$ is a group homomorphism from $\widehat{G^{(0)}} \rightarrow G^{(0)}$. For an in depth discussion of these, see [67]; similar modifications also occur in appendix B of [61]. Similarly, for 3d TQFTs we can stack with theories that have non-trivial topological order, and more specific theory-dependent TQFTs. If there is an anomaly coupling, we can stack with TQFTs that have that same anomaly; we will encounter an example of this type in the form of the $U(1)_K$ CS-theory. These correspond more generally to twisted theta defects (which are not simply condensation defects).

## 2.3 Symmetries

The SymTFT for the theory $\mathcal{T}$ is constructed in such a way that the symmetry boundary $\mathcal{B}^{\text{sym}}$ has symmetry category given by $\mathcal{S}$ of the theory $\mathcal{T}$. Given a SymTFT we can recover the symmetry by projecting the bulk topological operators to the symmetry boundary (note: this is not true in general, e.g. for non-abelian group-symmetries, but in the present instance of abelian symmetries it is).

In the present instance we can simplify the analysis further, by stating that the boundary conditions are specified by a subset $\mathcal{L}$ of topological defects $\mathbf{Q}$ of the SymTFT, which have Dirichlet boundary conditions on $\mathcal{B}^{\text{sym}}$. This means the topological defects can end. Furthermore, requiring that this subset is mutually local and maximal defines a polarization.

All the defects in $\mathcal{L}$ end on the boundary and will define generalized charges – which we will discuss in the next subsection. The symmetry generators are the projections of the bulk topological operators onto the symmetry boundary. An in-depth analysis of all consistency conditions and possibilities in general was undertaken in [46]. The projection of a bulk topological defect onto the symmetry boundary is generically not a simple topological defect

$$\text{Proj}(\mathbf{Q}_{p+1}) = \bigoplus_i n_i S_{p+1}^{(i)}, \tag{10}$$

where $S_{p+1}^{(i)}$ are the simple defects.

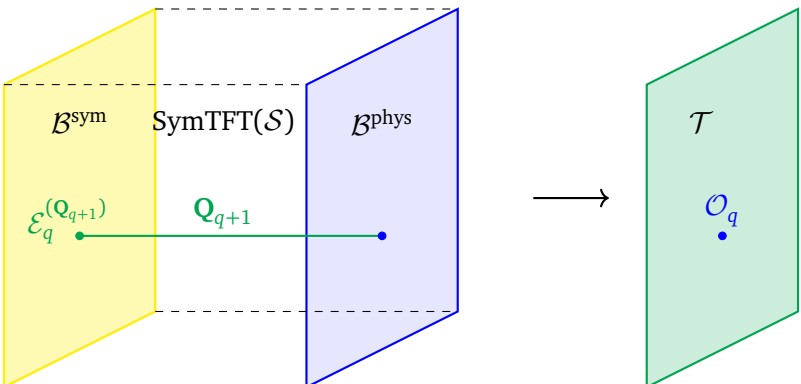

Figure 3: Generalized charges from bulk topological defects that end on the symmetry and physical boundaries: genuine $q$-charge $\mathcal{O}_q$. The left hand side shows the SymTFT sandwich, with the bulk topological operator $\mathcal{O}_{q+1}$ ending on both physical and symmetry boundaries. After interval compactification it gives rise to a $q$-charge in the theory $\mathcal{T}$.

**Example.** Consider a 0-form symmetry $\mathcal{S} = \mathbb{Z}_N^{(0)}$. The topological operators that generate this symmetry project simply to the generators on $\mathcal{B}^{\mathrm{sym}}$

$$\mathrm{Proj}(\mathbf{Q}_{d-1}^{(\hat{b}_{d-1})}) = S_{d-1}^{(1)}, \tag{11}$$

of the 0-form symmetry, i.e. the objects in the higher-fusion category

$$\mathcal{S}: \quad \{S_{d-1}^{(i)}, i = 0, \cdots, N-1\}. \tag{12}$$

In addition we also have condensation defects for the dual symmetry

$$\mathbf{C}\left(\mathbf{Q}_{d-1}^{(\hat{b}_{d-1})}, \mathbf{Q}^{(b_1)}\right). \tag{13}$$

These have fusion

$$\mathbf{C}\left(\mathbf{Q}_{d-1}^{(\hat{b}_{d-1})}, \mathbf{Q}^{(b_1)}\right) \otimes \mathbf{C}\left(\mathbf{Q}_{d-1}^{(\hat{b}_{d-1})}, \mathbf{Q}^{(b_1)}\right) = N\mathbf{C}\left(\mathbf{Q}_{d-1}^{(\hat{b}_{d-1})}, \mathbf{Q}^{(b_1)}\right). \tag{14}$$

Projecting these to the simple objects in $\mathcal{S}$, i.e. $S_{d-1}^{(0)}$ and $S_{d-1}^{(1)}$, shows that the only projection that is consistent with the fusion is

$$\mathrm{Proj}\left(\mathbf{C}\left(\mathbf{Q}_{d-1}^{(\hat{b}_{d-1})}, \mathbf{Q}_1^{(b_1)}\right)\right) = NS_{d-1}^{(1)}. \tag{15}$$

## 2.4 Generalized charges

The charges under generalized symmetries were recently identified as being simply the topological defects of the associated SymTFT [46]. This applies to several invertible and non-invertible symmetries and has been shown to hold in many such instances [45, 101, 110]. Particularly interesting is the observation that there are generalized charges even for invertible higher-form symmetries. Again the SymTFT plays the central tool to succinctly characterize the charges.

As proposed in [45], we refer to a $q$-dimensional, not necessarily topological, defect operator $\mathcal{O}_q$ that is charged under a generalized symmetry as a $q$-charge. The statement of [45, 110] is that for an invertible higher-form symmetry $G^{(p)}$

$$\text{Genuine } q\text{-charges } \mathcal{O}_q \quad \longleftrightarrow \quad (q+1) - \mathbf{Rep}\left(G^{(p)}\right), \tag{16}$$

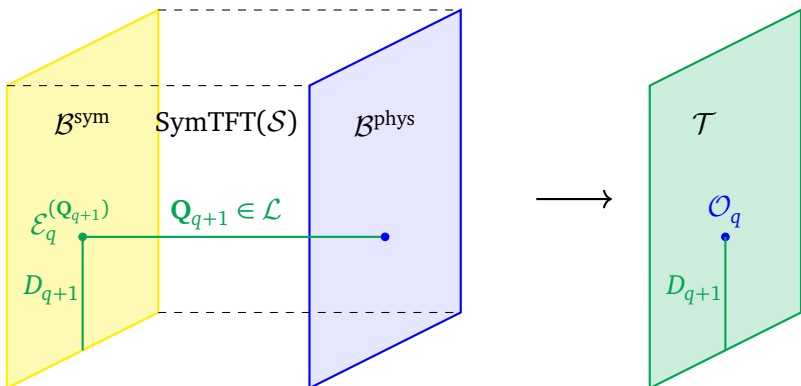

Figure 4: Twisted sector operators: L-shape projection of a bulk topological defect $\mathbf{Q}_{q+1}$ onto the symmetry boundary, creates a junction $\mathcal{E}_p$ which is attached to a topological defect $D_{q+1}$.

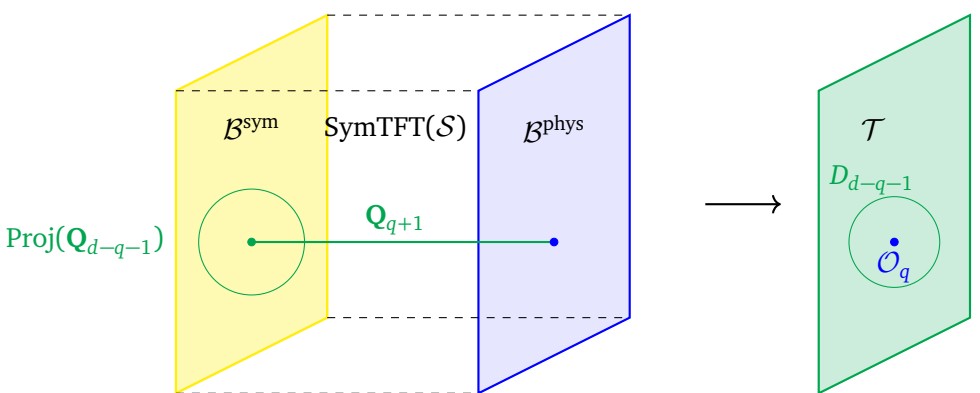

Figure 5: Generalized symmetry acting on generalized charge via linking. The topological operator $\mathbf{Q}_{p+1}$ in the bulk SymTFT ends and gives rise to the (genuine) $q$-charge $\mathcal{O}_q$ in $\mathcal{T}$. In turn, the topoglogical operator $\mathbf{Q}_{d-q-1}$ projects onto the symmetry boundary and gives rise to a symmetry generator after the interval compactification. The non-trivial linking of these topological defects in the SymTFT results in the generalized charge.

where the right hand side is the fusion higher-category of higher-representations of $G^{(p)}$ (see e.g. [45,67,106] for physics-motivated discussions of these categories). Here $q = 0, \cdots, d-2$. The genuine $q$-charges are not attached to $(q+1)$-dimensional defects (topological or not), see figure 3, and arise after interval compactification as endpoints of bulk topological operators $\mathbf{Q}_{q+1}$ that end on both physical and topological boundaries in $q$-dimensional operators.

In addition to genuine charges, there can be non-genuine (attached at the end of $\mathcal{O}_{q+1}$) and twisted sector (attached to the end of topological $S_{q+1}$ defects) $q$-charges. In the SymTFT picture, the twisted sector charges arise from projecting L-shaped bulk topological defects, see figure 4: we project a bulk topological defect onto the symmetry boundary in an L-shape, which results in a junction operator $\mathcal{E}_p^{(\mathbf{Q}_{q+1})}$ attached to a topological defect $D_{q+1} \in \mathcal{S}$. After interval compactification this is a $q$-charge $\mathcal{O}_q$, attached to a topological $(q+1)$-dimensional defect $D_{q+1}$, which is thus a twisted sector operator.

## 2.5 Charges from linking

Finally let's consider the action of symmetry defects on charges. This arises by computing the linking of bulk topological defects projected onto the symmetry boundary. There is the

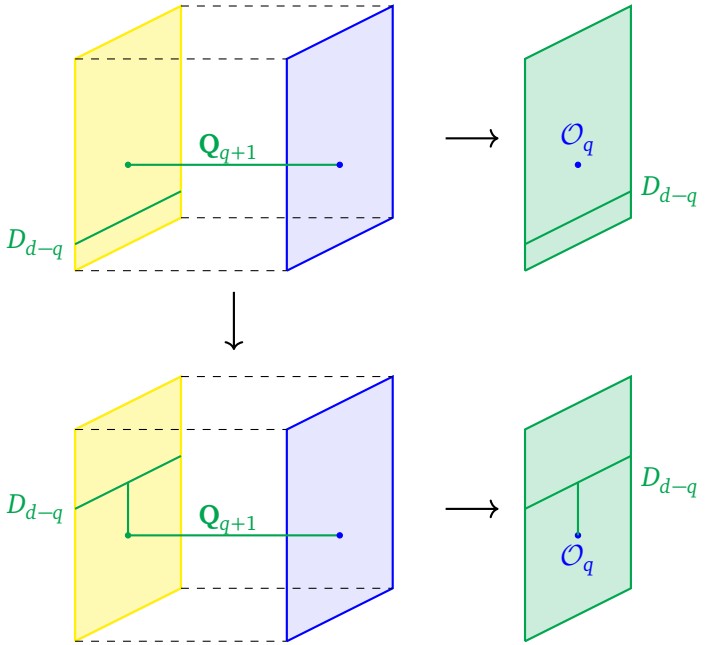

Figure 6: Action of non-invertible symmetries: mapping genuine to non-genuine operators. The genuine defect $\mathcal{O}_q$ is acted upon by the topological defect $D_{d-q}$, which maps it to a non-genuine operator, with an attachment of a $(q+1)$-dimensional operator.

standard linking action of higher-form symmetry generators on defects, which follows from the linking in (4), where the mutually non-local defects are either Neumann or Dirichlet:

$$\text{Proj}(\mathbf{Q}_{d-p-1}^{\hat{b}})(\partial\mathbf{Q}_{p+1}|_{\mathcal{B}^{\text{sym}}}) \longrightarrow D_{d-p-1}(\mathcal{O}_p) = q_{\mathcal{O}_p}\mathcal{O}_p\,, \tag{17}$$

where the arrow denotes the interval compactification, and $q$ is the charge under the higher-form symmetry. This is shown in figure 5. The configuration shown here has various generalizations which were discussed in [46]. For the constructions in string theory, this is the most general setup we will require.

Non-invertible defects can also act by taking an operator in between genuine and twisted sectors. For example, passing the topological defect for a non-invertible symmetry through the end of the bulk topological defect, results in a twisted sector defect, as shown in figure 6. This action is well-known in various contexts of non-invertible symmetries in a variety of dimensions [56,111] and was realized in terms of branes as Hanany-Witten moves in [1]. We will provide various generalizations in this paper.

# 3 SymTFT and its topological defects from branes

## 3.1 Geometric engineering, holography, and the SymTFT

Let us first describe how a holographic or geometric engineering construction in string theory leads to a bulk topological description of the abelian and finite symmetry sector of a relative QFT living at the boundary. In most cases[4] this bulk topological description will play the role

---

[4]In some cases from string theory we obtain a theory that is not a SymTFT, because generically there are no topological boundary conditions. In these instances one cannot generically have an absolute theory at the boundary. For instance, this is indeed the case for 6d $(2,0)$ theories, which are generically relative theories.

of the SymTFT, once suitable boundary conditions are imposed to make the theory absolute. One of the main tasks of this paper is to incorporate branes that realize symmetry defects and generalized charges into this framework in a very general fashion.

**Flux Sector of Supergravity.** First of all we focus on the flux sector of 10/11-dimensional supergravity (depending on a string or M-theory starting point, respectively), and in particular in the Bianchi identities and equations of motion that the fluxes will satisfy. We will also include brane sources, which magnetically charge branes. This is because the bulk gauge potentials will provide background fields for the finite, abelian generalized symmetries. Their holonomies will also give rise to the topological operators defining the generalized symmetries. This information is equivalently encoded in the brane actions [1]. We will not consider the rest of the supergravity action which includes scalar fields, the metric and other modes. This is justified by the fact that we will be interested in the physics of flat discrete gauge fields (vanishing fluxes on-shell), which give non-trivial holonomies. The dilaton and the metric equations of motion will depend only on modes for which the fluxes are non-vanishing. The flux action of 10/11-dimensional supergravity has two pieces, a kinetic term and cubic Chern-Simons topological coupling, which can depend on one or more fluxes. It will be useful to adopt a formulation in which we include both the supergravity fluxes and their Hodge duals in 10/11 dimensions in a democratic way, as detailed below. However before presenting the democratic formulation let us describe the precise relation between the SymTFT and the bulk theory in holographic and geometric/brane engineering setups.

**Dimensional Reduction of Flux Sector.** The second step consists of dimensionally reducing the flux action with brane sources on the geometry dictated by the holographic description or the boundary at infinity of the geometric engineering setup. Concretely let

$$\partial \mathbf{X}_{n+1} = L_n \,, \tag{18}$$

where either the holographic solution is $M_{d+1} \times L_n$ or the geometric engineering corresponds to a compactification on $\mathbf{X}_{n+1}$. This is given by specifying the following geometric background, which is a solution of the supergravity equations of motion

$$M_{n+d+1} = M_{d+1} \times L_n \,, \tag{19}$$

where

$$n + d + 1 = \left\{ \begin{matrix} 10 \\ 11 \end{matrix} \right\} = D + 1 \,, \tag{20}$$

depending on string or M-theory. The QFT is $d$-dimensional, and we start with a $D+1$ dimensional supergravity theory. In the resulting lower-dimensional theory we generically obtain an action that consists of kinetic terms as well as cubic Chern-Simons couplings. This theory is defined on $M_{d+1}$, which has a $d$-dimensional boundary. It can be $\text{AdS}_{d+1}$ or more general spaces with a boundary which sits at infinity.

**Topological Limit.** In this dimensionally reduced theory, we are interested in very long-distance regimes, which are realized very close to the boundary at infinity [37]. In this sense, we can implement a derivative expansion of the kinetic and topological couplings of the flux action. The lowest derivatives dominate, which usually consists of topological couplings when they are non-trivial. This is also valid for the dimensionally reduced brane action at very large-distances, where the kinetic terms obtained from expanding the DBI part of the action are subleading with respect to the topological couplings. In addition, the Wess-Zumino part will always provide topological couplings when non-trivial.

The main reason why we can truncate the dimensionally reduced bulk and brane action is that we really want to focus on finite, abelian, global symmetries. The gauge fields of these symmetries in the bulk, which are flat, have vanishing flux on-shell. The modes which we remove in the truncation do not couple to the symmetry sector described by flat fields, and therefore can be ignored. For instance, the kinetic term for the dimensionally reduced fluxes will be non-trivial only for the non-vanishing part of the fluxes and therefore can be ignored [37]. To reconcile the large-distance limit and the truncation to the flat finite abelian gauge fields, we can say that from the point of view of large-distances, i.e. the close to the boundary limit, the modes with non-vanishing flux are massive, and they can be integrated out, effectively leaving only the topological couplings describing the non-trivial fluctuation of flat fields and their non-trivial holonomies.

Finally, having truncated the dimensionally-reduced bulk and brane action to their topological sector, which describes the physics of finite abelian flat gauge fields, we can deform the space without changing its topology such as

$$M_{d+1} \to M_d \times \mathbb{R}. \tag{21}$$

**SymTFT, Boundaries and Holography/String Theory.** We can then connect this to the standard notion of SymTFT: The choice of absolute theory, i.e. a choice of polarization, will be implemented by partially compactifying the $\mathbb{R}$ direction, i.e the semi-infinite $[0, \infty)$ interval [39]

$$S_{\text{top}}|_{\text{polarization}} = S_{\text{SymTFT}}. \tag{22}$$

We can then think of the physical boundary (not necessarily gapped boundary) $\mathcal{B}^{\text{phys}}$ as placed at $r = 0$. Likewise, the symmetry boundary, i.e. topological boundary condition, $\mathcal{B}^{\text{sym}}$, is at $r = \infty$.

The position of the two boundaries reflects what appears to be the position of the physical theory and the topological boundary, respectively, both in holography and string theory geometric engineering. The interpretation of the coordinate $r$ and therefore of these precise positions, depends on the metric. For simplicity, for AdS we work in hyper-polar coordinates

$$ds^2(\text{AdS}_{d+1}) \sim r^2 ds^2(\mathbb{R}^{1,d}) + r^{-2} dr^2, \tag{23}$$

(the conformal boundary is located at $r = \infty$), while for geometric engineering setups we identify $r$ with the real cone direction in the space $\mathbf{X}_{n+1}$

$$ds^2_{\mathbf{X}_{n+1}} = dr^2 + r^2 ds^2_{L_n(\mathbf{X}_{n+1})}, \tag{24}$$

with the link $L_n$ ($r = 0$ is the singularity at the tip of the cone).

In particular, in holographic setups we associate $\mathcal{B}^{\text{phys}}$ with the origin of AdS space ($r = 0$), and not with the conformal boundary ($r = \infty$). This might seem counter-intuitive, since the CFT lives on a spacetime isomorphic to the conformal boundary. Our perspective stems from the fact that the CFT is *dual* to the dynamics of the gravity theory in the bulk of AdS spacetime.

The presence of the conformal boundary on the gravity side teaches us that we need to supplement the bulk action with boundary conditions for the supergravity fields, in order to obtain a gravitational system that is holographically dual to a QFT. This is quantified in the following key relation of the AdS/CFT correspondence [112]

$$Z_{\text{sugra}}[(\text{b.c. for } \varphi) = J_\varphi] = \left\langle e^{\int \mathcal{O}_\varphi J_\varphi} \right\rangle_{\text{CFT}}, \tag{25}$$

where $\mathcal{O}_\varphi$ is the gauge-invariant operator dual to the supergravity field $\varphi$. In the string theory origin of holography, where we look at the near-horizon limit of some back-reacted brane

system, the theory that we are describing is the one living on the stack of branes in some low-energy decoupling limit. Therefore we can practically consider the physical theory to live at $r = 0$ (radial position of the stack of branes) and the boundary of AdS to be at $r = \infty$. Once we truncate to the topological sector the latter becomes $\mathcal{B}^{\text{sym}}$. In geometric engineering, where the compactification space $\mathbf{X}_{n+1}$ is a real cone over a link $L_n(\mathbf{X}_{n+1})$, this works in a very similar way: $\mathcal{B}^{\text{phys}}$ is placed at $r = 0$ and $\mathcal{B}^{\text{sym}}$ at $r = \infty$.

The real difference between the SymTFT and holography/string theory is that in the latter we cannot really perform the partial compactification of the semi-interval direction between $[0, \infty)$. The main reason is that before truncating to the topological sector, we have gravity and other fields related to the full string theory construction in the bulk, as well as non-local excitations. All these are not necessarily related to the symmetry sector. In particular we cannot deform or compactify the space as we like. Indeed, in holography the geometry of the radial direction provides the non-trivial correspondence between the gravitational theory in the bulk and the QFT.

Rather, we have two different procedures to specify an absolute theory in string theory/holography, and in the SymTFT: in the former, we choose boundary conditions at $r = \infty$; in the latter, we perform the interval compactification. Heuristically, in string theory/holography (quantum) gravity mediates between the topological boundary and the choices of the boundary conditions at infinity with the theory living at $r = 0$ without the need of an actual interval compactification. Let us emphasize, however, that these two procedures can be connected to each other on the string/holography side, if we perform a truncation to the topological sector. Once the truncation is performed, we are indeed free to specify $M_{d+1}$ as in (21) and establish a direct link with the SymTFT interval picture.

**Singleton Theory.** In many string-theory/holographic setup we have to deal with the center of mass degree of freedom. For instance we could consider the stack of branes before taking the near-horizon limit. In this case the theory that is realized on the brane-stack is an absolute theory (with $U(N)$ gauge group for $N$ D-branes). On the gravity side, the $\mathfrak{u}(1)$ in $\mathfrak{u}(N) \cong \mathfrak{su}(N) \oplus \mathfrak{u}(1)$ is described by what is called the singleton mode [109]. This mode has been analyzed in detail in the case of the $\mathcal{N} = 4$ $\mathfrak{su}(N)$ SYM and its holographic construction. In particular, as it was shown by [113], this is a mode in the KK supergravity spectrum (entire supersymmetric multiplet, which contains the two forms $B_2, C_2$), which comes from an expansion in spherical harmonics of $S^5$ and satisfies specific conditions, that are different from the other bulk fields. This bulk multiplet is dual to a $U(1)$ gauge field in 4d (the center of mass of the stack of brane). The extra conditions on this bulk multiplet, which we do no repeat here, make the $U(1)$ gauge field pure gauge that is eaten by the combination of $B_2, C_2$ that has Dirichlet boundary conditions at $r = \infty$. In terms of bulk fields, the BF topological coupling between $(B_2, C_2)$ can be seen as Stückelberg mechanism for the combination of $(B_2, C_2)$ that becomes Dirichlet, in the spirit of [114]. Therefore the singleton mode is pure gauge in the bulk and at the topological boundary, and it is not dual to any propagating physical mode of the $\mathcal{N} = 4$ $\mathfrak{su}(N)$ SYM. The near-horizon limit decouples the center of mass mode of a stack of branes by making it pure gauge and hence non-propagating in the bulk. The mode then localizes on the boundary at $r = \infty$. We expect this to generalize in the context discussed in this paper, and it would be insightful to repeat this analysis in other contexts or to generalize it.

## 3.2 A democratic formulation for fluxes and brane sources

It is convenient to work in a democratic formulation, including both the fluxes and their Hodge duals. Let us describe our notation in a general setting, for a theory defined in $D+1$ dimensions. Here $D + 1 = 10, 11$ depending on string or M-theory.

**Magnetic Sources.** Let $F^{(i)}$ denote the entire collection of fluxes in $D+1$ dimensions, labeled by $(i)$ (unrelated to the form degree). In order to avoid redundancies, we consider only magnetic sources. (An electric source for a given flux $F^{(i)}$ is a magnetic source for the Hodge dual of $F^{(i)}$.) We denote the magnetic source for $F^{(i)}$ by $J^{(i)}$. If the source is localized, $J^{(i)}$ is delta-function supported on a submanifold $\mathcal{W}^{(i)}$. The magnetic source modifies the Bianchi identity for $F^{(i)}$,

$$dF^{(i)} = J^{(i)} = \delta(\mathcal{W}^{(i)}), \qquad D+1-\dim\mathcal{W}^{(i)} = \deg J^{(i)} = \deg F^{(i)} + 1. \qquad (26)$$

At the moment, we are neglecting the effect of possible Chern-Simons terms. Those will be considered below.

**Linking.** The $(D+1)$-dimensional linking number between two magnetic sources $J^{(i)} = \delta(\mathcal{W}^{(i)})$ and $J^{(j)} = \delta(\mathcal{W}^{(j)})$ is defined as

$$\mathcal{L}_{D+1}(\mathcal{W}^{(i)}, \mathcal{W}^{(j)}) = \int_{M_{D+1}} J^{(i)} \wedge d^{-1} J^{(j)} = \int_{M_{D+1}} dF^{(i)} \wedge F^{(j)}, \qquad (27)$$

where the dimensions of $\mathcal{W}^{(i)}$, $\mathcal{W}^{(j)}$ inside the total $(D+1)$-dimensional space satisfy

$$\dim\mathcal{W}^{(i)} + \dim\mathcal{W}^{(j)} = D. \qquad (28)$$

The linking number has the symmetry property

$$\mathcal{L}_{D+1}(\mathcal{W}^{(i)}, \mathcal{W}^{(j)}) = (-1)^{1+\dim\mathcal{W}^{(i)}\dim\mathcal{W}^{(j)}} \mathcal{L}_{D+1}(\mathcal{W}^{(j)}, \mathcal{W}^{(i)}), \qquad (29)$$

which follows from integration by parts in the last integral in (27), together with (28).

The compact notation with the symbol $d^{-1}$ introduced in (27), and used below, is understood as follows. We assume that $\mathcal{W}^{(i)}$, $\mathcal{W}^{(j)}$ are homologically trivial, $\mathcal{W}^{(i)} = \partial\mathcal{S}^{(i)}$, $\mathcal{W}^{(j)} = \partial\mathcal{S}^{(j)}$. The chains $\mathcal{S}^{(i)}$, $\mathcal{S}^{(j)}$ are usually called Seifert (hyper)surfaces [115]. We can then write $\delta(\mathcal{W}^{(i)}) = d\delta(\mathcal{S}^{(i)})$, $\delta(\mathcal{W}^{(j)}) = d\delta(\mathcal{S}^{(j)})$ and interpret (27) as

$$\mathcal{L}_{D+1}(\mathcal{W}^{(i)}, \mathcal{W}^{(j)}) = \int_{M_{D+1}} d\delta(\mathcal{S}^{(i)}) \wedge \delta(\mathcal{S}^{(j)}) = \int_{M_{D+1}} \delta(\mathcal{W}^{(i)}) \wedge \delta(\mathcal{S}^{(j)}) = \mathcal{W}^{(i)} \cdot_{M_{D+1}} \mathcal{S}^{(j)}, \qquad (30)$$

where $\mathcal{W}^{(i)} \cdot_{M_{D+1}} \mathcal{S}^{(j)}$ is the number of intersection points of $\mathcal{W}^{(i)}$ and $\mathcal{S}^{(j)}$ inside $M_{D+1}$, counted with signs depending on orientation.

It is also useful to consider a slight generalization of the notion of linking discussed above, along the following lines. Let's suppose that the supports $\mathcal{W}^{(i)}$, $\mathcal{W}^{(j)}$ span some common directions along some space $\mathcal{V}$, while extending in distinct directions in the rest of spacetime, i.e. $\mathcal{W}^{(i,j)} = \mathcal{V} \times \mathcal{U}^{(i,j)}$. We can define the linking of $\mathcal{W}^{(i)}$ and $\mathcal{W}^{(j)}$ using the same formula as above, but focusing on the $\mathcal{U}^{(i)}$ and $\mathcal{U}^{(j)}$ parts, whose dimensions are such that

$$\dim(\mathcal{W}^{(i)} \cup \mathcal{W}^{(j)}) = D. \qquad (31)$$

This notion of linking is naturally associated to Hanany-Witten moves in string/M-theory, as we discuss in greater detail in section 4.

**Topological Action in $D+2$ dimensions.** The Bianchi identities (26) can be derived from a topological action in $D+2$ dimensions,

$$S_{D+2} = \sum_{i,j} \int_{M_{D+2}} \left[ \frac{1}{2}\kappa_{ij} F^{(i)} \wedge dF^{(j)} - \kappa_{ij} F^{(i)} \wedge J^{(j)} \right]. \qquad (32)$$

This is regarded as a functional of the fluxes $F^{(i)}$ (as opposed to the associated gauge potentials). Extending from $D + 1$ to $D + 2$ dimensions allows us to deal efficiently with gauge invariance. The quantity $\kappa_{ij}$ is a constant non-degenerate matrix, satisfying

$$\kappa_{ij} = 0, \qquad \text{if } \deg F^{(i)} + \deg F^{(j)} \neq D + 1,$$
$$\kappa_{ij} = (-1)^{[\deg F^{(i)}+1][\deg F^{(i)}+1]}\kappa_{ji}. \tag{33}$$

The symmetry property in the second equality reflects the freedom to integrate by parts in the $F^{(i)} \wedge dF^{(j)}$ term. It also ensures that, upon variation of $F^{(i)}$, the topological action yields

$$\sum_j \kappa_{ij}\big(dF^{(j)} - J^{(j)}\big) = 0, \tag{34}$$

which, by non-degeneracy of $\kappa_{ij}$, is equivalent to $dF^{(j)} = J^{(j)}$, in agreement with our parametrization of magnetic sources.

Let us stress that the relations obtained upon variation of the $(D+2)$-dimensional topological action are still to be supplemented by Hodge duality relations in $D + 1$ dimensions. This is illustrated in the following example.

**Example: Generalized Maxwell in $D + 1$ dimensions.** Let us consider a generalized Maxwell theory in $D + 1$ dimensions, with a single $p$-form gauge potential $a$, with field strength $f$. In the absence of sources, the action reads

$$S_{D+1} = \int_{M_{D+1}} \frac{1}{2e^2} f \wedge *f, \qquad \deg f = p + 1. \tag{35}$$

The Bianchi identity and equation of motion read

$$df = *\mathcal{J}^{(\mathrm{m})}, \qquad e^{-2}d*f = *\mathcal{J}^{(\mathrm{e})}, \tag{36}$$

where $e$ is the gauge coupling, $*$ is the Hodge star in $D+1$ dimensions, $\mathcal{J}^{(\mathrm{e})}$ is the electric source for $a$, and $\mathcal{J}^{(\mathrm{m})}$ is the magnetic source for $a$. In the democratic formulation we introduce two field strengths and two magnetic currents,[5]

$$F^{(1)} = f, \qquad F^{(2)} = e^{-2}*f, \qquad J^{(1)} = *\mathcal{J}^{(\mathrm{m})}, \qquad J^{(2)} = *\mathcal{J}^{(\mathrm{e})}. \tag{37}$$

The topological action in $d + 2$ dimensions is of the form quoted above, with labels $i, j$ ranging from 1 to 2, and with $\kappa_{ij}$ matrix

$$\kappa_{ij} = \begin{pmatrix} 0 & 1 \\ (-)^{(p+2)(D+1-p)} & 0 \end{pmatrix}. \tag{38}$$

More explicitly,

$$S_{D+2} = \int_{M_{D+2}} \left[ F^{(1)} \wedge dF^{(2)} - F^{(1)} \wedge J^{(2)} - (-)^{(p+2)(D+1-p)}F^{(2)} \wedge J^{(1)} \right], \tag{39}$$

which, upon variation with respect to $F^{(1)}$, $F^{(2)}$ reproduces $dF^{(1)} = J^{(1)}$, $dF^{(2)} = J^{(2)}$. The $(D + 1)$-dimensional Hodge duality relation that has to be supplemented is

$$*F^{(1)} = e^2 F^{(2)}. \tag{40}$$

---

[5]Notice in particular that in our notation the currents $J^{(i)}$ are closed forms, as opposed to co-closed forms, which is perhaps a more common convention for conserved currents in the literature.

**Inclusion of Chern-Simons terms.** We can include non-trivial Chern-Simons terms by modifying the $(D + 2)$-dimensional topological action. The required modification is a polynomial in the $F^{(i)}$ fluxes, denoted $\text{CS}(\{F^{(i)}\})$,

$$S_{D+2} = \int_{M_{D+2}} \left[ \frac{1}{2} \sum_{i,j} \kappa_{ij} F^{(i)} \wedge dF^{(j)} + \text{CS}(\{F^{(i)}\}) - \sum_{i,j} \kappa_{ij} F^{(i)} \wedge J^{(j)} \right]. \tag{41}$$

Our notation stems from the fact that $\text{CS}(\{F^{(i)}\})$ is a closed $(D + 2)$-form which is related to the physical Chern-Simons couplings in $D + 1$ dimensions by descent,

$$S_{D+1}^{\text{top}} = \int_{M_{D+1}} I_{D+1}^{(0)}, \qquad dI_{D+1}^{(0)} = \text{CS}(\{F^{(i)}\}). \tag{42}$$

Varying (41) with respect to $F^{(i)}$ we get

$$\sum_j \kappa_{ij}(dF^{(j)} - J^{(j)}) + \frac{\partial \text{CS}(\{F^{(k)}\})}{\partial F^{(i)}} = 0. \tag{43}$$

The notation introduced in this section is summarized in table 1.

**Actions for Type II and M-theory.** To make the above discussion more concrete, let us describe the $(D + 2)$-dimensional topological actions for type II ($D + 1 = 10$) and M-theory ($D + 1 = 11$). They are of the form (41) with

$$\text{IIA:} \begin{cases} F^{(i)} = (F_0, F_2, F_4, F_6, F_8, F_{10}, H_3, H_7), \\ S_{11} = \int_{M_{11}} \left[ F_0 dF_{10} - F_2 dF_8 + F_4 dF_6 + H_3 dH_7 - H_3 \left( F_0 F_8 - F_2 F_6 + \frac{1}{2} F_4^2 + X_8 \right) \right], \end{cases}$$

$$\text{IIB:} \begin{cases} F^{(i)} = (F_1, F_3, F_5, F_7, F_9, H_3, H_7), \\ S_{11} = \int_{M_{11}} \left[ F_1 dF_9 - F_3 dF_7 + \frac{1}{2} F_5 dF_5 + H_3 dH_7 + H_3 (F_1 F_7 - F_3 F_5) \right], \end{cases} \tag{44}$$

$$\text{M:} \begin{cases} F^{(i)} = (G_4, G_7), \\ S_{12} = \int_{M_{12}} \left[ G_4 dG_7 - \frac{1}{6} G_4^3 - G_4 X_8 \right]. \end{cases}$$

Here we have suppressed wedge products for clarity. For simplicity, we have recorded the actions without the source terms. We refer the reader to appendix A for further details and for a discussion of the sources $J^{(i)}$. The 8-form $X_8$ is a higher-derivative correction constructed with the curvature form [116, 117], see appendix A. The topological actions (44), in $D + 2$ dimensions are supplemented by Hodge duality relations in $D + 1$ dimensions:

$$\begin{aligned} \text{Type II:} \quad & H_7 = e^{-2\phi} *_{10} H_3, \qquad F_p = (-1)^{\lfloor \frac{p}{2} \rfloor} *_{10} F_{10-p}, \\ \text{M-theory:} \quad & G_7 = - *_{11} G_4, \end{aligned} \tag{45}$$

where in Type II, $\phi$ is the dilaton and we work in string frame in natural units. A democratic formulation for type II based on 11d Chern-Simons theories is presented in [47]; a democratic formulation based on a non-topological 10d (pseudo)action is presented in [118].

Table 1: Summary of Notation: we consider a supergravity theory in $D + 1 = 10, 11$, dimensions with fluxes $F^{(i)}$. The auxiliary topological action is formulated in $D + 2$ dimensions. The magnetic source for $F^{(i)}$ is denoted by $J^{(i)}$.

| | |
|---|---|
| Dimensions | QFT dim $= d$; SymTFT dim $= d + 1$; sugra dim $= D + 1 = 10, 11$ <br> internal space dim $= n$; $D + 1 = n + d + 1$ |
| Top Action | $S = \int_{M_{D+2}} \left[ \dfrac{1}{2} \sum_{i,j} \kappa_{ij} F^{(i)} \wedge dF^{(j)} + \mathrm{CS}(\{F^{(i)}\}) - \sum_{i,j} \kappa_{ij} F^{(i)} \wedge J^{(j)} \right]$ |
| Magnetic Sources | $dF^{(i)} - \sum_{j} (\kappa^{-1})^{ij} \dfrac{\mathrm{CS}(\{F^{(k)}\})}{\partial F^{(j)}} = J^{(i)}$, $\qquad J^{(i)}$ supported on $\mathcal{W}^{(i)}$ <br><br> $D + 1 - \dim \mathcal{W}^{(i)} = \deg J^{(i)} = \deg F^{(i)} + 1$ |
| Linking | $\mathcal{L}_{D+1}(\mathcal{W}^{(i)}, \mathcal{W}^{(j)}) = \int_{M_{D+1}} J^{(i)} \wedge d^{-1} J^{(j)} = \int_{M_{D+1}} dF^{(i)} \wedge F^{(j)}$ <br><br> $\dim \mathcal{W}^{(i)} + \dim \mathcal{W}^{(j)} = D$, $\quad \left( \dim(\mathcal{W}^{(i)} \cup \mathcal{W}^{(j)}) = D \text{ for HW linking} \right)$ |

**$(D + 2)$-dimensional Topological Action and Dimensional Reduction.** Recall that we are interested in studying setups in which the physical spacetime in 10/11 dimensions is of the form (19). This corresponds to $D = d + n$. The auxiliary topological action in $D + 2$ dimensions is formulated on a spacetime of the form

$$M_{D+2} = M_{d+2} \times L_n, \tag{46}$$

where the external spacetime $M_{d+1}$ has been extended to an auxiliary $M_{d+2}$, while the internal geometry remains $L_n$. Our task is to integrate $S_{D+2}$ on $L_n$ to obtain a topological action $S_{d+2}$. Next, we reconstruct the physical SymTFT action $S_{d+1}$ from $S_{d+2}$,

$$\text{auxiliary top. action} \quad S_{d+2} = \int_{L_n} S_{D+2} \quad \longrightarrow \quad \text{SymTFT action} \quad S_{d+1}. \tag{47}$$

These steps are exemplified below in a variety of setups.

## 3.3 BF-terms from branes

The first aspect to address is how to generate the BF terms of the SymTFT.[6] We adopt the strategy described at the end of the previous section: we work on a $(D + 2)$-dimensional spacetime of the form (46). The fluxes can have background values on topologically non-trivial cohomologies of the internal manifold $L_n$, and in the reduction ansatz they are generically expanded along representatives of these cohomologies.

Generically we face two possibilities that generate BF-terms:

1. BF-terms from the term $\mathrm{CS}(\{F^{(i)}\})$ in (41)

2. BF-terms from the term $\kappa_{ij} F^{(i)} \wedge dF^{(j)}$ in (41).

Let us analyze each possibility in turn.

---

[6] In some cases these can be pure Chern-Simons term like $c_3 dc_3$ in $7d$. When this happens the theory is not properly defined as a SymTFT, because of the absence of gapped boundary conditions. An example of this is provided by 6d $(2,0)$ theories.

**BF-terms from CS($\{F^{(i)}\}$).**   The first possibility is when the Chern-Simons functional descend to a non-trivial quadratic wedge product of two fluxes upon compactification on $L_n$. For instance this is the case when there are non-trivial background fluxes on $L_n$ or $F^{(i)}$ are expanded on non-trivial cohomologies representatives.

Let us describe schematically the general mechanism for generating BF-terms in this case. The relevant ansatz for the higher-dimensional fluxes and sources reads

$$F^{(i)} = F^{(i)}_{\text{bkrg}} + \sum_a f^{(ia)} \wedge \omega^{(a)}, \qquad J^{(i)} = \sum_a j^{(ia)} \wedge \omega^{(a)}. \tag{48}$$

Here $\omega^{(a)}$ are closed internal forms, with integral periods, representing cohomology classes on $L_n$, enumerated by the label $a$. The quantities $f^{(ia)}$ are external fluxes, while $F^{(i)}_{\text{bkrg}}$ denotes a possible non-zero background value for the $F^{(i)}$ flux. The latter is also proportional to the volume form of cycles of $L_n$ and always integrates to an integer on these cycles because of flux quantization. Finally, we use $j^{(ia)}$ to denote the external parts of the higher-dimensional sources $J^{(i)}$.

If we start from the topological action (41) in $D+2$ dimensions and integrate over $L_n$, we obtain a topological action in $d+2$ dimensions with couplings of the form (we suppress wedge products for brevity)

$$S_{d+2} = \int_{M_{d+2}} \sum_{i,j,a,b} \left[ \frac{1}{2} \kappa_{(ia)(jb)} f^{(ia)} d f^{(jb)} - \kappa_{(ia)(jb)} f^{(ia)} j^{(jb)} + \alpha_{(ia)(jb)} f^{(ia)} f^{(jb)} \right] + \cdots. \tag{49}$$

On the one hand, the constants $\kappa_{(ia)(ib)}$ are determined from the original constants $\kappa_{ij}$ in (41) and the intersection pairing of the $\omega^{(a)}$ forms on $L_n$. On the other hand, the terms $\alpha_{(ia)(jb)} f^{(ia)} f^{(jb)}$ originate from cubic terms in CS($\{F^{(i)}\}$) in (41), with the constants $\alpha_{(ia)(jb)}$ determined as integrals over $L_n$ of internal top forms constructed with the background fluxes $F^{(i)}_{\text{bkgr}}$ and with the forms $\omega^{(a)}$. Integrality of $F^{(i)}_{\text{bkgr}}$, $\omega^{(a)}$ implies integrality of $\alpha_{(ia)(jb)}$.

The first two terms in the auxiliary $(d+2)$-dimensional action (49) correspond to kinetic terms in the physical action on $M_{d+1}$. As we describe at the beginning of this section, the kinetic terms of the gauge potential do not capture the fluctuations of finite discrete Abelian gauge field, and therefore can be ignored in the truncation to the topological sector. We are left with

$$S^{\text{BF+sources}}_{d+2} = \int_{M_{d+2}} \sum_{i,j,a,b} \left[ \alpha_{(ia)(jb)} f^{(ia)} \wedge f^{(jb)} - \kappa_{(ia)(jb)} f^{(ia)} \wedge j^{(jb)} \right]. \tag{50}$$

This action furnishes a $(d+2)$-dimensional description of a BF theory in $d+1$ dimensions.

**Example.**   For illustration purposes, let us consider the simple case in which we only have two relevant external fluxes, denoted $f^{(1)}, f^{(2)}$, and one $\alpha$ constant,

$$S^{\text{BF+sources}}_{d+2} = \int_{M_{d+2}} \left[ \alpha f^{(1)} \wedge f^{(2)} - f^{(1)} \wedge j^{(1)} - f^{(2)} \wedge j^{(2)} \right], \tag{51}$$

where we have performed a linear redefinition of the external currents to reabsorb the $\kappa$ constants. An action of this form appears for example in many holographic setups like $\text{AdS}_5 \times S^5$ or $\text{AdS}_7 \times S^4$ in IIB or M-theory respectively, where the we have background fluxes such that $\alpha = \int_{S^5} F_5 = N$ and $\alpha = \int_{S^4} G_4 = N$. The Bianchi identities read

$$\alpha f^{(2)} = j^{(1)}, \qquad (-1)^{\deg f^{(1)} \deg f^{(2)}} \alpha f^{(1)} = j^{(2)}. \tag{52}$$

Plugging this back into the action and evaluating the exponential of the action with brane sources we get

$$
\begin{aligned}
\langle e^{2\pi i \int_{\mathcal{M}^{(1)}} f^{(1)}} e^{2\pi i \int_{\mathcal{M}^{(2)}} f^{(2)}} \rangle &= \exp\left( \frac{2\pi i}{\alpha} \int_{M_{d+1}} j^{(1)}(\Sigma^{(1)}) \wedge d^{-1} j^{(2)}(\Sigma^{(2)}) \right) \\
&= \exp\left( \frac{2\pi i}{\alpha} \mathcal{L}_{d+1}(\Sigma^{(1)}, \Sigma^{(2)}) \right),
\end{aligned}
\tag{53}
$$

where $\Sigma^{(1)} = \partial \mathcal{M}^{(1)}$, $\Sigma^{(2)} = \partial \mathcal{M}^{(2)}$ are (homologically trivial) cycles in $M_{d+1}$ and we have used (27) with $D$ replaced by $d$.

**Flux Non-Commutativity.** We can now relate the above to flux non-commutativity. By inserting the above operators on alternative $\widetilde{\Sigma}^{(1)} = \partial \widetilde{\mathcal{M}}^{(1)}, \widetilde{\Sigma}^{(2)} = \partial \widetilde{\mathcal{M}}^{(2)}$:

$$
\begin{aligned}
\left\langle e^{2\pi i \int_{\widetilde{\mathcal{M}}^{(1)}} f^{(2)}} e^{2\pi i \int_{\widetilde{\mathcal{M}}^{(2)}} f^{(1)}} \right\rangle &= \exp\left( \frac{2\pi i}{\alpha} \int_{M_{d+1}} j^{(2)}(\widetilde{\Sigma}^{(1)}) \wedge d^{-1} j^{(1)}(\widetilde{\Sigma}_2) \right) \\
&= \exp\left( \frac{2\pi i}{\alpha} \mathcal{L}_{d+1}(\widetilde{\Sigma}^{(2)}, \widetilde{\Sigma}^{(1)}) \right).
\end{aligned}
\tag{54}
$$

This implies that

$$
\left\langle e^{2\pi i \oint_{\Sigma_1} f^{(1)}} e^{\oint_{\Sigma_2} f^{(2)}} \right\rangle = \left\langle e^{\oint_{\widetilde{\Sigma}_1} f^{(2)}} e^{\oint_{\widetilde{\Sigma}_2} f^{(1)}} \right\rangle e^{2\pi i \frac{\mathcal{L} - \widetilde{\mathcal{L}}}{\alpha}},
\tag{55}
$$

where $\mathcal{L}, \widetilde{\mathcal{L}}$ are short-hand for the linking numbers $\mathcal{L}_{d+1}$ for $\Sigma^{(i)}$ and $\widetilde{\Sigma}^{(i)}$ respectively.

If the two branes *unlink* in the second configuration, i.e. $\widetilde{\mathcal{L}} = 0$, we exactly get flux non-commutativity as a consequence of the two branes linking in $d + 1$ dimensions.

**Example: 4d $\mathcal{N} = 1$ SYM with $\mathfrak{g} = \mathfrak{su}(M)$.** To exemplify this, consider the holographic realization of pure super-Yang-Mills (SYM). We first consider the Klebanov-Strassler solution [119] dual to 4d $\mathfrak{g} = \mathfrak{su}(M) \, \mathcal{N} = 1$ SYM. This is the back-reacted configuration of $N$ D3-branes probing the resolved conifold, i.e. the cone of the $T^{1,1}$ Sasaki-Einstein space, $C(T^{1,1})$ and $M$ D5-branes wrapping $S^2 \subset T^{1,1}$. The relevant flux quantization is

$$
\int_{S^3} F_3 = M.
\tag{56}
$$

The ansatz for the higher-dimensional fluxes is

$$
\begin{aligned}
F^{(1)} = F_3 &= M \mathrm{vol}_{S^3} + f^{(1,1)} \wedge \mathrm{vol}_{S^2} + f^{(1,2)}, \\
F^{(2)} = H_3 &= f^{(2)}, \\
F^{(3)} = F_5 &= F_{\mathrm{bkrg}}^{(3)} + f^{(3,1)} \wedge \mathrm{vol}_{S^2} + f^{(3,2)} \wedge \mathrm{vol}_{S^3}.
\end{aligned}
\tag{57}
$$

From the IIB topological action, we derive the term

$$
S_{d+2} = \int_{M_{d+2}} M f^{(2)} \wedge f^{(3,1)}.
\tag{58}
$$

where $d = 4$. We must also include sources for the external fluxes

$$
\begin{aligned}
dH_7 = J^{(2)}: \quad J^{(2)} &= j^{(2,1)} \wedge \mathrm{vol}_{T^{1,1}} \dots, \\
dF_5 = J^{(3)}: \quad J^{(3)} &= j^{(3,1)} \wedge \mathrm{vol}_{S^3} + \dots,
\end{aligned}
\tag{59}
$$

where we recall that the labels on top of the $F^{(i)}$, $f$ and $j$ do not reflect the form degrees, which can instead be read off from the identification with the IIB fluxes $F_3, H_3, F_5$ in (57) and from their derivatives in (59). We then obtain

$$S_{d+2}^{\text{BF+sources}} = \int_{M_{d+2}} M f^{(2)} \wedge f^{(3,1)} - j^{(2,1)} \wedge f^{(2)} - j^{(3,1)} \wedge f^{(3,1)}. \tag{60}$$

This is the IR BF term which describes flux non-commutativity [120].

**BF-terms from $\kappa_{ij} F^{(i)} \wedge dF^{(j)}$.**   The second case is when the BF-terms or quadratic terms in the topological action, after compactification on $L_{\text{int}}$, are generated by one of the $F^{(i)} \wedge dF^{(j)}$ terms in (41). For ease of exposition, instead of considering the general action (41) we can consider a simpler action in $D + 2$ dimensions, with only two fluxes and no Chern-Simons interactions,

$$S_{D+2} = \int_{M_{D+2}} \left[ F^{(1)} \wedge dF^{(2)} + (\text{sources}) \right]. \tag{61}$$

For simplicity we also assume that there is only one pair of relevant cycles onto which $F^{(1)}$, $F^{(2)}$ are expanded, so that the relevant terms in the reduction ansatz are

$$F^{(1)} = f \wedge \phi, \qquad F^{(2)} = \tilde{f} \wedge \tilde{\phi}. \tag{62}$$

In the previous expressions the internal forms $\phi$, $\tilde{\phi}$ are not closed. Rather, they represent torsional cohomology elements in $H^\bullet(L_{\text{int}}, \mathbb{Z})$ [121, 122], as explained in greater detail below. The degrees of the forms $f$, $\tilde{f}$, $\phi$, $\tilde{\phi}$ must satisfy

$$\deg f + \deg \tilde{f} = d + 2, \qquad \deg \phi + \deg \tilde{\phi} = n - 1. \tag{63}$$

The reduction of (61) yields a $(d + 2)$-dimensional action of the form

$$S_{d+2}^{\text{BF+sources}} = \int_{M_{d+2}} \left[ \alpha f \wedge \tilde{f} - f \wedge j - \tilde{f} \wedge \tilde{j} \right]. \tag{64}$$

Here the coefficient $\alpha$ is given by

$$\alpha = (-1)^{(1+\deg \tilde{f})(1+\deg \phi)} \int_{L_n} d\phi \wedge \tilde{\phi}. \tag{65}$$

The external source terms $j$, $\tilde{j}$ originate from the source terms in (61). From $dF^{(2)}$ we also get a term in which the derivative acts on $\tilde{f}$. This term, however, yields $f \wedge d\tilde{f}$ in $d + 2$ dimensions. Such terms will lead to kinetic terms in the $(d + 1)$-dimensional action that we ignore once we consider the theory of flat gauge potentials only, as it was for the first case.

The integral in the $\alpha$ coefficient can be interpreted as linking in the internal space: this is illustrated below. We then see how the BF-terms as well as flux non-commutativity are directly equivalent to the brane and its magnetic dual brane linking also in external space as it was for the first case. Notice here the branes link doubly, i.e. both internally and externally. The internal linking of the branes leads to the coefficient of the BF-term. On top of this the external linking leads to flux non-commutativity.

As anticipated above, the non-closed forms $\phi$, $\tilde{\phi}$ encode torsional cohomology classes. More precisely, let us consider the pairs $(\phi, \Phi)$, $(\tilde{\phi}, \tilde{\Phi})$ with [121, 122]

$$\ell \Phi = d\phi, \qquad \tilde{\ell} \tilde{\Phi} = d\tilde{\phi}, \tag{66}$$

where $\ell$, $\tilde{\ell}$ are positive integers. The pair $(\phi, \Phi)$ models an element of $H^{\deg \phi + 1}(L_n, \mathbb{Z})$ of torsional degree $\ell$, while the pair $(\tilde{\phi}, \tilde{\Phi})$ models an element of $H^{\deg \tilde{\phi} + 1}(L_n, \mathbb{Z})$ of torsional degree $\tilde{\ell}$. The relation $\ell \Phi = d\phi$ corresponds to a statement of the form $\ell \Sigma = \partial \mathcal{M}$, where $\Sigma$ is the cycle dual to the closed form $\Phi$, hence of dimension $n - \deg \phi - 1$, and $\mathcal{M}$ is a chain of dimension $n - \deg \phi$. Similarly, $\tilde{\ell} \tilde{\Phi} = d\tilde{\phi}$ translates to $\tilde{\ell} \tilde{\Sigma} = \partial \tilde{\mathcal{M}}$, where $\tilde{\Sigma}$ is a cycle of dimension $n - \deg \tilde{\phi} - 1$. The torsional pairing of the cycles $\Sigma$, $\tilde{\Sigma}$ can be computed by taking the intersection number between $\Sigma$ and $\tilde{\mathcal{M}}$ and dividing by $\tilde{\ell}$, the torsional order of $\tilde{\Sigma}$,

$$\mathcal{T}_{L_n}(\Sigma, \tilde{\Sigma}) = \frac{1}{\tilde{\ell}} \Sigma \cdot_{L_n} \tilde{\mathcal{M}} \mod \mathbb{Z}. \tag{67}$$

Using $\Sigma \cdot_{L_n} \tilde{\mathcal{M}} = \int_{L_n} \Phi \wedge \tilde{\phi}$ and (65), (66), we can write

$$(-1)^{(1 + \deg \tilde{f})(1 + \deg \phi)} \alpha = \int_{L_{\text{int}}} d\phi \wedge \tilde{\phi} = \ell \, \Sigma \cdot_{L_n} \tilde{\mathcal{M}} = \ell \tilde{\ell} \, \mathcal{T}_{L_n}(\Sigma, \tilde{\Sigma}). \tag{68}$$

We have thus established a general relation between the BF coefficient $\alpha$, the torsional pairing $\mathcal{T}_{L_{\text{int}}}(\Sigma^{(1)}, \tilde{\Sigma}^{(1)})$, and the torsional orders $\ell$, $\tilde{\ell}$.[7] We also confirm the integrality of the BF coefficient $\alpha$.

To make this approach more concrete, consider the simple example of $S^3/\mathbb{Z}_k$. We write the metric in Hopf coordinates as follows:

$$g = \frac{1}{k^2}(d\psi + A)^2 + \frac{1}{4}(d\beta^2 + \sin^2 \beta \, d\theta^2), \tag{69}$$

where both $\psi$ and $\theta$ have period $2\pi$ and $A = \frac{k}{2} \cos \beta \, d\theta$. In the above notation, we construct two forms

$$\phi = \frac{1}{2\pi}(d\psi + A), \qquad \Phi = -\frac{1}{4\pi} \sin \beta \, d\beta \, d\theta, \tag{70}$$

such that

$$d\phi = k\Phi. \tag{71}$$

We can then compute integrals directly, such as

$$\int_{S^3/\mathbb{Z}_k} \phi \wedge d\phi = k. \tag{72}$$

**BF-terms from both Mechanisms.** Finally, we can consider cases where both situations show up, namely where we have non-zero background fluxes as well as non-trivial torsional pairings. Examples are furnished by $\text{AdS}_5 \times \mathbb{RP}^5$ in type IIB [88] or $\text{AdS}_7 \times \mathbb{RP}^4$ in M-theory [123] (the fluxes are $F_5$ and $G_4$, respectively). In the standard setting (case 2 above), torsional flux non-commutativity applies to branes that are electro-magnetic duals in the original $D + 1$ dimensional theory. In the hybrid case, the non-zero background flux induces torsional flux non-commutativity for branes that are not duals in $D + 1$ dimensions. As a result, some technical aspects of the computation of BF couplings in this class of scenarios require a more refined analysis; we refer the reader to the references above.

---

[7]In various examples throughout this work we will employ these pairs when dealing with torsional cohomology classes. However, when it comes to explicitly computing their associated integrals, we will swap frames to differential cohomology or torsional pairings.

### 3.4 Topological couplings in the SymTFT from branes

So far we have been focusing on how to get the BF-terms of the SymTFT from the branes in the holographic or geometric constructions. Dimensional reduction of the 10/11-dimensional flux sector with brane sources can lead to additional topological couplings, which, upon choosing suitable gapped boundary conditions, provides an invertible topological theory. This corresponds to the anomalies involving finite, abelian symmetries of an absolute QFT at the boundary. We refer to such couplings in the SymTFT as anomaly couplings (with the understanding that these couplings result in 't Hooft anomalies after certain choices of boundary conditions).

The strategy to obtain these extra topological (non BF-type) couplings is similar to the one implemented for identifying the BF-terms as linking of branes. It consists of dimensionally reducing the action (41) on $L_n$ and then applying the (dimensionally reduced) Bianchi identities, with sources, (43).[8] By substituting the fluxes in terms of brane sources, we can directly connect the extra topological couplings to brane linking. We now describe how this works in general, where we limit though to quadratic or cubic couplings, which are the cases of interest for us. On the other hand, extending to higher topological coupling is straightforward.

The extra topological (non BF-type) couplings of interest are couplings in the SymTFT in $d + 1$ dimensions. As in the previous section, however, we find it convenient to describe these couplings using an action in $d + 2$ dimensions, since this is what we naturally get from (41). We use $j^{(i)}$ to denote external currents. Their form degrees are left unspecified. In each case, it is assumed that they are such that the integrals we write can be non-zero.

#### 3.4.1 Quadratic Couplings

For the quadratic couplings there are two types of extra topological couplings, depending on their expression in terms of the brane sources.

**Quadratic Couplings 1.** The first case is

$$S_{\text{extra}} = a \int_{M_{d+2}} j^{(1)} \wedge d^{-1} j^{(2)} . \tag{73}$$

These sorts of terms in $d + 2$ dimensions are expected to combine into a total derivative, which we can rewrite as an integral in $M_{d+1}$, of the form

$$S_{\text{extra}} = a \int_{M_{d+1}} d^{-1} j^{(1)} \wedge d^{-1} j^{(2)} . \tag{74}$$

In this case the branes can link in $M_d$, where the QFT lives.

**Example.** A Pontryagin square $\mathfrak{P}(b_2)$ coupling in the 4d SymTFT for a QFT in 3d is an example of such a coupling. The finite 2-form gauge field $b_2$ is BF-dual to $\hat{b}_1$ in 3d. The topological operators realized by the dimensionally reduced branes, are identified with the holonomies of $\hat{b}_1$. They are lines that can link in the 3d spacetime, where the QFT lives. For example, this coupling can be found in setups where a stack of M5-branes is wrapped on a compact 3-manifold $\Sigma_3$ with an appropriate topological twist to preserve 3d $\mathcal{N} = 2$ or $\mathcal{N} = 1$ supersymmetry. In this case, the link geometry $L_7$ is an $S^4$ fibration over $\Sigma_3$. Depending on the geometry of $\Sigma_3$, $L_7$ can admit non-trivial torsional cohomology classes of degree 2. Expansion of $G_4$ onto such classes yields both discrete gauge fields 2-forms as $b_2$, and $\mathfrak{P}(b_2)$ couplings

---

[8]Alternatively one can reduce the Bianchi identities with brane sources directly and construct the lower-dimensional action from this. The two procedures are equivalent.

in the SymTFT, by applying the techniques of [39], see [124] where this will be utilized. Couplings of the form $\mathfrak{P}(b_2)$ are also found in the SymTFTs of supersymmetric 3d QFTs realized in M-theory using geometric engineering or M2-branes [42].

**Quadratic couplings 2.** The second case is

$$S_{\text{extra}} = a \int_{M_{d+2}} j^{(1)} \wedge j^{(2)} = (-)^{\deg j^{(1)}} a \int_{M_{d_{\text{ext}}}} j^{(1)} \wedge d^{-1} j^{(2)}. \tag{75}$$

This is instead a case where the two branes link in $M_{d+1}$, where the SymTFT lives.

**Example.** A coupling

$$b_2 \cup \text{Bock}(b_2'), \tag{76}$$

in a 5d SymTFT for a 4d QFT is of this type. Here $b_2$, $b_2'$ are $\mathbb{Z}_N$ 2-form fields, and Bock is the Bockstein homomorphism associated to the short exact sequence

$$0 \to \mathbb{Z}_N \xrightarrow{\times N} \mathbb{Z}_{N^2} \to \mathbb{Z}_N \to 0. \tag{77}$$

The dimensionally reduced branes are identified with the holonomies of $\hat{b}_2$, $\hat{b}_2'$, the 5d BF-duals of $b_2$, $b_2'$. Thus the brane sources are 2d surfaces, and indeed can link in the 5d spacetime where the SymTFT lives.

In both these quadratic couplings, $a$ is an integer constant coupling coefficient which is determined by an integral over $L_n$ of non-trivial fluxes components over the internal space. It depends on the representatives of non-trivial cohomology or non-trivial geometric linking of cycles in the internal space, wherever we face the first or the second situation described in the previous section, respectively. In addition $a$ is an integer because of flux quantization.

### 3.4.2 Cubic Couplings

For cubic couplings we face three distinct possibilities.

**Cubic Couplings 1.** The first case is

$$S_{\text{extra}} = a \int_{M_{d+2}} j^{(1)} \wedge d^{-1} j^{(2)} \wedge d^{-1} j^{(3)}. \tag{78}$$

Terms of this form correspond to an integral in $M_{d+1}$ of the form

$$S_{\text{extra}} = a \int_{M_{d+1}} d^{-1} j^{(1)} \wedge d^{-1} j^{(2)} \wedge d^{-1} j^{(3)}. \tag{79}$$

This is a triple linking configuration (cf. Milnor's triple intersection number [125]). We can formally recast it as a standard linking in $M_{d+1}$, namely a quantity of the form $\int_{M_{d+1}} j^{(12)} \wedge d^{-1} j^{(3)}$, with the identification

$$j^{(12)} = d^{-1} j^{(1)} \wedge d^{-1} j^{(2)}. \tag{80}$$

Recall that $j^{(i)}$ is supported on a cycle $\Sigma^{(i)}$ that is the boundary of a chain $\mathcal{M}^{(i)}$, which is usually referred to as Seifert (hyper)surface [115]. Then the RHS of (80) represents the intersection inside $M_{d+1}$ of the Seifert surfaces $\mathcal{M}^{(1)}$, $\mathcal{M}^{(2)}$ associated to $j^{(1)}$, $j^{(2)}$.

**Example: $B^3$ Anomaly in 5d.** A cubic coupling $b_2 b_2 b_2$ in the 6d SymTFT of a QFT in $d = 5$ dimensions, such as have appeared in [39, 126]. Here $b_2$ is a $\mathbb{Z}_N$ discrete 2-form. The dimensionally reduced branes are again identified with the holonomies of the BF-dual field $\hat{b}_3$ in six dimensions, which arise from M5-branes wrapping torsional 3-cycles of $L_n$. They are therefore 3d surfaces, which in six dimensions can form a non-trivial triple linking configuration as in (79).

**Example: 4d $\mathcal{N} = 1$ SYM with $G = SU(M)$.** This theory has mixed 't Hooft anomaly

$$\mathcal{A} = -2\pi \frac{1}{M} \int A_1 \cup \frac{\mathfrak{P}(B_2)}{2}, \tag{81}$$

where $B_2$ is the background for a $\mathbb{Z}_M^{(1)}$ 1-form symmetry and $A_1$ is the background for a $\mathbb{Z}_{2M}^{(0)}$ 0-form symmetry. Using the Klebanov-Strassler solution, a detailed supergravity origin of this anomaly is given in [1, 120].

We continue with the field notation introduced around (60). The generator of the 0-form symmetry was identified with a D5-brane wrapped on $S^3 \subset T^{1,1}$ in [1]. We introduce a source for the external field corresponding to background for this symmetry via

$$dF_3 = J^{(1)}, \qquad J^{(1)} = j^{(1,1)} \wedge \text{vol}_{S^2} + \dots, \tag{82}$$

where we use the expansion in (57), and with (59) we can rewrite the anomaly as follows

$$\frac{1}{2M} \int_{M_{d+2}} j^{(1,1)} \wedge d^{-1} j^{(3,1)} \wedge d^{-1} j^{(3,1)}, \tag{83}$$

where $d = 4$ and the form degrees of $j^{(3,1)}$ and $j^{(1,1)}$ are 3 and 2 respectively, and they can be read off from (57), (59) and (82).

**Cubic Couplings 2.** The second case is

$$S_{\text{extra}} = a \int_{M_{d+2}} j^{(1)} \wedge j^{(2)} \wedge d^{-1} j^{(3)}, \tag{84}$$

with corresponding term in $M_{d+1}$ of the form

$$S_{\text{extra}} = a \int_{M_{d+1}} j^{(1)} \wedge d^{-1} j^{(2)} \wedge d^{-1} j^{(3)}. \tag{85}$$

Again this is interpreted as suitable triple linking configuration. We can formally recast it as a standard linking in $M_{d+1}$, namely a quantity of the form $\int_{M_{d+1}} j^{(12)} \wedge d^{-1} j^{(3)}$, with the identification

$$j^{(12)} = j^{(1)} \wedge d^{-1} j^{(2)}. \tag{86}$$

The RHS represent the intersection inside $M_{d+1}$ of the cycle associated to $j^{(1)}$ with the Seifert surface associated to $j^{(2)}$.

**Example.** The coupling $b_2 b_2 \text{Bock}(a_1)$ in the 6d SymTFT of a QFT in $d = 5$ dimensions. Here $b_2$ is a $\mathbb{Z}_N$ 2-form field, $a_1$ a $\mathbb{Z}_M$ 1-form field, and the Bockstein homomorphism is analogous to the one introduced above in the $b_2 \text{Bock}(b_2')$ example.[9] The dimensionally reduced branes

---

[9]This example may be realized using 5d gauge theories. More precisely, we may start with a 5d gauge theory in which the $U(1)$ instanton 0-form symmetry and the center 1-form symmetry have a mixed anomaly, encoded in a coupling $b_2 b_2 f_2$ in the 6d SymTFT, where $f_2$ is the field strength of a continuous 1-form gauge field. If we restrict to a $\mathbb{Z}_M$ subgroup, this coupling becomes $b_2 b_2 \text{Bock}(a_1)$.

provide the holonomies of the BF-dual fields in six dimensions, $\hat{b}_3$ and $\hat{a}_4$, and are therefore 3d and 4d surfaces, respectively. Such 3d-3d-4d system in 6d can exhibit the triple linking described in (85).

**Cubic Couplings 3.** The third case is

$$S_{\text{extra}} = a \int_{M_{d+2}} j^{(1)} \wedge j^{(2)} \wedge j^{(3)}, \tag{87}$$

corresponding to the following in $M_{d+1}$,

$$S_{\text{extra}} = a \int_{M_{d+1}} j^{(1)} \wedge j^{(2)} \wedge d^{-1} j^{(3)}. \tag{88}$$

This is again a suitable triple linking configuration. We can formally recast it as a standard linking in $M_{d+1}$, namely a quantity of the form $\int_{M_{d+1}} j^{(12)} \wedge d^{-1} j^{(3)}$, with the identification

$$j^{(12)} = j^{(1)} \wedge j^{(2)}. \tag{89}$$

The RHS represent the intersection inside $M_{d+1}$ of the cycles associated to $j^{(1)}, j^{(2)}$.

**Example.** An example is the coupling $a_1 \text{Bock}(a_1) \text{Bock}(a_1)$ in the 5d SymTFT of a 4d QFT. Here $a_1$ is a $\mathbb{Z}_N$ 1-form gauge field, and Bock is the same Bockstein homomorphism as in the previous examples of this section. The dimensionally reduced branes give the holonomies of $\hat{a}_3$, the BF-dual of $a_1$ in 5d, and are therefore 3d surfaces. Three such branes can link as in (88).

**Example: $G^{(0)}$ Anomaly of 4d $\mathcal{N} = 1$ SYM with $\mathcal{G} = SU(M)$.** There is a pure 0-form symmetry anomaly [1]

$$S_{\text{top}} \supset \int -\frac{\kappa^2 M^2}{4} \mathcal{F}_2^3, \tag{90}$$

where $\mathcal{F}_2 = dA_1$ is the field strength of a background gauge field for the 0-form symmetry and $N = \kappa M$ is the number of D3-branes. In terms of brane sources, this is a pure 0-form symmetry anomaly of the third type

$$\frac{\kappa^2}{32M} j^{(1,1)} \wedge j^{(1,1)} \wedge j^{(1,1)}, \tag{91}$$

where using (57) and with $df^{(1,1)} = 2M\mathcal{F}_2$ (see Appendix A of [1]), we find

$$\mathcal{F}_2 = \frac{j^{(1,1)}}{2M}. \tag{92}$$

It would be interesting to compare this with a direct field theory analysis, in the large rank limit.

For the cubic coupling the coefficient, $a$, is integer and constant because of flux quantization and depends on the internal sources and how they integrate on $L_n$ non-trivially. In particular, $a$ can originate from either situation listed in the previous section depending on how the fluxes and their Bianchi identities get compactified. We notice that the three cases 1, 2, 3 listed above correspond to triple linkings of type 0, 1, 2, respectively, in the notation of [43], see also [115].

## 3.5 Anomaly couplings as charges of defect junctions from branes

These extra topological couplings in the SymTFT can lead to anomalies of an absolute QFT once suitable boundary conditions are chosen. This is also true for the BF-couplings, there are some choice of boundary condition for which some left over BF-couplings can lead to a quadratic anomaly. The choice of boundary condition depends on the specific theory itself. In terms of branes these corresponds to picking a radial direction of the SymTFT and understanding how the branes providing topological and charged defects can extend in $M_{d+1}$. For instance charged defects come from branes extended in the radial direction (the direction perpendicular to the boundary where the relative QFT lives), i.e. the field electrically charging the branes has Dirichlet boundary condition.

Whereas topological operators comes from branes parallel to the boundary, i.e. the field which electrically charge the brane is freely varying. From this point of view it is easy to interpret the quadratic anomaly as the charge of a brane, corresponding to a topological defect, with respect to the same or a different brane, corresponding to the same or a different topological defect. These correspond to a pure or a mixed 't Hooft anomaly, respectively. A cubic anomaly can be interpreted as charges of a brane intersection, corresponding to a junction of topological defects (equal or different, for pure or mixed 't Hooft anomalies respectively), with respect to the same or another brane, corresponding to the same or another topological defect (for pure or mixed 't Hooft anomalies respectively). The charges computed here correspond to the number $a$ which is an integral over the internal manifold times the linking of the branes in the external space as specified for the different cases of quadratic and cubic extra topological couplings above. The anomalies can be interpreted as an ambiguity of the topological defects whenever they link or unlink in the radial direction.

**Example 1.** We now illustrate the above ideas for two concrete classes of mixed anomalies for discrete $p$-form symmetries. Firstly, let us consider an anomaly action of the schematic form

$$\mathcal{A} = \alpha \int_{M_{d+1}} A_{p_1}^{(1)} \dots A_{p_n}^{(n)}, \qquad \sum_{j=1}^{n} p_j = d + 1. \tag{93}$$

Each $A_{p_j}^{(j)}$ is a discrete background field for a global symmetry, a cohomology class of degree $p_j$. The constant $\alpha$ is the anomaly coefficient. We denote the topological defect implementing the global symmetry associated to $A_{p_j}^{(j)}$ as $D_{d-p_j}^{(j)}$. (Notice that these topological defects live in $d$ dimensions.) Let us consider topological defects $D_{p_2}^{(2)}, \dots, D_{p_n}^{(n)}$ in generic positions in $d$-dimensional spacetime. They intersect along a locus of dimension $p_1 - 1$. The mixed anomaly (93) means that this intersection has non-zero charge (proportional to $\alpha$) under the topological operator $D_{d-p_1}^{(1)}$. For ease of discussion, we have singled out the symmetry associated to $A_{p_1}^{(1)}$, but clearly analogous statements can be made by singling out any other $A_{p_j}^{(j)}$.

**Example 2.** Next, let us consider a mixed anomaly for two finite global symmetries, of the form

$$\mathcal{A} = \alpha \int_{W_{d+1}} A_{d-p} \mathrm{Bock}(B_p). \tag{94}$$

We denote the topological defects operator generating the global symmetries associated to the background fields $A_{d-p}$, $B_p$ as $D_p^{(A)}$, $D_{d-p}^{(B)}$, respectively. For simplicity, we assume that $B_p$ is associated to a $\mathbb{Z}_k$ $(p-1)$-form symmetry. Then Bock denotes the Bockstein homomorphism associated to the short exact sequence $0 \to \mathbb{Z} \xrightarrow{k} \mathbb{Z} \to \mathbb{Z}_k \to 0$. This anomaly can only be detected on spacetimes with torsion, because $\mathrm{Bock}(B_p)$ lies in the torsion subgroup of $H^{p+1}(W_{d+1}, \mathbb{Z})$.

An interpretation in terms of junctions of topological defects can be given along the lines of appendix F of [127] (and many subsequent works). The relevant torsion in $d$ dimensions is in $H^{p+1}(W_d, \mathbb{Z})$, or $H_{d-p-1}(W_d, \mathbb{Z})$ by Poincaré duality. We thus consider a torsional $(d-p-1)$-dimensional cycle $M_{d-p-1}$ in $d$ dimensions, satisfying $r M_{d-p-1} = \partial N_{d-p}$. We may insert a topological defect $D_{d-p}^{(B)}$ supported on $N_{d-p}$, with $M_{d-p-1}$ regarded as a codimension one junction inside $D_{d-p}^{(B)}$. The anomaly (94) means that this junction on $M_{d-p-1}$ has non-zero charge (proportional to $\alpha$) under the topological defects $D_p^{(A)}$ (indeed, they link in $d$ dimensions). In the action (94) of the anomaly theory, the Bockstein map can be "integrated by parts" and the roles of $A$ and $B$ in the previous discussion can be exchanged. This sort of anomaly can be found, for example, in 4d gauge theory with gauge algebra $\mathfrak{su}(N)$, with $N = \ell \ell'$ for integers $\ell > 1$, $\ell' > 1$. We can specify a global form of the theory with both non-trivial electric and magnetic 1-form symmetries. The mixed anomaly between the latter is of the form (94) with $d = 4$, $p = 2$.

## 3.6 Condensation defects from branes

From a symmetry categorical point of view the condensation completion (or Karoubi completion) corresponds to adding all possible condensation defects. We have seen how this is realized in terms of the SymTFT by including the couplings to lower-dimensional DW-theories in (7).

We now turn to the string theory interpretation. For definiteness we work in type II, but similar remarks apply to M-theory.[10] Let us consider a D$p$-brane on $M_{p+1}$. We are interested in writing down a topological action, formulated on an auxiliary manifold $M_{p+2}$ in one dimension higher, that captures the topological couplings on the D$p$-brane. Moreover, we also want to capture the kinetic term $f_2 \wedge *f_2$ from the DBI action using the auxiliary topological action. We propose the following,

$$S_{p+2}^{\mathrm{D}p} = \int_{M_{p+2}} \left[ \hat{f}_{p-1} \wedge df_2 + \left( e^{f_2} \sum_q F_q \sqrt{\frac{\hat{A}(T)}{\hat{A}(N)}} \right)_{p+2} \right]. \tag{95}$$

Here $f_2$ is the field strength of the gauge field on the D$p$-brane and $\hat{f}_{p-1}$ is its Hodge dual in $M_{p+1}$. The quantities $F_q$ are the RR fluxes, pulled back from the bulk, and we have also included the standard A-roof terms from the Wess-Zumino couplings, for the tangent and normal bundles, respectively. If we consider an anti D$p$-brane, we flip the sign of action (95).[11]

We want to argue that considering a combined D$p/\overline{\mathrm{D}p}$ system provides a possible stringy origin for the condensation-completed SymTFT action (7). We proceed by considering a couple of illustrative examples.

**Example: 4d $\mathcal{N} = 1$ Holographic dual.** We continue with the Klebanov-Strassler example 3.3. For this we need to consider the action of D5-branes. Their action is given by

$$S_7^{\mathrm{D}5} = \int_{M_7} \left[ \hat{f}_4 \wedge df_2 + F_7 + f_2 \wedge F_5 + \left( \frac{1}{2} f_2^2 + \frac{p_1(N) - p_1(T)}{48} \right) \wedge F_3 + \left( \frac{1}{3!} f_2^3 + f_2 \frac{p_1(N) - p_1(T)}{48} \right) \wedge F_1 \right]. \tag{96}$$

---

[10]Here we stress that we are not analyzing the full dynamical brane/ anti-brane annihilation process. Specifically, we are considering only the topological couplings on the brane/ anti-brane pair.

[11]The sign of the DBI term for a brane and an antibrane is the same. The flip in sign in the BF term reformulation of the DBI kinetic term is compensated by a flip in sign in the Hodge duality relation between $f_2$ and $\hat{f}_{p-1}$.

As a result, the combined action for a D5-brane/anti-D5-brane reads

$$S_7^{\text{D5/}\overline{\text{D5}}} = \int_{M_7} \left[ \hat{f}_4 \wedge df_2 - \hat{f}_4' \wedge df_2' + (f_2 - f_2') \wedge F_5 + \frac{1}{2}(f_2 - f_2') \wedge (f_2 + f_2') \wedge F_3 \right.$$
$$\left. + (f_2 - f_2') \wedge \left( \frac{f_2^2 + f_2 \wedge f_2' + f_2'^2}{6} + \frac{p_1(N) - p_1(T)}{48} \right) \wedge F_1 \right]. \tag{97}$$

We have used a prime to denote the gauge field $f_2'$ on the anti-D5-brane and its partner $\hat{f}_4'$.

In the Klebanov-Strassler holographic setup, the D5/$\overline{\text{D5}}$ system is wrapped on $M_7 = M_4 \times S^3$ with $M$ units of $F_3$ through the $S^3$. Our task is to integrate the 7d topological action on $S^3$. The discussion parallels exactly the two cases discussed in section 3.3. The terms $\hat{f}_4 \wedge df_2$ would correspond to kinetic terms in the lower-dimensional theory on $M_4$, which we neglect because we are studying the topological sector. Next, the terms quadratic in $f_2, f_2'$ yield a 4d description of a set of abelian CS-terms in 3d. Moreover, $F_5$ admits a non-trivial component along $S^3$: from $(f_2 - f_2') \wedge F_5$ on $S^3$ we get a coupling of $(f_2 - f_2')$ to a 2-form bulk field, denoted $g_2$. In summary, the relevant terms are

$$S^{\text{D5/}\overline{\text{D5}}} = \int_{M_4} \left[ \frac{M}{2}(f_2 - f_2') \wedge (f_2 + f_2') + (f_2 - f_2') \wedge g_2 \right]. \tag{98}$$

We suggest the following interpretation, making contact with the general expression (7) for the condensation-completed SymTFT. The combination $f_2 - f_2'$ is identified with the localized field $a_1$ in the lower-dimensional DW type theory in the SymTFT that accounts for a class of condensation defects. The combination $f_2 + f_2'$ corresponds instead to $\hat{a}_1$, the BF-dual to $a_1$ in the lower-dimensional DW type theory, Finally, $g_2$ corresponds to one of the bulk fields $b_{p+1}$ (here $p = 1$). This can be seen more explicitly in the case of $M$ odd. We perform a field redefinition implemented by an integral, unimodular matrix, and we rename $g_2$,

$$\begin{pmatrix} f_2 \\ f_2' \end{pmatrix} = \begin{pmatrix} 1 & 1 \\ 0 & 1 \end{pmatrix} \begin{pmatrix} da_1 \\ d\hat{a}_1 \end{pmatrix}, \qquad g_2 = b_2. \tag{99}$$

We get the action

$$\int_{M_3} \left[ M\hat{a}_1 \wedge da_1 + a_1 \wedge b_2 + \frac{K}{2} a_1 \wedge da_1 \right], \qquad K = M. \tag{100}$$

The Lagrangian in bracket describes the $(\mathcal{Z}_M)_K$ discrete gauge theory [128], coupled to the bulk field $b_2$. On a spin manifold, $K$ can be any integer and the periodicity is $K \sim K + 2M$ if $M$ is even and $K \sim K + M$ if $M$ is odd. We then see that, for $M$ odd, the Lagrangian (100) on a spin manifold is equivalent to

$$\int_{M_3} \left[ M\hat{a}_1 \wedge da_1 + a_1 \wedge b_2 \right], \tag{101}$$

which matches (7). For $M$ even, (100) still describes a condensation defect, but with non-trivial discrete torsion $K = M$, see e.g. appendix B of [61].

**Example: 4d $\mathcal{N} = 4$ $\mathfrak{so}(4n)$ SYM.** Let us now discuss an example that illustrates the importance of the terms $\hat{f}_{p-1}df_2$ in (95) in the presence of torsion. The action (95) for a D3-brane reads

$$S_5^{\text{D3}} = \int_{M_5} \left[ \hat{f}_2 \wedge df_2 + F_5 + f_2 \wedge F_3 + \left( \frac{1}{2}f_2^2 + \frac{p_1(N) - p_1(T)}{48} \right) \wedge F_1 \right], \tag{102}$$

and therefore a D3/$\overline{\text{D3}}$ system is described by

$$S_5^{\text{D3}/\overline{\text{D3}}} = \int_{M_5} \left[ \hat{f}_2 \wedge df_2 - \hat{f}_2' \wedge df_2' + (f_2 - f_2') \wedge F_3 + \frac{1}{2}(f_2 - f_2') \wedge (f_2 + f_2') \wedge F_1 \right]. \tag{103}$$

We consider the holographic dual setup to 4d $\mathcal{N} = 4$ SYM with gauge algebra $\mathfrak{so}(4N)$. In this case $M_5 = M_4 \times \mathbb{RP}^1$, with $\mathbb{RP}^1$ regarded as an element of $H_1(\mathbb{RP}^5, \mathbb{Z}) \cong \mathbb{Z}_2$. From the point of view of $M_5$, the $\mathbb{RP}^1$ factor provides a torsional class of degree one $t_1$, of torsional order 2. Following the approach of [121, 122], we can model this by introducing a pair of differential forms on $\mathbb{RP}^1$,

$$2\Phi_1 = d\phi_0, \tag{104}$$

see (66). We expand $f_2$ and $\hat{f}_2$ onto $\phi_0$,

$$f_2 = \mathcal{F}_2 \phi_0 + \dots, \qquad \hat{f}_2 = \hat{\mathcal{F}}_2 \phi_0 + \dots, \tag{105}$$

and similarly for the primed fields, and we have

$$\int_{M_4 \times \mathbb{RP}^1} \left[ \hat{f}_2 \wedge df_2 - \hat{f}_2' \wedge df_2' \right] = \left[ \int_{\mathbb{RP}^1} d\phi_0 \wedge \phi_0 \right] \int_{M_4} \left[ \hat{\mathcal{F}}_2 \wedge \mathcal{F}_2 - \hat{\mathcal{F}}_2' \wedge \mathcal{F}_2' + \dots \right]. \tag{106}$$

As in section 3.3, the terms where the derivative acts on the $\mathcal{F}$'s can be neglected, because they describe kinetic terms after integrating over $\mathbb{RP}^1$. The integral of $d\phi_0 \wedge \phi_0$ encodes the torsional self-pairing of $t_1$,

$$\int_{\mathbb{RP}^1} d\phi_0 \wedge \phi_0 = 2. \tag{107}$$

Finally, we also expand the bulk field $F_3$ onto $(\Phi_1, \phi_0)$: the relevant term is $F_3 = g_2 \wedge \Phi_1$. As a result, the term $(f_2 - f_2') \wedge F_3$, after integration on $\mathbb{RP}^1$, yields a term $(\mathcal{F}_2 - \mathcal{F}_2') \wedge g_2$. In conclusion, we arrive at the following set of couplings,

$$S^{\text{D3}/\overline{\text{D3}}} = \int_{M_4} \left[ 2\hat{\mathcal{F}}_2 \wedge \mathcal{F}_2 - 2\hat{\mathcal{F}}_2' \wedge \mathcal{F}_2' + (\mathcal{F}_2 - \mathcal{F}_2') \wedge g_2 \right]. \tag{108}$$

Because of S-duality, however, we know that the presence of a coupling of $f_2$ to $C_2$ implies the presence of a $\hat{f}_2$ (which is the electromagnetic dual of $f_2$) to $B_2$. We then expect the full set of relevant couplings to be

$$S^{\text{D3}/\overline{\text{D3}}} = \int_{M_4} \left[ 2\hat{\mathcal{F}}_2 \wedge \mathcal{F}_2 - 2\hat{\mathcal{F}}_2' \wedge \mathcal{F}_2' + (\mathcal{F}_2 - \mathcal{F}_2') \wedge g_2 + (\hat{\mathcal{F}}_2 - \hat{\mathcal{F}}_2') \wedge h_2 \right]. \tag{109}$$

Here we have expanded $H_3$ onto $\Phi_1$ as $H_3 = h_2 \wedge \Phi_1$. The full action might be derived using the $SL(2, \mathbb{Z})$-covariant formulation of [129].

To make contact with (7) we perform a redefinition implemented a matrix in $GL(4, \mathbb{Z})$, and we rename $g_2$ and $h_2$,

$$\begin{pmatrix} \mathcal{F}_2 \\ \mathcal{F}_2' \\ \hat{\mathcal{F}}_2 \\ \hat{\mathcal{F}}_2' \end{pmatrix} = \begin{pmatrix} 1 & 0 & 1 & 1 \\ 0 & 0 & 1 & 1 \\ 1 & 1 & 0 & 0 \\ 1 & 1 & -1 & 0 \end{pmatrix} \begin{pmatrix} da_1 \\ d\hat{a}_1 \\ da_1' \\ d\hat{a}_1' \end{pmatrix}, \qquad g_2 = b_2, \qquad h_2 = \hat{b}_2. \tag{110}$$

We obtain the action

$$\int_{M_3} \left[ \left( 2\hat{a}_1 \wedge da_1 + a_1 \wedge b_2 + \frac{K}{2} a_1 \wedge da_1 \right) + \left( 2\hat{a}_1' \wedge da_1' + a_1' \wedge \hat{b}_2 + \frac{K'}{2} a_1' \wedge da_1' \right) \right], \qquad \begin{array}{l} K = 4, \\ K' = 4. \end{array} \tag{111}$$

We recognize two copies of a $(\mathcal{Z}_M)_K$ discrete gauge theory [128] with $M = 2$, $K = 4$. On a spin manifold with $M$ even, $K \sim K + 2M$ and hence the above action is equivalent to

$$\int_{M_3} \left[ \left( 2\hat{a}_1 \wedge da_1 + a_1 \wedge b_2 \right) + \left( 2\hat{a}'_1 \wedge da'_1 + a'_1 \widehat{b}_2 \right) \right], \tag{112}$$

matching with (7).

### 3.7 Non-genuine and twisted sector operators

Non-genuine or twisted sector operators arise from branes that couple to backgrounds which cannot necessarily end on the boundary, i.e. do not have Dirichlet boundary conditions.

Lets see how this is encoded in terms of the branes. If we were to end a brane that is electrically charged under a $(p + q + 1)$-form field $C_{p+q+1}$ on the boundary, then imposing Neumann boundary conditions on $\mathcal{B}^{\text{sym}}$ reads

$$\text{Neumann:} \quad \partial_{[r}C_{ij\ldots]}\big|_{\mathcal{B}^{\text{sym}}} \wedge \omega(\Sigma_q) = 0, \tag{113}$$

where $r$ is the direction transverse to the boundary (i.e. the radial direction), $i, j = 1\ldots p + 1$ denote the direction parallel to $\mathcal{B}^{\text{sym}}$, the brane can also wrap internal submanifold of $\Sigma_q \subset L(\mathbf{X})$, and $\omega(\Sigma_q)$ is transverse to the SymTFT. The internal manifold $\Sigma_q$ is important to determine properties of the $(p + 1)$-dimensional defect of the SymTFT, but it is a spectator with respect to the boundary conditions, hence we can put $\omega(\Sigma_q)$ aside for the moment.

Expanding out the Neumann boundary conditions along the direction of the SymTFT and restricting to the symmetry boundary we obtain

$$(\partial_r C_{ij\ldots} - \partial_i C_{rj\ldots})|_{\mathcal{B}^{\text{sym}}} = 0. \tag{114}$$

There are various configurations we can consider:

- Symmetry generators: for a brane without a radial component, this means we simply have

$$\partial_r C_{ij\ldots} = 0, \tag{115}$$

  which corresponds to the projection in figure 2 of the brane parallel to the boundary. This gives rise to $(p + 1)$-dimensional (topological) symmetry defects.

- Twisted Sector: if the second term in (114) is present, it electrically charges a $(p + q)$-brane ($(p + 1)$-dimensional operator when integrated on $\omega(\Sigma_q)$) extended along the radial direction ending at the boundary in a $p$-dimensional operator, which forms a junction with a $(p + q)$-brane ($(p + 1)$-dimensional operator when integrated on $\omega(\Sigma_q)$) extending along $\mathcal{B}^{\text{sym}}$. When we consider the first term as well, this correspond exactly to the L-shaped configuration, where the gauge transformation of the first term is cancelled by the gauge transformation of the second, in figure 4.

**Example: BF-couplings in AdS₅.** The simplest example to consider is the BF-theory for $\mathbb{Z}_N$ 2-form fields in 5d, which is the SymTFT for the 4d $SU(N)$ maximal SYM theory

$$S_{\text{SymTFT}} = N \int_{M_5} b_2 \wedge dc_2. \tag{116}$$

For example, imposing the boundary conditions

$$b_2 \text{ Dirichlet}, \qquad c_2 \text{ Neumann}, \tag{117}$$

the topological defects $\mathbf{Q}_2^{(b)}$, which are realized in terms of F1-strings, can end on the physical boundary and give rise to line operators in the gauge theory. On the other hand the D1-strings, which give rise to the bulk topological defects $\mathbf{Q}_2^{(c)}$, cannot end. There are two configurations:

- $\mathbf{Q}_2^{(c)}$ project parallel to the boundary as in figure 2 and give rise to the topological defects $D_2$ that generate the $\mathbb{Z}_N^{(1)}$ 1-form symmetry.

- $\mathbf{Q}_2^{(c)}$ project in an L-shape as in figure 4, and give rise (after interval compactification) to twisted sector 't Hooft lines, i.e. non-genuine, in this case, twisted sector, line operators that are attached to a topological surface.

This is of course well-known in the context of this standard holographic setup [37, 127] and was recently expanded upon in [130].

## 3.8 Example: 4d $\mathcal{N} = 4$ $\mathfrak{so}(4n)$ SYM

It is known that theories with an array of global structures based on the algebra $\mathfrak{so}(4n)$ contain non-invertible topological operators [58]. In this section we will use the holographic dual of these theories to study the SymTFT, in particular the BF terms.

**Holographic Dual.** The holographic solution relevant for these theories is IIB on $\text{AdS}_5 \times \mathbb{RP}^5$ [131]. The various global forms of the gauge group correspond to different choices of boundary conditions for various bulk gauge fields [130].

We refer the reader to [88] for more details on this setup. For convenience we collect the co/homology groups of the internal space $\mathbb{RP}^5$ with un/twisted coefficients below

$$
\begin{aligned}
H^\bullet(\mathbb{RP}^5, \mathbb{Z}) &= \{\mathbb{Z}, 0, \mathbb{Z}_2, 0, \mathbb{Z}_2, \mathbb{Z}\}, & H^\bullet(\mathbb{RP}^5, \widetilde{\mathbb{Z}}) &= \{0, \mathbb{Z}_2, 0, \mathbb{Z}_2, 0, \mathbb{Z}_2\}, \\
H_\bullet(\mathbb{RP}^5, \mathbb{Z}) &= \{\mathbb{Z}, \mathbb{Z}_2, 0, \mathbb{Z}_2, 0, \mathbb{Z}\}, & H_\bullet(\mathbb{RP}^5, \widetilde{\mathbb{Z}}) &= \{\mathbb{Z}_2, 0, \mathbb{Z}_2, 0, \mathbb{Z}_2, 0\}.
\end{aligned}
\tag{118}
$$

For $\mathfrak{so}(4n)$ the dual supergravity solution contains 5-form flux

$$
\int_{\mathbb{RP}^5} F_5 = 2n.
\tag{119}
$$

**BF Terms.** Before we begin, we introduce notation for the forms on which we will be expanding fluxes and sources

$$
\begin{aligned}
H_i(\mathbb{RP}^5, \widetilde{\mathbb{Z}}): & \quad (\widetilde{\phi}_i, \widetilde{\Phi}_i), \quad d\widetilde{\phi}_i = 2\widetilde{\Phi}_i, \quad i \in \{0, 2, 4\}, \\
H_i(\mathbb{RP}^5, \mathbb{Z}): & \quad (\phi_i, \Phi_i), \quad d\phi_i = 2\Phi_i, \quad i \in \{1, 3\}.
\end{aligned}
\tag{120}
$$

The BF terms come from more than one source in this case since we have both flux and torsion in the internal space.

First, we consider terms coming from the IIB Chern-Simons term. For this we require the fluxes:

$$
\begin{aligned}
F^{(1)} &= F_3 = f^{(1)} \wedge \widetilde{\phi}_4 + \dots, \\
F^{(2)} &= H_3 = f^{(2)} \wedge \widetilde{\phi}_4 + \dots.
\end{aligned}
\tag{121}
$$

Due to (119), we obtain a term

$$
S_{d+2}^{\text{BF}} \supset \int_{M_{d+2}} 2n f^{(1)} \wedge f^{(2)}.
\tag{122}
$$

Including sources

$$dF_7 = J^{(1)}: \quad J^{(1)} = j^{(1)} \wedge \widetilde{\Phi}_0 + \dots,$$
$$dH_7 = J^{(2)}: \quad J^{(2)} = j^{(2)} \wedge \widetilde{\Phi}_0 + \dots. \tag{123}$$

We recall that the form degrees of $F^{(i)}$, $f$ and $j$ are not specified by their labels on top, but they can be read off from (121) and (123), as well as from (125), (127) and (131) for what follows, once identified with the IIB fluxes $H_3, F_3, F_5, F_7, H_7$ and derivatives thereof. We then have

$$S_{d+2}^{\text{BF+sources}} \supset \int_{M_{d+2}} 2n f^{(1)} \wedge f^{(2)} - f^{(1)} \wedge j^{(1)} - f^{(2)} \wedge j^{(2)}, \tag{124}$$

where $d = 4$. Now we look to BF terms coming from $\kappa_{ij} F^{(i)} dF^{(j)}$ terms. There are three such terms. We reduce the IIB kinetic terms $H_3 \wedge dH_7$, $F_3 \wedge dF_7$ and $F_5 \wedge dF_5$ in turn.

Beginning with the first,

$$F^{(3)} = H_7 = \widetilde{f}^{(3,\widetilde{\phi}_0)} \wedge \widetilde{\phi}_0 + \dots. \tag{125}$$

The new BF term coefficient comes from the integral identity

$$\int_{\mathbb{RP}^5} d\widetilde{\phi}_0 \wedge \widetilde{\phi}_4 = 2. \tag{126}$$

Including the F1 string source for $H_7$

$$dH_3 = J^{(3)}: \quad J^{(3)} = j^{(3)} \wedge \widetilde{\Phi}_4. \tag{127}$$

Finally, we obtain the new contributions:

$$S_{d+2}^{\text{BF+sources}} \supset \int_{M_{d+2}} 2 f^{(2)} \wedge \widetilde{f}^{(3,\widetilde{\phi})} + \widetilde{f}^{(3,\widetilde{\phi})} \wedge j^{(3)}. \tag{128}$$

For $F_3$ there is also the Bianchi identity

$$dF_3 = J^{(4)}: \quad J^{(4)} = j^{(4)} \wedge \widetilde{\Phi}_4. \tag{129}$$

There is an identical contribution from the $F_3 \wedge F_7$ term which we denote with (4) superscripts

$$S_{d+2}^{\text{BF+sources}} \supset - \int_{M_d+2} 2 f^{(1)} \wedge \widetilde{f}^{(4,\widetilde{\phi})} + \widetilde{f}^{(4,\widetilde{\phi})} \wedge j^{(4)}. \tag{130}$$

Lastly we also consider the $dF_5 \wedge F_5^D$ term:

$$F^{(5)} = F_5 = f^{(5,1)} \wedge \phi_1 + f^{(5,3)} \wedge \phi_3,$$
$$dF_5 = J^{(5)}: \quad J^{(5)} = j^{(5,1)} \wedge \Phi_3 + j^{(5,3)} \wedge \Phi_1, \tag{131}$$

such that we obtain

$$S_{d+2}^{\text{BF+sources}} \supset \int_{M_{d+2}} 2 f^{(5,1)} \wedge f^{(5,3)} - f^{(5,3)} \wedge j^{(5,3)} - f^{(5,1)} \wedge j^{(5,1)}. \tag{132}$$

Putting all of these pieces together, we match the BF terms of [88, 130].

The $\mathfrak{so}(4n)$ theory also has an additional topological coupling, which depending on boundary conditions lead to a mixed anomaly,

$$\mathcal{A} = \frac{1}{2} \int_{M_{d+1}} A_1 C_2' B_2, \tag{133}$$

where $A_1$ is a background for $\mathbb{Z}_2^{(0)}$ and $C_2', B_2$ are both $\mathbb{Z}_2^{(1)}$ backgrounds. We can re-write this coupling in terms of sources using the identifications (e.g. using table 3)

$$f^{(2)} \leftrightarrow dB_2, \quad f^{(1)} \leftrightarrow dC_2', \quad f^{(5,1)} \leftrightarrow dA_1. \tag{134}$$

The anomaly term comes from the IIB cubic Chern-Simons coupling which by using the Bianchi identities (127), (129) and (131) can be re-written in terms of brane sources as

$$\mathcal{A} = \frac{1}{2} \int_{M_{d+1}} d^{-1} j^{(3)} \wedge d^{-1} j^{(4)} \wedge d^{-1} j^{(5,3)}, \tag{135}$$

where the coefficient is given by the following integration on $\mathbb{RP}^5$,[12]

$$\frac{1}{8} \int_{\mathbb{RP}^5} d\tilde{\phi}_4 \wedge d\tilde{\phi}_4 \wedge \phi_1 = \int_{\mathbb{RP}^5} \tilde{\Phi}_4 \wedge \tilde{\Phi}_4 \wedge d^{-1}\Phi_1 = \frac{1}{2}. \tag{136}$$

This is cubic coupling of type 1 (79) coming from three type of brane sources: NS5 on $\mathbb{RP}^4$, D5 on $\mathbb{RP}^4$ and D3 on $\mathbb{RP}^1$, which model the topological defects once properly compactified on the torsional cycles.

## 3.9 Example: Duality and triality defects for $\mathcal{N} = 2\,[A_2, D_4]$ theory

In this section we use our general setup to construct symmetry defects as branes in the isolated hypersurface singularity (IHS) (Calabi-Yau threefold) describing the 4d $\mathcal{N} = 2\,[A_2, D_4]$ SCFT in IIB string theory. This theory admits generalized symmetries [11,14,22,91,132]. In particular, we will propose a new construction of symmetry defects as lower-dimensional branes induced by world-volume flux for a higher-dimensional brane. The singularity $\mathbf{X}$ is described by the following hypersurface equation [11],

$$x_1^2 + x_2^3 + x_3^3 + x_4^3 = 0 \subset \mathbb{C}^4. \tag{137}$$

We construct now the symmetry defects wrapping topological cycles of the link geometry, $\partial\mathbf{X} = L(\mathbf{X})$. There is no flux in the background, but $L(\mathbf{X})$ has non-trivial torsional cycles [11]

$$H_2(L(\mathbf{X}), \mathbb{Z}) = \mathfrak{f} \oplus \mathfrak{f}' = \mathbb{Z}_2 \oplus \mathbb{Z}_2'. \tag{138}$$

In the last equality we specialize to $[A_2, D_4]$. Wrapping D3-branes on these torsion cycles results in the topological defects of the SymTFT.

There is a non-trivial linking of the generators obtained by wrapping D3s on the two $\mathbb{Z}_2$ factors

$$\text{Link}_{L(\mathbf{X})}(\gamma, \gamma') = \frac{1}{2}, \qquad \gamma \in \mathbb{Z}_2, \quad \gamma' \in \mathbb{Z}_2'. \tag{139}$$

Depending on the symmetry boundary conditions on $\mathcal{B}^{\text{sym}}$, these branes become symmetry generators or generalized charges.

We now apply the general procedure described in section 3.3 for the case of torsional cycles. For instance we have that

$$J^{\text{D3}} = dF_5 = g_3 \wedge d\phi_{\mathfrak{f}} - g_3' \wedge d\phi_{\mathfrak{f}'} + \dots, \tag{140}$$

---

[12]Where $\int_{\mathbb{RP}^5} \tilde{\Phi}_4 \wedge \tilde{\Phi}_4 \wedge d^{-1}\Phi_1$ is identified with the differential cohomology integral of [88], i.e. $\int_{\mathbb{RP}^5} \breve{u}_4 \star \breve{t}_1^{\text{RR}} \star \breve{t}_1^{\text{NSNS}}$. This identification similarly holds for the coefficients of the BF couplings, $\int_{\mathbb{RP}^5} d\tilde{\phi}_0 \wedge \tilde{\phi}_4 = 2$ means that $\int_{\mathbb{RP}^5} \tilde{\Phi}_0 \wedge d^{-1}\tilde{\Phi}_4 = \frac{1}{2}$, the latter is identified in differential cohomology $\int_{\mathbb{RP}^5} \breve{u}_4 \star \breve{u}_2 = \frac{1}{2}$.

where $g_3, g_3'$ are flat in the space where the SymTFT lives, $M_{4+1}$, and we describe the torsional cohomology in the continuum as

$$2\Phi_{\mathfrak{f}} = d\phi_{\mathfrak{f}}, \qquad 2\Phi_{\mathfrak{f}'} = d\phi_{\mathfrak{f}'}, \tag{141}$$

and from (63) and (64) we get the following BF action

$$S_{BF} = \alpha \int_{M_{4+2}} g_3 \wedge g_3', \tag{142}$$

where

$$\alpha = -\int_{L(\mathbf{X})} \phi_{\mathfrak{f}} \wedge d\phi_{\mathfrak{f}'} = 2, \tag{143}$$

which is exactly analogous to the BF-action for the bulk theory of $\mathcal{N} = 4$ $\mathfrak{su}(2)$ theory.

Let us now go back to the IHS equation (137) and look at the complex structure deformation that corresponds to the marginal coupling of the theory [91]. The deformed equation reads

$$x_1^2 + x_2^3 + x_3^3 + x_4^3 + \tau x_2 x_3 x_4 = 0 \subset \mathbb{C}^4, \tag{144}$$

where $\tau$ corresponds to the marginal coupling of the SCFT and therefore does not desingularize the geometry, as expected when activating the deformation corresponding to marginal couplings of the theory. This $\tau$ also corresponds to the complexified gauge coupling of the $\mathfrak{su}(2)$ when we think of this AD theory as a gauging of three AD$[A_1, A_3]$ theories, and it is identified with the complex structure of the torus when the theory is constructed via compactification of the $E_6$ minimal 6d $\mathcal{N} = (1, 0)$ SCFT on $T^2$ [91]. We now exploit the identification of the complex structure deformation parameter with $\tau$ in (144) and use how S-duality, $S$, and the $ST$ transformations act on it, to argue that $S$ and $ST$ are symmetries of the IHS equation, hence of the geometry, when $\tau = e^{i\pi/2}, e^{i\pi/3}$, respectively. For instance, the S-duality action by definition exchanges the magnetic 1-form symmetry with the electric one at an 5d effective topological field theory (BF-theory) level. This is indeed achieved when $S$ and $ST$ act on the torsional two cycles as follows

$$(\Phi_{\mathfrak{f}}, \Phi_{\mathfrak{f}'}) \mapsto M_S(\Phi_{\mathfrak{f}}, \Phi_{\mathfrak{f}'}), \qquad (\Phi_{\mathfrak{f}}, \Phi_{\mathfrak{f}'}) \mapsto M_{ST}(\Phi_{\mathfrak{f}}, \Phi_{\mathfrak{f}'}), \tag{145}$$

where $M_S$ and $M_{ST}$ are the monodromies defined by

$$M_S = \begin{pmatrix} 0 & 1 \\ -1 & 0 \end{pmatrix}, \qquad M_{ST} = \begin{pmatrix} 0 & 1 \\ -1 & -1 \end{pmatrix}. \tag{146}$$

At this level the symmetry acts geometrically, and the topological defect generating self-duality and -triality in this frame are hard to engineer as branes.[13] However, we can activate worldvolume fluxes on torsional cycles that induces $(p, q)$-string on the D3-brane.

**Induced $(p, q)$-String Charges on D3-branes and Symmetry Generators.** Instead of expanding the 5-form fluxes on torsional cycles we consider $(p, q)$-string charges on the D3-branes. As we explained in general in appendix A.1, in terms of magnetic sources we have

$$J^{D1} = f^{D3}\delta(D3), \qquad J^{F1} = (f')^{D3}\delta(D3), \tag{147}$$

and we choose

$$f^{D3} = d\phi_{\mathfrak{f}}, \qquad (f')^{D3} = d\phi_{\mathfrak{f}'}, \tag{148}$$

---

[13]See [4], for a geometric construction of these defects as degeneration of the link geometry at the boundary.

Table 2: Summary of topological defects construction in $[A_2, D_4]$ via IIB branes on $L(\mathbf{X})$ where $\mathbf{X}$ is the IHS defined in (137). D3-branes with and without world-volume flux provide two alternative but equivalent description of topological defects.

| Topological surface defects | Topological duality/triality defects |
|---|---|
| D3 on $H_2(L(\mathbf{X}), \mathbb{Z})$ | Isometry acting on $\mathfrak{f} \oplus \mathfrak{f}' \in H_2(L(\mathbf{X}), \mathbb{Z})$ |
| $(p,q)$-strings induced by $f^{\text{D3}} = d\phi_\mathfrak{f}, (f')^{\text{D3}} = d\phi_{\mathfrak{f}'}$ on D3 | 7-branes with monodromies $M_S, M_{ST} \in SL(2, \mathbb{Z})$ |

where the torsional pairs $(\phi_\mathfrak{f}, \Phi_\mathfrak{f})$ and $(\phi_{\mathfrak{f}'}, \Phi_{\mathfrak{f}'})$ have been introduced in (141). This induces the backgrounds for the 3-forms $H_3, F_3$,

$$H_3 = h_3 + d\phi_\mathfrak{f}, \qquad F_3 = f_3 + d\phi_{\mathfrak{f}'}, \tag{149}$$

where the second identity follows from the $SL(2, \mathbb{Z})$ covariant formalism [129], and we also turned on flat $f_3, h_3$ in $M_{4+1}$. Now the magnetically sourced Bianchi identity (140) gets modified,

$$J^{\text{D3}} = dF_5 = g_3 \wedge d\phi_\mathfrak{f} - g'_3 \wedge d\phi_{\mathfrak{f}'} = f_3 \wedge d\phi_\mathfrak{f} - h_3 \wedge d\phi_{\mathfrak{f}'}. \tag{150}$$

This implies that we can identify

$$g_3 \leftrightarrow f_3, \qquad g'_3 \leftrightarrow h_3. \tag{151}$$

It is now easy to verify that in this frame the action of $S$ and $ST$ on the torsional cycles (145) is equivalent to the action of the monodromy matrices $M_S$ and $M_{ST}$ on the $(f_3, h_3)$ pair and hence on the electrically charged $(p,q)$-strings. As we know from $\mathcal{N} = 4$ and its holographic construction, the self-duality and self-triality defects are engineered by 7-branes where the corresponding monodromy matrices act on the $(p,q)$-strings that generate the 1-form symmetries. In the next section, we will study properties of the SymTFT, the topological defects that generate the 1-form symmetries of the theory at the boundary from $(p,q)$-strings, and the self-dualities and -trialities topological defects from 7-branes. To summarize and conclude, mapping a discrete isometry of the geometry, which generates duality and triality defects for the engineered QFT, to the standard action of $SL(2, \mathbb{Z})$ on $(p,q)$-strings via monodromy matrices generated by 7-branes wrapping $L(\mathbf{X})$ is possible only when a world-volume flux on the D3-brane along torsional cycles is turned on, see table 2.

**Example: 4d $\mathcal{N} = 4$ from Type IIB.** In addition, as a cross check of our proposal, we can also apply this construction directly to the 4d $\mathcal{N} = 4$ SYM theories engineered in IIB on $\mathbf{X} = T^2 \times \mathbb{C}^2/\Gamma_{ADE}$, with link $L = T^2 \times S^3/\Gamma$. Consider the $A$-type theories, then $L(\mathbf{X})$ has non-trivial torsion link homology

$$\text{Tor}(H_2(L(\mathbf{X}), \mathbb{Z})) = \mathfrak{f} \oplus \mathfrak{f}' = \mathbb{Z}_N \oplus \mathbb{Z}'_N, \tag{152}$$

where

$$\mathfrak{f} = \Sigma_1 \otimes \gamma_1, \qquad \mathfrak{f}' = \Sigma'_1 \otimes \gamma_1, \tag{153}$$

with torsional $\gamma_1 \in H_1(S^3/\mathbb{Z}_N, \mathbb{Z})$ and $\Sigma'_1 \oplus \Sigma'_2 = H_1(T^2, \mathbb{Z})$. We can wrap D3-branes to generate topological surface defects of the SymTFT. The action of duality and triality defects corresponds to a finite subset of large diffeomorphisms of the $T^2$ acting on its complex structure. For fixed values of the complex structure $\tau = e^{i\pi/2}, e^{i\pi/3}$ they provide symmetries of the 4d QFT, where the action on the 1-cycles of the torus induces an action on the torsional part of

$H_2(L(\mathbf{X}), \mathbb{Z})$ via (153). We can then turn on fluxes on the D3-brane world-volume to map the topological defect to induced $(p, q)$-strings and to 7-branes with monodromies acting on the strings like for the $[A_2, D_4]$ theory case.

We can extend this also to more complicated examples, straightforwardly when the dimension of the conformal manifold is 1, or when we are able to identify the action of $S$ and $ST$ on a 1-dimensional subspace of the conformal manifold, [91]. It would be also interesting to generalize these to theories with a more complicated conformal manifold. We leave this to future work.

# 4 Hanany-Witten effect: Generalized charges and anomalies

We have so far introduced the notion of charges of topological defects in terms of brane linking. In all of the above, we explained the brane origin of the action of codimension-$(q + 1)$ topological defects on charged $q$-dimensional extended operators, i.e. $q$-charges.

**Generalized Charges.** It is however also known field theoretically that codimension-$(p + 1)$ topological defects can act on extended operators of dimension $q \neq p$ as higher-representations [45, 110]. In this section we demonstrate how branes know about this generalized concept of $q$-charges through the so-called Hanany-Witten effect [133]. We will furthermore show that this effect is intimately related to additional couplings in the topological bulk theory, corresponding to 't Hooft anomalies of the symmetries generated by these same branes, depending on the boundary conditions, or leading to a twisted DW theory.

Our earlier notion of charge had two origins: either via the $d$-dimensional flux sector dimensionally reduced on $L_{\text{int}}$ or the Bianchi identities, where we truncate everything to the topological sector that describes the behaviour of finite flat abelian fields.

The starting point of our present discussion is one particular case of interest when the dimensionally reduced Bianchi identities feature three external fluxes satisfying

$$df^{(1)} = f^{(2)} \wedge f^{(3)} + j^{(1)}, \qquad df^{(2)} = j^{(2)}, \qquad df^{(3)} = j^{(3)}, \tag{154}$$

with the $f^{(2)} \wedge f^{(3)}$ term in the first relation originates from a non-trivial Chern-Simons term in the original $(D + 1)$-dimensional action (41). These are exactly the type of Bianchi identities that lead to Hanany-Witten transitions [133]. One can quickly notice the potential for non-trivial physics in this situation by differentiating the above equation

$$0 = j^{(2)} \wedge d^{-1} j^{(3)} + (-1)^{(\deg f^{(2)} + 1)(\deg f^{(3)} + 1)} j^{(3)} \wedge d^{-1} j^{(2)} + d j^{(1)}. \tag{155}$$

The first consequence of this relation is that the two branes corresponding to magnetic sources $j^{(2)}$ and $j^{(3)}$ link in the $(d + 1)$-dimensional space-time. Exchanging the position of the two branes in the linking direction generates a difference in the total linking number. This number must be fixed, due to the Bianchi identity realizing charge conservation, by the creation of branes corresponding to the $j^{(1)}$ magnetic source extending along the linking direction[14] (see figure 7), see [134]. The crucial insight we provide in this work is how to interpret this bulk property of branes in terms of the symmetry generators which they correspond to in the field theory. The Hanany-Witten (HW) effect can be interpreted in two ways depending on the allowed topological boundary conditions, which concretely means how we place the branes in $M_{d_{\text{QFT}}+1}$: this encodes

---

[14]See [133] for the electric point of view on how the change of linking leads to the creation of a brane.

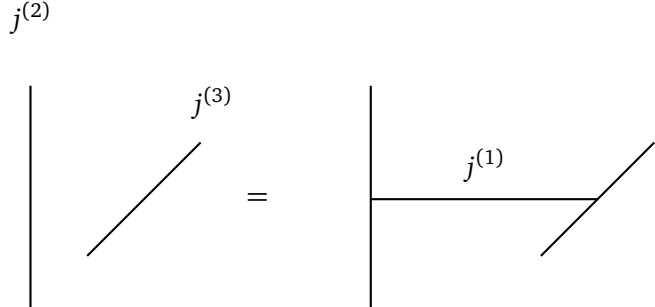

Figure 7: Passing two branes corresponding to magnetic sources $j^{(2)}$ and $j^{(3)}$ through each other can result in the creation of a new brane, stretching between them, corresponding to magnetic source $j^{(1)}$, if the currents are linked by relation (155).

1. the $q$-charges (or generalized charges) of a symmetry. This occurs, when the branes in the HW-configuration are such that one wraps the radial direction, and the other does not.

2. the (mixed) 't Hooft anomalies or the topological coupling leading to a twisted DW theory: this occurs when none of the branes extend along the radial direction.

These implications will be discussed in subsequent sections.

## 4.1 The Hanany-Witten effect

We will now discuss the relevant Hanany-Witten (HW) transitions, following the original effect discussed in [133]. For our symmetry considerations we will require various HW-setups, in type II and M-theory.

Before exploring generalizations, we first illustrate the effect in a simple example.

**Motivating Example.** Consider the following configuration of branes in type IIA on a generic 10d spacetime parameterized by coordinates $\{x_i : i = 0, \ldots, 9\}$.

| Brane | $x_0$ | $x_1$ | $x_2$ | $x_3$ | $x_4$ | $x_5$ | $x_6$ | $x_7$ | $x_8$ | $x_9$ |
|-------|-------|-------|-------|-------|-------|-------|-------|-------|-------|-------|
| NS5 | X | X | X | X | X | X | | | | |
| D8 | X | X | X | X | X | X | X | X | X | |

(156)

The NS5-brane is a magnetic source for the NS-NS gauge field $B_2$ with field strength $H_3$. Using this fact, one can consider the concept of *linking* between the two branes by computing the flux

$$\int_{x_6, x_7, x_8} H_3 \,. \tag{157}$$

This computes the total linking number of D8-branes with all NS5-branes, in a way we will shortly explain.

The key observation is that the NS-NS 3-form flux $H_3$, pulled back to the world-volume of the D8-brane, is trivial in cohomology. Indeed, let $a_1$ denote the $U(1)$ gauge field localized on the D8-brane, and let $f_2$ denote its field strength. The pullback of the NS-NS 2-form $B_2$ to the D8-brane world-volume combines with $f_2$ in the gauge-invariant and globally defined combination $\mathcal{F}_2 = f_2 - B_2$. Making use of the Bianchi identity $df_2 = 0$, we see that $H_3 = -d\mathcal{F}_2$. Naïvely, we may conclude that the linking number defined above is therefore always necessarily zero, if the space spanned by $x_6$, $x_7$, $x_8$ is a closed, compact, oriented 3-manifold. If this

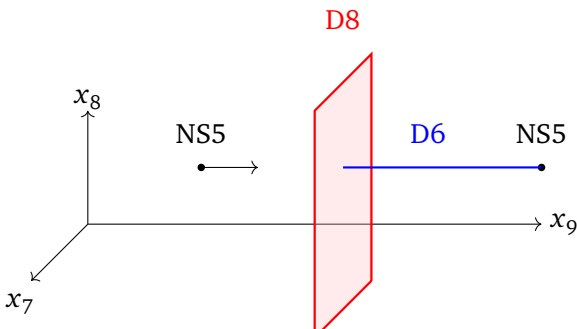

Figure 8: The D8/ NS5 Hanany-Witten configuration projected onto $\{x_7, x_8, x_9\}$ directions. The NS5 brane is a point and the D8 is a plane in the $\{x_7, x_8\}$ directions. Passing the NS5 brane through the D8 brane generates a D6 brane attachment (a line along the $x_9$ direction).

were the case, it would not be possible to move the NS5-brane across the D8-brane. Such a move is allowed, however at the cost of creating a D6-brane in the process (see figure 8).

Recall that a D6-brane ending on a D8-brane acts as a magnetic source for the $a_1$ gauge field on the D8-brane, modifying the Bianchi identity for $f_2$ to $df_2 = \pm\delta_3$, where $\delta_3$ represents the locus inside the D8-brane where the D6-brane ends, and the sign keeps track of orientation.

We will utilize tables of the following type as a compact way of summarizing Hanany-Witten configurations:

| Brane | $x_0$ | $x_1$ | $x_2$ | $x_3$ | $x_4$ | $x_5$ | $x_6$ | $x_7$ | $x_8$ | $x_9$ |
|-------|-------|-------|-------|-------|-------|-------|-------|-------|-------|-------|
| D8 | X | X | X | X | X | X | X | X | X | |
| NS5 | X | X | X | X | X | X | | | | |
| D6 | X | X | X | X | X | X | | | | X |

(158)

We now look to find general classes of brane configurations which undergo HW transitions.

**Using Brane Linking.** A natural language to discuss these transitions in generality is the notion of string theoretic linking of two magnetic sources introduced in section 3.2

$$\mathcal{L}_d(\mathcal{W}^{(i)}, \mathcal{W}^{(j)}) = \int_{M_d} J^{(i)} \wedge d^{-1} J^{(i)} = \int_{M_d} dF^{(i)} \wedge F^{(j)}. \tag{159}$$

Recall that this is a topological property associated to two branes, whose magnetic sources are localized on sub-manifolds $\mathcal{W}^{(i)}, \mathcal{W}^{(j)}$. Notice that since $dF^{(i)} = \delta(\mathcal{W}^{(i)})$, this integral is readily re-written in terms of a lower-dimensional integral as in (157)

Let us consider two branes in string/M-theory. We look for configurations in which a subset of the directions in the world-volumes of the branes link (in the above sense) inside a subset of the total directions of spacetime. The general situation we face in this section is indeed such that the dimension formula reads (31). We have several cases.

**Direct Linking in Spacetime.** In the simplest case, all world-volume directions of both branes link inside the entire spacetime. An example is furnished by an NS5-brane and a D2-brane in Type IIA string theory:

| Brane | $x_0$ | $x_1$ | $x_2$ | $x_3$ | $x_4$ | $x_5$ | $x_6$ | $x_7$ | $x_8$ | $x_9$ |
|-------|-------|-------|-------|-------|-------|-------|-------|-------|-------|-------|
| NS5 | X | X | X | X | X | X | | | | |
| D2 | | | | | | | X | X | X | |

(160)

The integer linking number for this configuration is

$$\text{Link}_{X_{10}}(M_6^{\text{NS5}}, M_3^{\text{D2}}), \qquad X_{10} = \{x_0, \dots, x_9\}, \quad M_6^{\text{NS5}} = \{x_0, \dots, x_5\}, \quad M_3^{\text{D2}} = \{x_6, \dots, x_8\}. \tag{161}$$

This case is however not relevant for the applications in this paper.

**HW-Configurations for Symmetries.** Next, we have the case in which the two branes are simultaneously extending along a subset of the directions of spacetime. The problem is effectively reduced from $D = 10$ or $11$ to a smaller dimensionality $D'$, in which the remaining world-volume directions of the branes link. This type of configuration corresponds to setups of HW type, which we classify below. An example is furnished by the original HW configuration of an NS5-brane and a D5-brane:

| Brane | $x_0$ | $x_1$ | $x_2$ | $x_3$ | $x_4$ | $x_5$ | $x_6$ | $x_7$ | $x_8$ | $x_9$ |
|---|---|---|---|---|---|---|---|---|---|---|
| NS5 | X | X | X | X | X | X | | | | |
| D5 | X | X | X | | | | X | X | X | |

$$\tag{162}$$

The common directions are $x_{0,1,2}$ and the relevant linking number is

$$\text{Link}_{X_7}(M_3^{\text{NS5}}, M_3^{\text{D5}}), \qquad X_7 = \{x_3, \dots, x_9\}, \quad M_3^{\text{NS5}} = \{x_3, \dots, x_5\}, \quad M_3^{\text{D5}} = \{x_6, \dots, x_8\}. \tag{163}$$

**HW-Configurations for Generalized Charges.** Finally, for completeness we tabulate all possible Hanany-Witten setups in type II and M-theory, which are relevant for computing generalized charges. An example appeared already in [1]. These configurations can be grouped together as follows:

I) The first class is realized in IIB or IIA and is given by the following brane system:

| Brane | $x_0$ | $x_1$ | $x_2$ | $x_3$ | $x_4$ | $x_5$ | $x_6$ | $x_7$ | $x_8$ | $x_9$ |
|---|---|---|---|---|---|---|---|---|---|---|
| D$p$ | X | X | X | X | X | X | X | X | | |
| D$p'$ | X | | | | | | | | X | |
| F1 | X | | | | | | | | | X |

$$\tag{164}$$

where $8 = p + p'$, and we can apply T-duality in the $x_{1,2,3,4,5,6,7,8}$ directions. In addition when $p = 7$ and $p' = 1$, the role of F1 and the D1 can be exchanged, and in generalised to $(p,q)$-strings and 7-branes.

II) The second class is a special case in IIB given by the following system:

| Brane | $x_0$ | $x_1$ | $x_2$ | $x_3$ | $x_4$ | $x_5$ | $x_6$ | $x_7$ | $x_8$ | $x_9$ |
|---|---|---|---|---|---|---|---|---|---|---|
| $[p,q]$7-brane | X | X | X | X | X | X | X | X | | |
| $(p',q')$5-brane | X | X | X | X | X | | | | $p'x_8 = q'x_9$ | |
| $(r,s)$5-brane | X | X | X | X | X | | | | $rx_8 = sx_9$ | |
| $(p,q)$5-brane | X | X | X | X | X | | | | $px_8 = qx_9$ | |

$$\tag{165}$$

where $px_8 = qx_9$ means that the 5-brane extend along this locus. The last 5-brane is the one created once the 7-brane crosses the junction between the $(p',q')$ 5-brane and the $(r,s)$ 5-brane. Finally the total 5-brane charge must be conserved, i.e. $p + p' + r = 0$ and $q + q' + s = 0$.

III) The third class is related to the original Hanany-Witten setup by T-dualities:

| Brane | $x_0$ | $x_1$ | $x_2$ | $x_3$ | $x_4$ | $x_5$ | $x_6$ | $x_7$ | $x_8$ | $x_9$ |
|---|---|---|---|---|---|---|---|---|---|---|
| D$p$ | X | X | X | X | X | X | | | | |
| NS5 | X | X | X | | | | X | X | X | |
| D$p'$ | X | X | X | | | | | | | X |

(166)

where $p' = p - 2$ and we can apply T-duality[15] in the $x_{1,2,6,7,8}$ directions. In the case $p = 5$, $p' = 3$ we also have a generalization, with a $(p, q)$ 5-brane in the first row, a $(p', q')$ 5-brane in the second row, and $pq' - p'q$ D3-branes in the third row [136].

IV) The fourth class is a single brane system in M-theory:

| Brane | $x_0$ | $x_1$ | $x_2$ | $x_3$ | $x_4$ | $x_5$ | $x_6$ | $x_7$ | $x_8$ | $x_9$ | $x_{10}$ |
|---|---|---|---|---|---|---|---|---|---|---|---|
| M5 | X | X | X | X | X | X | | | | | |
| M5' | X | X | | | | | X | X | X | X | |
| M2 | X | X | | | | | | | | | X |

(167)

**Details of HW-configurations.** We have already discussed class (III) above. We note that similar remarks apply in general to $D(p-2)$ branes ending on D$p$-branes, and refer the reader to [136] for a generalization of the setup of class (III), involving the creation of $pq' - p'q$ D3-branes when a $(p, q)$ 5-brane and a $(p', q')$ 5-brane are passed across each other.

Let us now turn to setups of class (IV) in M-theory. This Hanany-Witten setup is discussed in [137] and can be derived in a way analogous to the argument for class (III). In this case, we use the fact that the world-volume of an M5-brane supports a localized 2-form field $b_2$, with self-dual field strength $h_3$. The latter combines with the pullback of the M-theory 3-form $C_3$ into the gauge-invariant and globally defined combination $\mathcal{H}_3 = h_3 - C_3$. As a result, on the world-volume of the M5-brane we have $G_4 = -d\mathcal{H}_3$, where we have made use of the Bianchi identity $dh_3 = 0$. Once again, this would naïvely suggest the vanishing of the linking number $\int_{M_4} G_4$ computed on the orthogonal directions of the second M5-brane $M_4$. The correct conclusion is that, when the two M5-branes are passed across each other, an M2-brane is generated. In fact, this is the correct object to modify the Bianchi identity for $h_3$ from $dh_3 = 0$ to $dh_3 = \pm\delta_4$, where $\delta_4$ represent the locus inside the M5-brane where the M2-brane ends, and the sign keeps track of orientation.

The brane setups of classes (I) may be derived from those of class (III) with the help of S- and T-dualities. Let us start from the class (III) setup with $p = 3$, $p' = 1$, describing a Hanany-Witten move in which a D1-brane is generated when an NS5-brane and a D3-brane are passed across each other. By S-duality, this is mapped to a setup of class (I) with $p = 3$, $p' = 5$: an F1-string is generated when a D5-brane and a D3-brane are passed across each other [137]. This can also be seen as follows. The D5-brane is a magnetic source for the RR 3-form field strength $F_3$. The relevant linking is then measured by integrating $F_3$ on the world-volume of the D3-brane. Invariance of the D3-brane under S-duality, however, implies that the electromagnetic dual $\tilde{a}_1$ of the gauge field $a_1$ on the brane combines with the pullback of the RR 2-form into the gauge-invariant combination $\tilde{\mathcal{F}}_2 = \tilde{f}_2 - C_2$, where $\tilde{f}_2$ is the field strength of $\tilde{a}_1$. Setting $C_0 = 0$ for simplicity, on the world-volume of the D3-brane we have $F_3 = -d\tilde{\mathcal{F}}_2$. The argument then proceeds as for class (III). Once the setup of class (I) is established for $p = 3$, other values for $p$ are derived by T-duality.

---

[15]Note that T-duality along the NS5-brane world-volume results in another NS5-brane, whereas transverse to it, results in a KK-monopole [135].

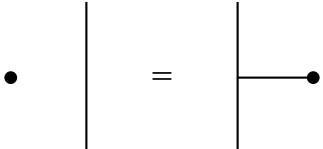

Figure 9: In this figure dots and lines are defects in the boundary QFT of interest (this could be thought of as a $(1+1)d$ system or a 2d projection of a higher-dimensional analogue). In the higher-dimensional configuration, dragging the point-like operator through the extended line operator generates a non-genuine operator (or twisted sector, if the line defect is topological).

## 4.2 Generalized charges

Let us denote the HW brane pair (brane$_1$, brane$_2$). Passing one through the other generates the third brane brane$_3$. Suppose that we pick brane$_1$ to be parallel to the boundary, and brane$_2$ to wrap the radial direction. Field theoretically, brane$_1$ corresponds to a topological symmetry generator $D_p$, whilst brane$_2$ is a non-topological (extended) defect $\mathcal{O}_q$.

The HW transition implies that passing $\mathcal{O}_q$ defect through $D_p$ creates a third topological operator $D_l$ (brane$_3$ necessarily does not wrap the radial direction) which is *attached* to $\mathcal{O}_q$. Generically this process maps a genuine operator to non-genuine one (see figure 9 for an example of this effect). This is precisely the charge of a non-invertible abelian categorical symmetry on charged defects, which does not preserve the dimensionality of the defects [1, 56, 61].

**Example: Klebanov-Strassler.** From the Bianchi identity $dF_5 + H_3 F_3 = J^{(D3)}$ we learn that there is a Hanany-Witten effect between a D3 brane and a D5 on $S^3$ which generates an F1 string stretched between the two.

It is known that the $\mathcal{G} = PSU(M)$ theory has a non-invertible 0-form symmetry. In [1] the non-invertible 0-form topological operator was given a string theory origin as a D5-brane wrapped on $S^3 \subset T^{1,1}$. Furthermore, its non-invertible action on 't Hooft lines was explained using the Hanany-Witten effect. Now the wrapped D3 is perpendicular to the boundary (the 1-form symmetry is gauged), and the brane creation turns a genuine line operator into a non-genuine one. We refer the reader to appendix B for a field theory analysis of non-invertible actions on line operators.

**Example: Maldacena-Nunez.** A second description (MN) of 4d $\mathcal{N} = 1$ SYM is given in [138]. We begin with the 6d $\mathcal{N} = (1, 1)$ LST living on $M$ NS5 branes in IIB. The four-dimensional theory is obtained via a topologically twisted $S^2$ reduction.

For the sake of brevity, we use the fact that the for the purposes of our computations the above background is S-dual to that of Klebanov Strassler, in the sense that we replace

$$\int_{S^3} F_3 = M \leftrightarrow \int_{S^3} H_3 = M \,. \tag{168}$$

The derivations of the BF terms and anomalies proceed identically. We therefore identify the brane responsible for the non-invertible 0-form symmetry as

$$D_3(M_3^{\mathrm{NS5}}) \leftrightarrow \mathrm{NS5}(M_3^{\mathrm{NS5}} \times S^3) \,. \tag{169}$$

On the LHS we use the notation for topological defects $D_q(M_q)$, i.e. a $q$-dimensional topological defect on the spacetime manifold $M_q$, whereas on the RHS we use the notation of a brane

(NS5 or D$p$ or M$p$ wrapped on an internal cycle an $M_q$). Following an analogous procedure as appendix B of [1] it is easy to see that this brane's topological world-volume terms correctly reproduce the expected TQFT stacking and therefore fusion rules known from field theory [55].

Once again we consider the three brane origins of 2-surfaces in the 5d bulk: F1-, D1- and wrapped D3-branes. However, since in this setup there is only $H_3$ flux over the $S^3$, the linking configurations are simpler. Only the D1 and D3 wrapped on $S^2$ link in the 5d bulk. We can therefore identify

$$
\begin{aligned}
D_2(M_2^{\mathrm{D3}}) &\longleftrightarrow D3(M_2^{\mathrm{D3}} \times S^2)\,, \\
\widehat{D_2}(M_2^{\mathrm{D1}}) &\longleftrightarrow D1(M_2^{\mathrm{D1}})\,,
\end{aligned}
\tag{170}
$$

as the generators of the electric (magnetic) 1-form symmetries in the $SU(N)(PSU(N))$ theories respectively.

The D3/ D5 Hanany-Witten effect responsible for generalized charges in the Klebanov-Strassler solution has an S-dual partner involving a D3/ NS5 transition.

Consider a boundary condition such that the NS5 and stretched D1-brane are topological, and the D3 is not ($\mathcal{G} = PSU(N)$). The HW transition describes the non-invertible action of $D_3$ on the charged 't Hooft line (the wrapped D3-brane) by attaching a topological 2-surface (the D1-brane).

**Example: M-theory on $G_2$.**  M-theory on the singular $G_2$ holonomy manifold $\mathbb{C}^2/\mathbb{Z}_N \to S^3$ models the UV of 4d $\mathcal{N} = 1$ pure SYM [139–142]. The boundary geometry is $S^3/\mathbb{Z}_N \to S_3$. The link $L_6$ therefore has homology groups

$$
H_\bullet(L_6, \mathbb{Z}) = \{\mathbb{Z}, \mathbb{Z}_N, 0, \mathbb{Z} \oplus \mathbb{Z}, \mathbb{Z}_N, 0, \mathbb{Z}\}\,.
\tag{171}
$$

We propose that the branes generating the 1- and 0-form symmetries respectively are[16]

$$
\begin{aligned}
\mathrm{M5}(\Sigma_2 \times \gamma_1 \times \mathrm{S}^3) &\longleftrightarrow D_2(\Sigma_2)\,, \\
\mathrm{M5}(\mathrm{M}_3 \times \mathrm{S}^3/\mathbb{Z}_\mathrm{N}) &\longleftrightarrow D_3(\mathrm{M}_3)\,.
\end{aligned}
\tag{172}
$$

From the Bianchi identity

$$
dG_7 - \frac{1}{2}G_4^2 = J^{G_7}\,,
\tag{173}
$$

one can see there is a Hanany-Witten transition involving two M5 branes, generating an M2 brane, as demonstrated in (167).

The global variant $\mathcal{G} = PSU(N)$ corresponds to picking the M5-brane wrapping the torsional 4-cycle to be perpendicular to the boundary, whilst the other is parallel. In this case, the Hanany-Witten effect produces a topological attachment to the non-topological string charged under the 1-form symmetry: turning it from a genuine to non-genuine line operator.

## 4.3  Hanany-Witten and 't Hooft anomalies

Now suppose that both (brane$_1$, brane$_2$) are *parallel* to the boundary. They therefore both correspond to topological defects $D_p, D_{p'}$ whose non-trivial linking forces the creation of a brane in the radial direction, corresponding to a non-topoloical (extended) defect $\mathcal{O}_q$.

We will now argue that such a configuration indicates the existence of certain 't Hooft anomalies using two complimentary approaches.

1. The first is that the created brane wrapped along the radial direction creates a non-topological ambiguity in terms of how the topological defects are separated in the bulk. When we push these to infinity we argue that this signals the presence of an anomaly.

---

[16]Identifying geometrically $U(1)_R$ symmetry and its breaking to $\mathbb{Z}_{2N}$ is still a challenge in the geometric engineering in M-theory [143].

2. The second involves directly projecting the bulk Hanany-Witten configuration to the boundary. The bulk picture becomes a junction in the boundary, which the Hanany-Witten computation tells us must be charged under certain symmetries. This is another hallmark of a 't Hooft anomaly.

The anomalies are computed using suitable intersections of branes which depend on the space-time dimension. In particular, we look for intersections such that one of the participating branes links with the intersection of the other two, as discussed in section 3.5.

**Anomalies from Topological Defects.** Coupling a theory to a background for a higher-form symmetry amounts to inserting a mesh of the corresponding topological defects. This mesh contains junctions, inconsistencies of which can signal the presence of anomalies [127].

For example, consider a theory with both a $p$ and $(d - p - 1)$-form symmetry. If the codimension-$p$ topological operators generating the former symmetry are charged under the codimension-$(d - p - 1)$ topological operators generating the latter, the two symmetries participate in a mixed 't Hooft anomaly [127]. This is because it is not possible to insert a mesh of both defects (i.e. couple to both backgrounds simultaneously) in a consistent manner, due to their action on one another.

The above is a special case where the two participating symmetries have appropriate dimension such that their operators link in spacetime. However, it is generically possible that a codimension$-(p + 1)$ operator can also act on an extended operator of dimension $q \neq p$. In this way, we are able to explore 't Hooft anomalies involving higher-form symmetries of different degrees from the perspective of their topological operators and their junctions.

In general, a mixed 't Hooft anomaly between two (or more) higher-form symmetries is encoded in the junctions of their corresponding defects. We argue that this information is naturally encoded in our understanding of branes. For example, two branes may intersect and generate a third (by Hanany-Witten). If this third brane is *charged* under one of the symmetries generated by one of the intersecting branes - this junction signals a mixed 't Hooft anomaly.

**Example: $\mathcal{N} = 1$ SYM.** In the three presentations of 4d $\mathcal{N} = 1$ $\mathfrak{su}(M)$ SYM presented earlier, in each case there was a Hanany-Witten configuration of branes which in the $\mathcal{G} = PSU(M)$ variant described a generalized charge. By the above argument, in the frame where we pick boundary conditions such that $\mathcal{G} = SU(M)$, these configurations also signal the mixed 't Hooft anomaly in these models.

## 4.4 Example: 4d $\mathcal{N} = 4$ $\mathfrak{so}(4n)$ SYM

In this section we demonstrate that the HW effect is responsible for generalized charges in several global variants of the $\mathfrak{so}(4n)$ theory, and the mixed anomaly

$$\mathcal{A} = \frac{1}{2} \int A_1 C'_2 B_2 \, , \tag{174}$$

in the $\mathcal{G} = SO(4n)$ theory, where $A_1$ is a background for $\mathbb{Z}_2^{(0)}$ and $C'_2, B_2$ are both $\mathbb{Z}_2^{(1)}$ backgrounds. From the SymTFT/ Gauss law perspective we can read off the brane origins of the topological symmetry generators [88]. For convenience we summarize these findings in table 3.

$\mathcal{G} =$**SO(4n).** For $\mathcal{G} =$SO(4n), the brane identification is [88]

$$\begin{aligned} D_2(M_2) &\leftrightarrow D5(M_2 \times \mathbb{RP}^4) \, , \\ D_3(M_3) &\leftrightarrow D3(M_3 \times \mathbb{RP}^1) \, , \end{aligned} \tag{175}$$

Table 3: SymTFT and brane origins of symmetry generators in various global forms of $\mathfrak{so}(4n)$ 4d SYM theories. We will not use the background field for $\mathbb{Z}_2^{1,v}$ in this work, but for completeness it will be a linear combination of $C_2$ and $B_2$.

| | Symmetry | Background Field | Brane Origin of Sym Generator |
|---|---|---|---|
| SO(4n) | $\mathbb{Z}_2^{(0)}$ | $A_1$ | D3 on $\mathbb{RP}^1$ |
| | $\mathbb{Z}_2^{(1)}$ | $C_2'$ | D5 on $\mathbb{RP}^4$ |
| | $\mathbb{Z}_2^{(1)}$ | $B_2$ | NS5 on $\mathbb{RP}^4$ |
| Spin(4n) | $\mathbb{Z}_2^{(0)}$ | $A_1$ | D3 on $\mathbb{RP}^1$ |
| | $\mathbb{Z}_2^{(1,s)}$ | $C_2$ | D1 on pt $\in \mathbb{RP}^5$ |
| | $\mathbb{Z}_2^{(1,c)}$ | $B_2$ | NS5 on $\mathbb{RP}^4 \oplus$ D1 on pt $\in \mathbb{RP}^5$ |
| | $\mathbb{Z}_2^{(1,v)}$ | | NS5 on $\mathbb{RP}^4$ |
| PO(4n) | $\mathbb{Z}_2^{(2)}$ | $A_3$ | D3 on $\mathbb{RP}^3$ |
| | $\mathbb{Z}_2^{(1)}$ | $C_2'$ | D5 on $\mathbb{RP}^4 \oplus$ F1 on pt $\in \mathbb{RP}^5 + \int A_1 B_2$ |
| | $\mathbb{Z}_2^{(1)}$ | $B_2'$ | F1 on pt $\in \mathbb{RP}^5$ |
| Pin$^+$(4n) | $\mathbb{Z}_2^{(2)}$ | $A_3$ | D3 on $\mathbb{RP}^3$ |
| | $\mathbb{Z}_2^{(1)}$ | $C_2$ | D1 on pt $\in \mathbb{RP}^5$ |
| | $\mathbb{Z}_2^{(1)}$ | $B_2$ | NS5 on $\mathbb{RP}^4 \oplus$ D1 on pt $\in \mathbb{RP}^5 + \int A_1 C_2'$ |
| Sc(4n) | $\mathbb{Z}_2^{(0)}$ | $A_1$ | D3 on $\mathbb{RP}^1 + \int B_2 C_2'$ |
| | $\mathbb{Z}_2^{(1)}$ | $C_2$ | D1 on pt $\in \mathbb{RP}^5$ |
| | $\mathbb{Z}_2^{(1)}$ | $B_2'$ | F1 on pt $\in \mathbb{RP}^5$ |

where $D_2, D_3$ are the generators of $\mathbb{Z}_2^{(1,C')} \subset \mathbb{Z}_2^{(1,C')} \times \mathbb{Z}_2^{(1,B)} = \Gamma^{(1)}$ and $\mathbb{Z}_2^{(0)}$ respectively. The charged lines under the $\mathbb{Z}_2^{(1,B)}$ 1-form symmetry factor are

$$\mathcal{O}_1(\Sigma_1) \longleftrightarrow \text{F1}(\Sigma_1 \times \mathbb{R}^{>0}). \tag{176}$$

These three wrapped branes form a HW configuration, from which we observe that the $\mathbb{Z}_2^{(1,C')}$ and $\mathbb{Z}_2^{(0)}$ symmetry defects intersect in 4d in a line which is charged under $\mathbb{Z}_2^{(1,B)}$ factor: this signals the presence of the mixed 't Hooft anomaly between all three symmetries. One can derive a similar result using the S-dual branes: pulling the NS5-brane, which generates $\mathbb{Z}_2^{(1,B)}$, across $D_3$ generates a D1-brane which is charged under $\mathbb{Z}_2^{(1,C')}$.

The theory with gauge group $\mathcal{G} = \text{Spin}(4n)$ is related to $\mathcal{G} = \text{SO}(4n)$ via gauging of the 1-form symmetry (for more details also the categorical structure, see [58]). This maps the mixed anomaly to a split 2-group symmetry. In this way the HW brane configuration explained above also encodes this split 2-group global symmetry.

$\mathcal{G} = $**Sc(4n).** We now consider how the non-invertible 0-form symmetry in the $\mathcal{G} = \text{Sc}(4n)$ variant acts on the charged lines of the theory. The 0-form symmetry is generated by

$$D_3(M_3) \longleftrightarrow \text{D3}(M_3 \times \mathbb{RP}^1). \tag{177}$$

Meanwhile the invertible 1-form symmetries have non-topological charged lines

$$\begin{aligned}
\mathcal{O}_1(\Sigma_1) &\longleftrightarrow \text{D5}(\Sigma_1 \times \mathbb{R}^{>0} \times \mathbb{RP}^4), \\
\mathcal{O}_1'(\Sigma_1') &\longleftrightarrow \text{NS5}(\Sigma_1' \times \mathbb{R}^{>0} \times \mathbb{RP}^4).
\end{aligned} \tag{178}$$

If we pass $\mathcal{O}1$ or $\mathcal{O}1'$ through $D_3(M_3)$, there is a non-trivial Hanany-Witten move which generates an F1 or D1 brane respectively. These are *topological* operators which respectively generate the invertible 1-form symmetry which acts on the other charged line. These results agree with the complementary field theory analysis, reported in appendix B.

$\mathcal{G}$ **=PO($4n$).** Now we look at how the non-invertible 1-form symmetry in the $\mathcal{G}$ =PO($4n$) variant acts on the 2-surfaces charged under the invertible 2-form symmetry.

In this case there are a number of non-invertible actions we should consider. The non-invertible 1-form symmetry is generated by

$$D_2(M_2) \longleftrightarrow \text{D5}(M_2 \times \mathbb{RP}^4).\tag{179}$$

On the other hand, the charged 2-surfaces are given by

$$\begin{aligned}
\mathcal{O}2(\Sigma_2) &\longleftrightarrow \text{D3}(\mathbb{R}^{>0} \times \Sigma_2 \times \mathbb{RP}^1), \\
\mathcal{O}2'(\Sigma_2) &\longleftrightarrow \text{NS5}(\mathbb{R}^{>0} \times \Sigma_2' \times \mathbb{RP}^3).
\end{aligned}\tag{180}$$

There is a non-trivial Hanany-Witten move for both of these. First, passing $\mathcal{O}_2$ through $D_2$ generates an F1-string stretched between the two: this is the generator of the invertible 2-form symmetry under which $\mathcal{O}_2'$ is charged. On the other hand, passing $\mathcal{O}_2'$ through $D_2$ generates a D3-brane: this is the generator of the other invertible 2-form symmetry which acts on $\mathcal{O}_2$.

$\mathcal{G}$ **=Pin$^+$($4n$).** The non-invertible 1-form symmetry in this case is generated by

$$D_2(M_2) \longleftrightarrow \text{NS5}(M_2 \times \mathbb{RP}^4).\tag{181}$$

On the other hand, the charged 2-surfaces are given by

$$\begin{aligned}
\mathcal{O}_2(\Sigma_2) &\longleftrightarrow \text{D3}(\mathbb{R}^{>0} \times \Sigma_2 \times \mathbb{RP}^1), \\
\mathcal{O}_2'(\Sigma_2) &\longleftrightarrow \text{D5}(\mathbb{R}^{>0} \times \Sigma_2' \times \mathbb{RP}^3).
\end{aligned}\tag{182}$$

There is also a non-trivial Hanany-Witten move for both of these. First, passing $\mathcal{O}_2$ through $D_2$ generates an D1-string stretched between the two: this is the generator of the invertible 2-form symmetry under which $\mathcal{O}_2'$ is charged. On the other hand, passing $\mathcal{O}_2'$ through $D_2$ generates a D3-brane: this is the generator of the other invertible 2-form symmetry which acts on $\mathcal{O}_2$.

**Outer-Automorphism Action on $\mathcal{G} = \text{Spin}(4n)$.** There is also a brane origin to the $\mathbb{Z}_2^{(0)}$ outer-automorphism in the $\mathcal{G} = \text{Spin}(4n)$ theory which exchanges

$$\mathcal{O}_1^{(s)} \longleftrightarrow \mathcal{O}_1^{(c)},\tag{183}$$

where $\mathcal{O}_1^{(s,c)}$ are the spinor/co-spinor Wilson lines. An equivalent way of describing this action is shown in figure 10. In terms of branes, these lines are [2]

$$\begin{aligned}
\mathcal{O}_1^{(s)} &\longleftrightarrow \text{D5}(\text{M}_1 \times \mathbb{R}^{>0} \times \mathbb{RP}^4), \\
\mathcal{O}_1^{(c)} &\longleftrightarrow \text{D5}(\text{M}_1 \times \mathbb{R}^{>0} \times \mathbb{RP}^4) \oplus \text{F1}(\text{M}_1 \times \mathbb{R}^{>0}).
\end{aligned}\tag{184}$$

Furthermore, it is known that the brane generating the outer-automorphism symmetry is

$$D_3(M_3) \longleftrightarrow \text{D3}(M_3 \times \mathbb{RP}^1).\tag{185}$$

We now discuss the action of this operator at the level of the branes. In the arrangement shown in figure 11, we consider what happens when the wrapped D5 pierces through the wrapped

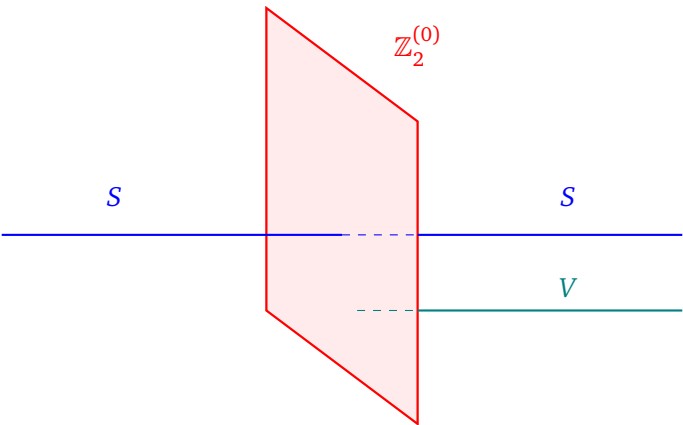

Figure 10: $\mathbb{Z}_2^{(0)}$ outer-automorphism action, depicted in terms of defects: the outer automorphism acts as $S$ maps to $C$. It is useful in the following to write $C = V \otimes S$, as this is how the branes will realize the action.

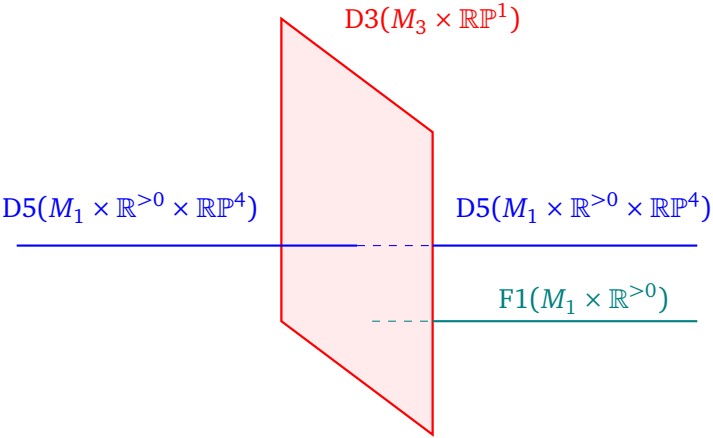

Figure 11: The same $\mathbb{Z}_2^{(0)}$ outer-autmorphism action as in figure 10, now in terms of branes. The wrapped D3 brane induces a jump in F1 flux which is absorbed by emitting an F1 string.

D3 representing the outer-automorphism generator. Since the D3-brane is a source for the RR $C_4$ field, as we pass from left to right there is a flux jump which induces a non-trivial F1 charge via the 6d 5-brane world-volume coupling

$$\int B_2 C_4 \,, \tag{186}$$

such that an F1 string (which couples to $B_2$) emanates from the defect. This is exactly the outer-automorphism action we expect from field theory.

Now consider the arrangement in figure 12. In this case the F1 string passes through the brane un-changed, there are not enough dimensions to run the same argument as above. This is exactly the invariance of the operator $\mathcal{O}_1^{(v)}$ (the vector Wilson line) under the outer-automorphism.

## 4.5 Example: Generalized charges for duality/triality defects in 4d

In this section we study duality and triality defects which generate non-invertible symmetries that arise from subgroups of $SL(2,\mathbb{Z})$. They provide 0-form symmetries at certain fixed loci under these groups on the conformal manifold of 4d SCFTs. Field theoretically these defects

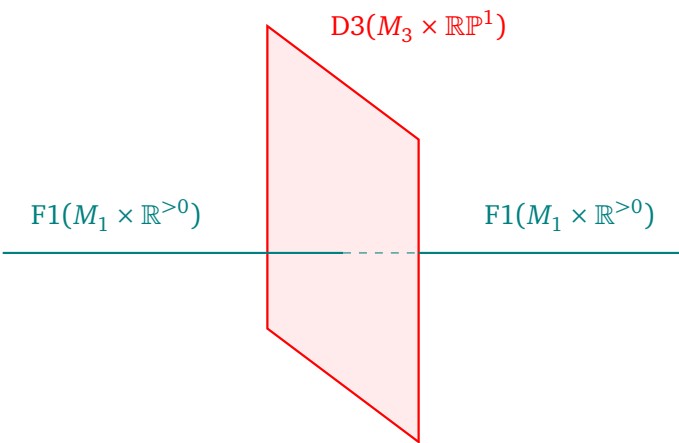

Figure 12: The F1-brane, that realizes the vector Wilson line is invariant under the outer-automorphism.

have been studied in [55, 59, 62, 72, 105, 144]. For the bulk theory we refer to [41, 44], and the realization of the topological defects in term of branes to [4].

In string theory the self-duality or triality symmetries are generated by $[p, q]$-7-branes as first observed in [1] and subsequently studied in detail in [4]. We will exemplify this brane-approach for $\mathcal{N} = 4$ $\mathfrak{su}(N)$ SYM. Moreover, since we have analogously constructed topological defects for the $\mathcal{N} = 2$ Argyres Douglas theory of type $[A_2, D_4]$ via geometric engineering in IIB, the properties highlighted in this section will be valid also for that case.

In this section we will put these defects into the context of the SymTFT and derive the generalized charges (in terms of the topological defects of the SymTFT) realized again in terms of "branes" and Hanany-Witten transitions among them.

The $(p, q)$-strings give rise to topological defects in the SymTFT, which depending on the $\mathcal{B}^{\text{sym}}$ boundary conditions give rise to either topological defects that generate the 1-form symmetry or to the line operators, i.e. generalized charges.

The Hanany-Witten effect between $(p, q)$-strings and $[p, q]$-7-branes encodes whether the resulting non-invertible symmetry is gauge equivalent to an invertible symmetry or not. In terms of the SymTFT couplings this was analyzed in [41, 44]. This allows the distinction between intrinsic and non-intrinsic non-invertible symmetries – if one wishes to use this formulation. More categorically, the SymTFT is either the same as for an invertible (i.e. higher group) symmetry or not.

We will focus on generalized charges via the Hanany-Witten effect between $(p, q)$-strings and $[p, q]$-7-branes. In particular, from the brane realization and the Hanany-Witten phenomenon we will be able to provide a diagnostic for intrinsic versus non-intrinsic non-invertible symmetries, even beyond $\mathfrak{su}(p)$ with $p$ prime.

Duality and triality defects for $\mathcal{N} = 4$ SYM arise for fixed values of $\tau = e^{i\pi/2}, e^{i\pi/3}$, respectively, i.e. the values that are invariant under $\mathbb{Z}_4$ or $\mathbb{Z}_6$ subgroups of $SL(2, \mathbb{Z})$ [4, 44, 59, 62, 72]. Our convention for the monodromy matrices labelled by $(p, q)$ charges are

$$M_{p,q} = \begin{pmatrix} pq + 1 & p^2 \\ -q^2 & 1 - pq \end{pmatrix}, \tag{187}$$

and we take the basis

$$a = M_{1,0} = \begin{pmatrix} 1 & 1 \\ 0 & 1 \end{pmatrix}, \qquad b = M_{1,1} = \begin{pmatrix} 2 & 1 \\ -1 & 0 \end{pmatrix}, \qquad c = M_{1,-1} = \begin{pmatrix} 0 & 1 \\ -1 & 2 \end{pmatrix}. \tag{188}$$

We summarize the fixed values of $\tau$ and associated monodromy matrices in table 4.[17]

---

[17]We use the conventions of [145], but act on tau from the right as to give rise to the canonical choice of fixed

Table 4: The Kodaira singularities, associated constant values of $\tau$, the monodromy group and the monodromy matrix $M$.

| Kodaira Type | $\tau$ | $G$ | Monodromy Matrix $M$ |
|---|---|---|---|
| $II$ | $e^{\pi i/3}$ | $\mathbb{Z}_6$ | $ab = \begin{pmatrix} 1 & 1 \\ -1 & 0 \end{pmatrix}$ |
| $II^*$ | $e^{\pi i/3}$ | $\mathbb{Z}_6$ | $a^6bcba = \begin{pmatrix} 0 & -1 \\ 1 & 1 \end{pmatrix}$ |
| $III$ | $e^{\pi i/2}$ | $\mathbb{Z}_4$ | $a^2b = \begin{pmatrix} 0 & 1 \\ -1 & 0 \end{pmatrix}$ |
| $III^*$ | $e^{\pi i/2}$ | $\mathbb{Z}_4$ | $a^6bcb = \begin{pmatrix} 0 & -1 \\ 1 & 0 \end{pmatrix}$ |
| $IV$ | $e^{\pi i/3}$ | $\mathbb{Z}_3$ | $a^2ba = \begin{pmatrix} 0 & 1 \\ -1 & -1 \end{pmatrix}$ |
| $IV^*$ | $e^{\pi i/3}$ | $\mathbb{Z}_3$ | $a^5bcb = \begin{pmatrix} -1 & -1 \\ 1 & 0 \end{pmatrix}$ |
| $I_0^*$ | $\tau$ | $\mathbb{Z}_2$ | $a^4bc = \begin{pmatrix} -1 & 0 \\ 0 & -1 \end{pmatrix}$ |

### 4.5.1 Hanany-Witten Setups with $[p,q]$-7-branes and $(r,s)$-strings

We now describe in more detail the specific Hanany-Witten configuration already introduced in (164) that is relevant for this example. Let us consider the original Hanany-Witten brane configuration, consisting of an NS5-brane extended along $x^{0,1,2,3,4,5}$ and a D5-brane extended along $x^{0,1,2,6,7,8}$; when these are moved past each other, a D3-brane extended along $x^{0,1,2,9}$ is created. By applying T-duality in the $x^{1,2}$ directions, followed by an S-duality $S$ transformation, followed by T-duality in the $x^{6,7}$ directions, we reach a Hanany-Witten setup with a D7-brane extended along $x^{0,\dots,7}$ and a D1-brane extended along $x^{0,8}$. When these are moved past each other, an F1-string extended along $x^{0,9}$ is created. This configuration conserves both the linking number between the D7-brane and the D1-brane, and the $(r,s)$-string charge of the system. The latter observation stems from the relation

$$(1\ \ 0)M_{1,0} = (1\ \ 0) + (0\ \ 1). \tag{189}$$

In our conventions the charges of an $(r,s)$-string are collected in the row vector $(s\ \ r)$. Thus, in the above relation, $(1\ \ 0)$ represents the D1-brane, $(0\ \ 1)$ the F1-string, while $M_{1,0}$ is the monodromy matrix of the D7-brane (see figure 13).

The generalization of (189) is the identity

$$(s\ \ r)M_{p,q} = (s\ \ r) + n(q\ \ p), \qquad n := ps - qr. \tag{190}$$

We interpret this relation as follows. Start with a configuration with a $[p,q]$-7-brane extended along $x^{0,\dots,7}$, and an $(r,s)$-string extended along $x^{0,8}$. If we move the $(r,s)$-string across the $[p,q]$-7-brane, $n$ copies of a $(p,q)$-string are generated, extended along $x^{0,9}$. If $n$ is negative, this is understood as $|n|$ copies of a $(-p,-q)$-string. In the special case $n = 0$, the $(r,s)$-string and the $[p,q]$-7-brane are mutually local and the $(r,s)$-string can end on the $[p,q]$-7-brane; there is no Hanany-Witten brane creation effect if these two objects are passed across each other.

---

values of $\tau$ as e.g. in [146].

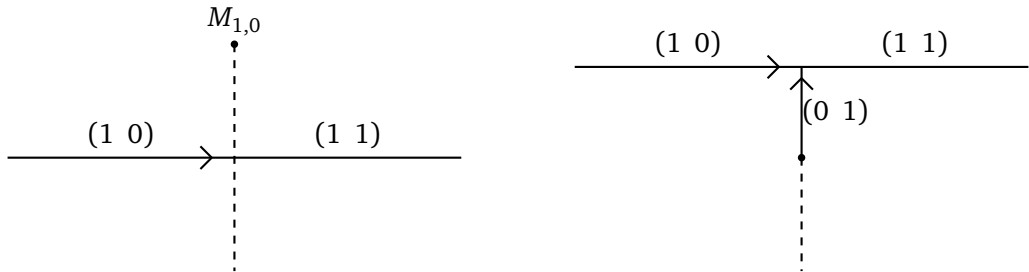

Figure 13: Conservation of charge during a Hanany-Witten move involving a D1 and D7 brane. On the left hand side; passing the D1 through the monodromy cut for the D7 brane modifies the charge from $(1,0) \to (1,1)$. On the right hand side; sliding the configuration off the cut must preserve charge, meaning an F1 string is created.

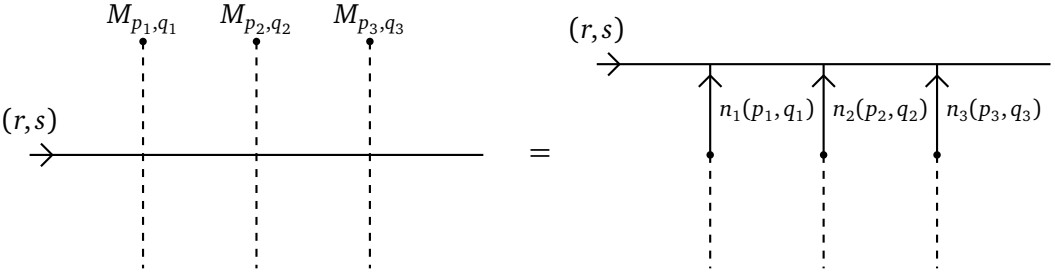

Figure 14: Hanany-Witten transitions for a general 7-brane configuration with no fixed $[p, q]$ charge as usually appear in F-theory and $(r, s)$-strings.

It is important to study the generalized charges, i.e. SymTFT topological defects, coming from $(r, s)$-strings and the 7-branes with monodromy $M$, which we take parallel to the boundary. In the spirit of section 4.3, instead of being directly related to a mixed 't Hooft anomaly, it has a SymTFT that is a DW theory with twisted cocyles. Let us now consider a 7-brane with monodromy matrix $M$, written as a product $M = M_{p_1, q_1} M_{p_2, q_2} M_{p_3, q_3} \dots$. A repeated application of the basic Hanany-Witten move encoded in (190) yields the configuration depicted in the figure, where the multiplicities $n_1, n_2, n_3, \dots$, of the created strings are determined by the charges of the $(r, s)$-string and by the $[p_k, q_k]$-7-brane labels,

$$n_1 = p_1 s - q_1 r, \tag{191}$$

$$n_2 = p_2(s + n_1 q_1) - q_2(r + n_1 p_1), \tag{192}$$

$$n_3 = p_3(s + n_1 q_1 + n_2 q_2) - q_3(r + n_1 p_1 + n_2 p_1), \tag{193}$$

and so on. In general due to $M = M_{p_1, q_1} M_{p_2, q_2} M_{p_3, q_3} \dots$, we will not have a single string creation event. If we have multiple string creation events, even modulo $N$, it signals that something more general than a mixed 't Hooft anomaly is at play. This indeed generically corresponds to a twisted cocyle in the SymTFT.

**Examples.** Let us consider the Kodaira type $IV^*$ monodromy matrix $M = \begin{pmatrix} -1 & -1 \\ 1 & 0 \end{pmatrix}$. We use the decomposition $M = a^5 bcb$, $a = M_{1,0}$, $b = M_{1,1}$, $c = M_{1,-1}$. The multiplicities $n_1, \dots, n_8$ of the created strings are

$$(n_1, \dots, n_8) = (s, s, s, s, s, -r - 4s, -r - 2s, r). \tag{194}$$

Alternatively we can use the decomposition $M = A^5 BC^2$, $A = M_{1,0}$, $B = M_{3,1}$, $C = M_{1,1}$. In this case the multiplicities are

$$(n_1, \dots, n_8) = (s, s, s, s, s, -r - 2s, r, r). \tag{195}$$

Next, let us consider the Kodaira type $IV$ with $M = \begin{pmatrix} 0 & 1 \\ -1 & -1 \end{pmatrix}$. We can write $M = a^2 ba$. The four multiplicities are

$$(n_1, \ldots, n_4) = (s, s, -r-s, -r). \tag{196}$$

In passing we note that it is not possible to write $M = A^x B^y C^z$ with non-negative $x$, $y$, $z$.

If we consider again Kodaira Type $IV$, but we work modulo $N = 3$, we can write $M = M_{1,2} = M_{1,1} = c \mod 3$. In this case there is a single event of string creation, with multiplicity

$$n_1 = r + s \mod 3. \tag{197}$$

In this case the HW transition is related to a mixed 't Hooft anomaly between self-triality and the 1-form symmetry.

### 4.5.2 Intrinsic vs. Non-Intrinsic

Let us recall the notion of intrinsic non-invertible symmetry [62]. Suppose $\mathcal{T}$ is a QFT that admits a non-invertible symmetry. We say that the non-invertible symmetry is of non-intrinsic type if $\mathcal{T}$ can be connected by gauging of a global symmetry to a QFT $\mathcal{T}'$ that only admits invertible symmetries (i.e. higher-form or higher-group symmetries). We say that the non-invertible symmetry of $\mathcal{T}$ is of intrinsic type if such $\mathcal{T}'$ does not exist.

**Global Forms of SYM.** The global variants of 4d $\mathcal{N} = 4$ $\mathfrak{su}(N)$ SYM are $SU(N)$ and $(SU(N)/\mathbb{Z}_k)_n$, where $k \neq 1$ is a divisor of $N$ and $n = 0, 1, \ldots, k-1$ [147]. Global variants can be acted upon by topological manipulations (gauging 1-form symmetry, stacking with SPT). They all form a unique orbit under topological manipulations. Indeed, we can start from $SU(N)$ and reach $(SU(N)/\mathbb{Z}_k)_n$ by selecting a $\mathbb{Z}_k$ subgroup of the $\mathbb{Z}_N$ center 1-form symmetry of $SU(N)$, and gauging it with a discrete torsion given by $n$.[18] Let $L_\mathcal{T}$ denote the set of line operators of the global variant $\mathcal{T}$. Explicitly [147]

$$\begin{aligned} \mathcal{T} &= SU(N), & L_\mathcal{T} &= \{a(1,0) \mod N, a \in \mathbb{Z}\}, \\ \mathcal{T} &= (SU(N)/\mathbb{Z}_k)_n, & L_\mathcal{T} &= \{a(n, N/k) + b(k, 0) \mod N, a, b \in \mathbb{Z}\}. \end{aligned} \tag{198}$$

In terms of topological defects of the symmetry TFT and their brane realization, these are provided by the full set of $(p, q)$-strings that can end on the boundary depending on the choice of the $\mathcal{B}^{\text{sym}}$ boundary conditions.

We want to study when a duality/triality symmetry associated to one of the monodromy matrices of table 4 is of intrinsic/non-intrinsic type, depending on $N$. This is equivalent to asking: for a given $N$ and a given monodromy matrix, is there a global variant in which the associated duality/triality defect acts invertibly on all line operators?

**Hanany-Witten Diagnostic of Intrinsicality.** This question can be addressed in terms of Hanany-Witten moves, as follows. The duality/triality defect specified by the monodromy matrix $M$ acts **invertibly on all lines** of the global variant $\mathcal{T}$ if the following condition holds,[19]

$$(M - \mathbb{I}_{2 \times 2}) \cdot (r, s) \in L_\mathcal{T}, \qquad \text{for all } (r, s) \in L_\mathcal{T}. \tag{199}$$

The quantity $(M - \mathbb{I}_{2 \times 2}) \cdot (r, s)$ is the total $(p, q)$-string charge created, when a line operator with charges $(r, s)$ crosses the 7-brane implementing the duality/triality defect. We demand

---

[18]In contrast, the set of global forms can split into disjoint non-empty orbits under the action of the $SL(2, \mathbb{Z})$ duality group, depending on $N$ [127, 130, 147].

[19]The notation $(r', s') = M \cdot (r, s)$ stands for the matrix equation $(s'\ r') = (s\ r)M$ in our conventions.

that the total $(p,q)$-string charge that is created can be written as a combination of the same charges as those of the lines in $L_{\mathcal{T}}$. This is because, in the global variant $\mathcal{T}$, a string with those charges, projected parallel to the boundary, yields the trivial surface defect. As a result, we are guaranteed an invertible action on all line operators of $\mathcal{T}$, as desired.

The condition (199) can be analyzed explicitly for each of the monodromy matrices in table 4, for some small values of $N$. We report the results of our analysis in table 5. For each monodromy matrix and $N$, we indicate the global variant(s) that satisfy (199); if none is found, the duality/triality symmetry is intrinsically non-invertible. The fact that the global variants indicated in the table are invariant under $S$ or $ST$ can also be checked directly making use of [147]

$$
\begin{aligned}
T &: (SU(N)/\mathbb{Z}_k)_n \rightarrow (SU(N)/\mathbb{Z}_k)_{n+N/k}, \\
S &: (SU(N)/\mathbb{Z}_k)_0 \rightarrow (SU(N)/\mathbb{Z}_{N/k})_0, \\
&\quad\,\, (SU(N)/\mathbb{Z}_k)_n \rightarrow (SU(N)/\mathbb{Z}_{k^*})_{n^*}, \qquad (n \neq 0).
\end{aligned}
\tag{200}
$$

In the above relations we let $k$ be any divisor of $N$, including $k=1$ and $k=N$. The label $n$ is understood mod $k$ (if $k=1$, then $n=0$; this is the $SU(N)$ variant). The new labels $k^*$, $n^*$ are given by

$$
k^* = \frac{N}{\gcd(n,k)}, \qquad \alpha k + \beta n = \gcd(n,k), \qquad n^* = -\beta \frac{N}{k} \mod k^*.
\tag{201}
$$

The global forms under the second column in table 5 are invariant under the action of $S$, followed by $T$, in our conventions.

For $N$ a prime number, we reproduce the results of [72]. This can also be seen from table 5 and algebraically as follows. If $N=p$ is prime, the global variants are $SU(p)$, $PSU(p)_n$, $n = 0, 1, \ldots, p-1$. For each of them, the corresponding set of lines $L_{\mathcal{T}}$ consists of multiples of a single line: $(1,0)$ for $SU(p)$ and $(n,1)$ for $PSU(p)_n$. As a result, the condition (199) boils down to the eigenvalue problem

$$
M \cdot (r,s) = \lambda(r,s) \mod p,
\tag{202}
$$

where $(r,s) = (1,0)$ for $\mathcal{T} = SU(p)$ and $(r,s) = (n,1)$ for $\mathcal{T} = PSU(p)_n$. In fact, as soon as (202) admits a non-trivial solution $(r,s)$, the latter can be identified as the line generating $L_{\mathcal{T}}$ for one of the global variants $\mathcal{T}$. Thus, for $N=p$ prime, (202) is a necessary and sufficient condition for finding a global variant $\mathcal{T}$ in which the duality/triality defects acts invertibly on all lines.

## 5 Conclusions and outlook

We constructed the SymTFT for QFTs realized either holographically or in geometric engineering, in terms of branes. The main results are as follows: in section 3 we demonstrated that branes encode the topological couplings of the SymTFT, and in section 4 we highlighted that our proposal also incorporates a notion of generalized charges via the Hanany-Witten effect.

Whilst presenting a general framework in both cases, we gave evidence for our proposal in various geometric and holographic examples, including 4d SYM theories.

In section 3.9 we use our general approach to give a brane origin to the symmetry generators in the 4d $\mathcal{N} = 2$ $[A_2, D_4]$ SCFT and in section 4.5 we use the generalized charge/ Hanany-Witten relationship to propose a sharp criterion to distinguish intrinsic and non-intrinsic non-invertible symmetries, for rank beyond $\mathfrak{su}(p = \text{prime})$.

Studying properties of topological symmetry generators from the perspective of branes is a new and exciting area of research. It would be interesting to apply our general approach

Table 5: For each monodromy matrix and each $N$, we indicate the global variant(s) on which the associated duality/triality defect acts invertibly on all line operators. If no such global variant exists, the non-invertible symmetry is of intrinsic type. In labeling global variants, we do not keep track of background fields and their counterterms. We do not include the monodromy $S^2 = I_0^*$ because every global variant is invariant under $S^2$.

| $N$ | $S = III$ | $ST = IV, (ST)^2 = IV^*, S^2(ST) = II^*, S^2(ST)^2 = II$ |
|---|---|---|
| 2 | $PSU(2)_1$ | intrinsic |
| 3 | intrinsic | $PSU(3)_2$ |
| 4 | $(SU(4)/\mathbb{Z}_2)_0$ | $(SU(4)/\mathbb{Z}_2)_0$ |
| 5 | $PSU(5)_{2 \text{ or } 3}$ | intrinsic |
| 6 | intrinsic | intrinsic |
| 7 | intrinsic | $PSU(7)_{3 \text{ or } 5}$ |
| 8 | $(SU(8)/\mathbb{Z}_4)_2$ | intrinsic |
| 9 | $(SU(9)/\mathbb{Z}_3)_0$ | $(SU(9)/\mathbb{Z}_3)_0$ |
| 10 | $PSU(10)_{3 \text{ or } 7}$ | intrinsic |
| 11 | intrinsic | intrinsic |
| 12 | intrinsic | $(SU(12)/\mathbb{Z}_6)_4$ |
| 13 | $PSU(13)_{5 \text{ or } 8}$ | $PSU(13)_{4 \text{ or } 10}$ |
| 14 | intrinsic | intrinsic |
| 15 | intrinsic | intrinsic |
| 16 | $(SU(16)/\mathbb{Z}_4)_0$ | $(SU(16)/\mathbb{Z}_4)_0$ |
| 17 | $PSU(17)_{4 \text{ or } 13}$ | intrinsic |
| 18 | $(SU(18)/\mathbb{Z}_6)_3$ | intrinsic |
| 19 | intrinsic | $PSU(19)_{8 \text{ or } 12}$ |
| 20 | $(SU(20)/\mathbb{Z}_{10})_{4 \text{ or } 6}$ | intrinsic |
| 21 | intrinsic | $PSU(21)_{5 \text{ or } 17}$ |
| 22 | intrinsic | intrinsic |
| 23 | intrinsic | intrinsic |
| 24 | intrinsic | intrinsic |
| 25 | $(SU(25)/\mathbb{Z}_5)_0, PSU(25)_{7 \text{ or } 18}$ | $(SU(25)/\mathbb{Z}_5)_0$ |
| 26 | $PSU(26)_{5 \text{ or } 21}$ | intrinsic |
| 27 | intrinsic | $(SU(27)/\mathbb{Z}_9)_6$ |
| 28 | intrinsic | $(SU(28)/\mathbb{Z}_{14})_{6 \text{ or } 10}$ |
| 29 | $PSU(29)_{12 \text{ or } 17}$ | intrinsic |

to more exotic non-Lagrangian QFTs where the use of standard field theory tools to study generalized symmetries is either obstructed or non-existent.

The study of generalized charges is another interesting avenue to pursue. In this work we demonstrated that the Hanany-Witten effect encodes the case where a non-invertible $p$ symmetry acts on extended operators of dimension $q = p+1$. It would be interesting to explore the full suite of generalized charges for invertible symmetry and non-invertible symmetries, e.g. understanding symmetry fractionalization from a brane perspective, as well as generalized charges for genuine and non-genuine operators, see [46]. The brane-perspective will be key to studying theories at strong coupling and in holographic settings.

# Acknowledgements

We are grateful to Ibou Bah, Lakshya Bhardwaj, Lea Bottini, Noppadol Mekareeya for discussions. We are grateful to the Simons Center for Geometry and Physics and the KITP Santa Barbara for hospitality while this work was in preparation.

*A paper by Ibou Bah, Enoch Leung and Thomas Waddleton will appear with some related, but complementary, content. We thank these authors for coordination.*

**Funding information** We acknowledge support through the Simons Foundation Collaboration on "Special Holonomy in Geometry, Analysis, and Physics", Award ID: 724073, Schafer-Nameki. The work of FB is supported by the Simons Collaboration Grant on Global Categorical Symmetries. SSN is in part supported by the EPSRC Open Fellowship EP/X01276X/1.

# A Effective actions with fluxes and branes

## A.1 Type II effective actions

The bosonic terms in the type IIA and type IIB low-energy effective actions, written in string frame, read (see *e.g.* [145])[20]

$$S_{\text{IIA}} = S_{\text{NSNS}} + \frac{1}{2\kappa_{10}^2} \int_{M_{10}} \left[ -\frac{1}{2} F_2 * F_2 - \frac{1}{2} F_4 * F_4 - \frac{1}{2} F_0 * F_0 \right] + S_{\text{IIA}}^{\text{top}}, \tag{A.1}$$

$$S_{\text{IIB}} = S_{\text{NSNS}} + \frac{1}{2\kappa_{10}^2} \int_{M_{10}} \left[ -\frac{1}{2} F_1 * F_1 - \frac{1}{2} F_3 * F_3 - \frac{1}{4} F_5 * F_5 \right] + S_{\text{IIB}}^{\text{top}}, \tag{A.2}$$

where the bosonic NSNS sector action is given as

$$S_{\text{NSNS}} = \frac{1}{2\kappa_{10}^2} \int_{M_{10}} e^{-2\phi} \left[ R * 1 + 4 d\phi * d\phi - \frac{1}{2} H_3 * H_3 \right]. \tag{A.3}$$

Note, in this appendix we suppress wedge products for notational convenience. The topological terms $S_{\text{IIA}}^{\text{top}}$, $S_{\text{IIB}}^{\text{top}}$ are conveniently described in terms of 11-form monomials in the field strengths,

$$S_{\text{IIA/B}}^{\text{top}} = \frac{1}{2\kappa_{10}^2} \int_{M_{10}} I_{10}^{(0)\text{IIA/B}}, \qquad dI_{10}^{(0)\text{IIA}} = -\frac{1}{2} H_3 F_4 F_4, \qquad dI_{10}^{(0)\text{IIB}} = -\frac{1}{2} F_3 F_5 H_3. \tag{A.4}$$

For our purposes, it is convenient to rescale of the $p$-form potentials, such that their fluxes have integral periods. To this end we use $\frac{1}{2\kappa_{10}^2} = 2\pi \ell_s^{-8}$ and we perform the field redefinitions

$$H_3^{\text{old}} = \ell_s^2 H_3^{\text{new}}, \qquad F_p^{\text{old}} = \ell_s^{p-1} F_p^{\text{new}}, \qquad p = 0, 1, \dots, 5. \tag{A.5}$$

We henceforth drop the label "new". The Bianchi identities read

$$dH_3 = 0, \qquad dF_p = H_3 F_{p-2}, \qquad p = 0, 1, \dots, 5, \tag{A.6}$$

and can be solved by writing (see *e.g.* [118])

$$H_3 = dB_2, \qquad F_p = dC_{p-1} - H_3 C_{p-3} + F_0 (e^{B_2})_p, \qquad p = 1, 2, \dots, 5. \tag{A.7}$$

---

[20]We suppress wedge products of forms for brevity. The Hodge star of a $p$-form $\alpha$ in $d$ dimensions is defined as $(*\alpha)_{\mu_1 \dots \mu_{d-n}} = \frac{1}{n!} \alpha^{\nu_1 \dots \nu_n} \epsilon_{\nu_1 \dots \nu_n \mu_1 \dots \mu_{d-n}}$ with $\epsilon_{012\dots} = \sqrt{-g}$ in mostly plus signature.

The $F_0$ term in the last relation is only present in type IIA. The IIB action is understood as a pseudoaction: the self duality constraint on $F_5$ has to be imposed by hand after varying the pseudoaction.

The equations of motion for the $p$-form fields $B_2$, $C_p$ can be written compactly by introducing Hodge dual field strengths $H_7$, $F_p$, $p = 6, 7, \ldots, 10$. The combined content of the Bianchi identities and equations of motion is then encoded in the relations

$$dH_3 = 0, \qquad dF_p = H_3 F_{p-2}, \qquad dH_7 = \begin{cases} F_0 F_8 - F_2 F_6 + \frac{1}{2}F_4^2 + X_8, & \text{type IIA}, \\ -F_1 F_7 + F_3 F_5, & \text{type IIB}, \end{cases} \tag{A.8}$$

where $p = 0, \ldots, 10$, together with the following Hodge star relations,

$$H_7 = \ell_s^{-4} e^{-2\phi} * H_3, \qquad \begin{aligned} F_6 &= -\ell_s^{-2} * F_4, & F_8 &= \ell_s^{-6} * F_2, & F_{10} &= -\ell_s^{-10} * F_0, \\ F_5 &= *F_5, & F_7 &= -\ell_s^{-4} * F_3, & F_9 &= \ell_s^{-8} * F_1. \end{aligned} \tag{A.9}$$

For now on, we set $\ell_s = 1$ for brevity. Notice that in type IIA we have included the effect of a topological higher-curvature correction,

$$X_8 = \frac{1}{192}\left[ p_1(TM_{10})^2 - 4p_2(TM_{10}) \right], \tag{A.10}$$

where $p_i(TM_{10})$ denote the Pontryagin forms of the tangent bundle to $M_{10}$ [116].

**Topological actions in 11 dimensions.** The full set (A.8) of Bianchi identities can be derived using 11d topological actions (cfr. [47])

$$S_{11}^{\text{IIA}} = \int_{M_{11}} \left[ F_0 dF_{10} - F_2 dF_8 + F_4 dF_6 + H_3 dH_7 - \frac{1}{2}H_3 F_4^2 - H_3 X_8 + H_3 F_2 F_6 - H_3 F_0 F_8 \right],$$

$$S_{11}^{\text{IIB}} = \int_{M_{11}} \left[ F_1 dF_9 - F_3 dF_7 + \frac{1}{2}F_5 dF_5 + H_3 dH_7 + H_3 F_1 F_7 - H_3 F_3 F_5 \right], \tag{A.11}$$

which are regarded as functionals of $H_3$, $H_7$, $F_p$. The 10d Hodge star relations (A.9) do not follow from the 11d topological actions and have to be imposed by hand after variation.

In terms of the general parametrization (41), we have

$$\text{IIA:} \quad F^{(i)} = (F_0, F_2, F_4, F_6, F_8, F_{10}, H_3, H_7), \qquad \kappa_{ij} = \begin{pmatrix} 0 & 0 & 0 & 0 & 0 & 1 \\ 0 & 0 & 0 & 0 & -1 & 0 \\ 0 & 0 & 0 & 1 & 0 & 0 \\ 0 & 0 & -1 & 0 & 0 & 0 \\ 0 & 1 & 0 & 0 & 0 & 0 \\ -1 & 0 & 0 & 0 & 0 & 0 \\ & & & & & & 0 & 1 \\ & & & & & & 1 & 0 \end{pmatrix},$$

$$\text{IIB:} \quad F^{(i)} = (F_1, F_3, F_5, F_7, F_9, H_3, H_7), \qquad \kappa_{ij} = \begin{pmatrix} 0 & 0 & 0 & 0 & 1 \\ 0 & 0 & 0 & -1 & 0 \\ 0 & 0 & 1 & 0 & 0 \\ 0 & -1 & 0 & 0 & 0 \\ 1 & 0 & 0 & 0 & 0 \\ & & & & & 0 & 1 \\ & & & & & 1 & 0 \end{pmatrix}. \tag{A.12}$$

**Compact notation using polyforms.** We may repackage the above results in a compact way by introducing a polyform $\mathbf{F}$ describing all RR field strengths, together with an involution $\mathbf{F} \mapsto \overline{\mathbf{F}}$ which flips some signs,

$$\begin{aligned} \text{IIA:} \quad & \mathbf{F} = F_0 + F_2 + F_4 + F_6 + F_8 + F_{10}, & \overline{\mathbf{F}} &= F_0 - F_2 + F_4 - F_6 + F_8 - F_{10}, \\ \text{IIB:} \quad & \mathbf{F} = F_1 + F_3 + F_5 + F_7 + F_9, & \overline{\mathbf{F}} &= -F_1 + F_3 - F_5 + F_7 - F_9. \end{aligned} \tag{A.13}$$

Our previous 11d topological actions can be written compactly as (cfr. [47])

$$S_{11}^{\text{IIA/B}} = \int_{M_{11}} \left[ -\frac{1}{2}\mathbf{F}d\bar{\mathbf{F}} + H_3 dH_7 - \frac{1}{2}\mathbf{F}H_3\bar{\mathbf{F}} - \delta_{\text{IIA}} H_3 X_8 \right], \tag{A.14}$$

where $\delta_{\text{IIA}}$ means that the $H_3 X_8$ term is only present in type IIA. The 10 Hodge duality relations are

$$H_7 = e^{-2\phi} * H_3, \qquad *\bar{\mathbf{F}} = -\mathbf{F}. \tag{A.15}$$

**Brane sources in type II.** Recall that $J^{(i)}$ is the magnetic source for the flux $F^{(i)}$. The identifications (A.12) imply

$$\text{type IIA:} \quad J^{(i)} = (J_1^{\text{D8}}, J_3^{\text{D6}}, J_5^{\text{D4}}, J_7^{\text{D2}}, J_9^{\text{D0}}, 0, J_4^{\text{NS5}}, J_8^{\text{F1}}),$$
$$\text{type IIB:} \quad J^{(i)} = (J_2^{\text{D7}}, J_4^{\text{D5}}, J_6^{\text{D3}}, J_8^{\text{D1}}, J_{10}^{\text{D(-1)}}, J_4^{\text{NS5}}, J_8^{\text{F1}}). \tag{A.16}$$

The Bianchi identities in the presence of sources read

$$\begin{aligned} dH_3 &= J_4^{\text{NS5}}, \\ dF_p &= J_{p+1}^{\text{D(8-p)}} + H_3 F_{p-2}, \\ dH_7 &= J_8^{\text{F1}} + F_0 F_8 - F_2 F_6 + \frac{1}{2}F_4^2 + X_8, \quad \text{(IIA)} \\ dH_7 &= J_8^{\text{F1}} - F_1 F_7 + F_3 F_5. \quad \text{(IIB)} \end{aligned} \tag{A.17}$$

They imply the following non-closure relations for the magnetic sources,

$$\begin{aligned} dJ_4^{\text{NS5}} &= 0, \\ dJ_p^{\text{D(9-p)}} &= H_3 J_{p-2}^{\text{D(11-p)}} - J_4^{\text{NS5}} F_{p-3}, \\ dJ_8^{\text{F1}} &= -(J_1^{\text{D8}} F_8 + J_9^{\text{D0}} F_0) + (J_3^{\text{D6}} F_6 + J_7^{\text{D2}} F_2) - J_5^{\text{D4}} F_4, \quad \text{(IIA)} \\ dJ_8^{\text{F1}} &= (J_2^{\text{D7}} F_7 - F_1 J_8^{\text{D1}}) - (J_4^{\text{D5}} F_5 - F_3 J_6^{\text{D3}}). \quad \text{(IIB)} \end{aligned} \tag{A.18}$$

These relations encode non-trivial aspects of brane physics. In particular, they furnish a magnetic description of Hanany-Witten brane creation effects. The non-closure of the $J_p^{\text{D(9-p)}}$ current describes a process in which a D$(9-p)$-brane is created upon crossing of a D$(11-p)$-brane and an NS5-brane: this is the Hanany-Witten effect of type (III) described in the main text. By a similar token, the non-closure of $J_8^{\text{F1}}$ captures Hanany-Witten moves in which an F1 is created if a D$p$-brane and D$p'$-brane cross each other ($p + p' = 8$): this is the Hanany-Witten move of type (I).

The relations (A.18) are also compatible with induced brane charges. For simplicity, let us consider a setup without NS5-branes, so that $H_3$ is closed. We use $\delta_p(\text{D}(8-p))$ for the closed delta-function-supported form that describes the insertion of a D$(8-p)$-brane. The total charge $J_p^{\text{D(8-p)}}$ is the sum of $\delta_p(\text{D}(8-p))$ and of terms constructed with lower-dimensional delta-function forms. The latter are associated to higher-dimensional branes with an induced D$(8-p)$-charge. The induced charge originates from world-volume 2-form flux, and tangent/normal bundle contributions, as follows from the Wess-Zumino couplings. For example, in type IIA we may write

$$J_5^{\text{D4}} = \delta_5(\text{D4}) + f_2^{\text{D6}} \delta_3(\text{D6}) + \left[ \frac{1}{2}(f_2^{\text{D6}})^2 + \frac{p_1(N\text{D8}) - p_1(T\text{D8})}{48} \right] \delta_1(\text{D8}),$$

$$J_7^{\text{D2}} = \delta_7(\text{D2}) + f_2^{\text{D4}} \delta_5(\text{D4}) + \left[ \frac{1}{2}(f_2^{\text{D6}})^2 + \frac{p_1(N\text{D6}) - p_1(T\text{D6})}{48} \right] \delta_3(\text{D6}) \tag{A.19}$$

$$+ \left[ \frac{1}{3!}(f_2^{\text{D8}})^3 + f_2^{\text{D8}} \frac{p_1(N\text{D8}) - p_1(T\text{D8})}{48} \right] \delta_1(\text{D8}),$$

where $f_2^{\mathrm{D}p}$ denotes the world-volume flux on a D$p$-brane, which satisfies $df_2^{\mathrm{D}p} = H_3|_{\mathrm{D}p}$, and $p_1(T\mathrm{D}p)$, $p_1(N\mathrm{D}p)$ are the first Pontryagin classes of the tangent and normal bundle to a D$p$-brane. We see that (A.19) is compatible with $dJ_7^{\mathrm{D}4} = H_3 J_5^{\mathrm{D}4}$.

## A.2 M-theory effective action

The bosonic terms in the M-theory low-energy two-derivative effective action are (see *e.g.* [145])

$$\frac{1}{2\kappa_{11}^2} \int_{M_{11}} \left[ R * 1 - \frac{1}{2} G_4 * G_4 - \frac{1}{6} C_3 G_4 G_4 \right], \tag{A.20}$$

where $G_4 = dC_3$. We rescale the 3-form potential,

$$\frac{1}{2\kappa_{11}^2} = 2\pi(2\pi\ell_p)^9, \qquad (2\pi\ell_p)^{-3} C_3^{\mathrm{old}} = C_3^{\mathrm{new}}, \tag{A.21}$$

and drop the label "new" from here on. The action reads

$$S_{\mathrm{M}} = 2\pi \int_{M_{11}} \left[ (2\pi\ell_p)^9 R * 1 - \frac{1}{2}(2\pi\ell_p)^6 G_4 * G_4 - \frac{1}{6} C_3 G_4 G_4 - C_3 X_8 \right]. \tag{A.22}$$

We have included the topological higher-derivative correction $C_3 X_8$, where $X_8$ is as in (A.10) with $TM_{10}$ replaced by $TM_{11}$ [117].

The Bianchi identity and equation of motion for $C_3$ can be written compactly as

$$dG_4 = 0, \qquad dG_7 = \frac{1}{2} G_4 + X_8, \qquad G_7 = -(2\pi\ell_p)^6 * G_4. \tag{A.23}$$

From now on, we set $2\pi\ell_p = 1$ for brevity.

**Topological actions in 12 dimensions.** The relations for $dG_4$, $dG_7$ given in (A.23) can be derived from the following 12d topological action,

$$S_{12}^{\mathrm{M}} = \int_{M_{12}} \left[ G_4 dG_7 - \frac{1}{6} G_4^3 - G_4 X_8 - G_4 J_8^{\mathrm{M}2} - G_7 J_5^{\mathrm{M}5} \right], \tag{A.24}$$

which is regarded as a functional of $G_4$, $G_7$. We have included magnetic sources $J_8^{\mathrm{M}2}$, $J_5^{\mathrm{M}5}$ The 12 action is supplemented with the 11d Hodge duality relation in (A.23).

**Comments on brane sources.** The Bianchi identities with magnetic sources read

$$dG_7 = \frac{1}{2} G_4^2 + X_8 + J_8^{\mathrm{M}2}, \qquad dG_4 = J_5^{\mathrm{M}5}. \tag{A.25}$$

They imply that the currents must satisfy

$$dJ_5^{\mathrm{M}5} = 0, \qquad dJ_8^{\mathrm{M}2} = -G_4 J_5^{\mathrm{M}5}. \tag{A.26}$$

The second relation encodes a Hanany-Witten move in which an M2-brane is created when two M5-branes cross each other: this is the effect of type (IV) in the main text. The above relations are also compatible with the identifications [148–150]

$$J_5^{\mathrm{M}5} = \delta_5(\mathrm{M}5), \qquad J_8^{\mathrm{M}2} = \delta_8(\mathrm{M}2) - h_3^{\mathrm{M}5} \delta_5(\mathrm{M}5), \tag{A.27}$$

where $h_3^{\mathrm{M}5}$ is the field strength of the chiral 2-form living on the M5-brane, which satisfies $h_3^{\mathrm{M}5} = G_4|_{\mathrm{M}5}$.

# B    Non-invertible symmetries acting on line operators

To complement the analysis in the main text using branes, we provide a field theoretic alternative to the derivation of the action of non-invertible 0-form symmetries on line operators in 4d QFTs, using half-space gauging as in [55, 61]. We consider two examples: 4d $\mathcal{N} = 1$ SYM with gauge algebra $\mathfrak{su}(M)$, and 4d pure YM with gauge algebra $\mathfrak{so}(4n)$, which are discussed in the main text in sections 4.2 and 4.4, respectively.

## B.1    4d $\mathcal{N} = 1$ SYM with gauge algebra $\mathfrak{su}(M)$

We want to study the non-invertible 0-form symmetry of the global variant $PSU(M)_0$. To this end, we use the $SU(M)$ global variant as starting point. It has a $\mathbb{Z}_{2M}$ 0-form symmetry (background field: $A_1 \in H^1(W_4; \mathbb{Z}_{2M})$) and a $\mathbb{Z}_M$ 1-form symmetry (background field: $B_2 \in H^2(W_4; \mathbb{Z}_M)$) with mixed anomaly

$$\mathcal{A} = \exp\left(2\pi i \frac{-1}{M} \int_{W_5} A_1 \cup \frac{\mathfrak{P}(B_2)}{2}\right). \tag{B.1}$$

We introduce the usual stacking operation $\tau$ and gauging operation $\sigma$

$$\begin{aligned}
Z_{\tau\mathcal{T}}[B_2] &= Z_{\mathcal{T}}[B_2] e^{2\pi i \frac{1}{M} \int_{W_4} \frac{\mathfrak{P}(B_2)}{2}}, \\
Z_{\sigma\mathcal{T}}[B_2] &= \sum_{b_2 \in H^2(W_4; \mathbb{Z}_M)} Z_{\mathcal{T}}[b_2] e^{2\pi i \frac{1}{M} \int_{W_4} b_2 B_2},
\end{aligned} \tag{B.2}$$

where $\mathcal{T}$ denotes a global variant of 4d $\mathcal{N} = 1$ SYM with gauge algebra $\mathfrak{su}(M)$. For simplicity, throughout this appendix we omit normalization factors in partition functions and we work on a Spin manifold up to gravitational counterterms. We also make use of the compact notation

$$\begin{aligned}
SU(M)_0 &:= SU(M), & SU(M)_p &:= \tau^p SU(M), \\
PSU(M)_{n,0} &:= PSU(M)_n, & PSU(M)_{n,p} &:= \tau^p PSU(M)_n.
\end{aligned} \tag{B.3}$$

One verifies the following identities,

$$\begin{aligned}
(\sigma SU(M)_p)[B_2] &= PSU(M)_{p,0}[B_2], \\
(\sigma PSU(M)_{n,0})[B_2] &= SU(M)_n[-B_2], \\
(\sigma PSU(M)_{n,p})[B_2] &= PSU(M)_{n-p^{-1},-p}[-p^{-1}B_2], & \text{if } p \in \mathbb{Z}_M^\times.
\end{aligned} \tag{B.4}$$

We notice that, for any integer $M \geq 2$, $\pm 1 \in \mathbb{Z}_M^\times$; if $p = \pm 1$, $p^{-1} = \pm 1$. A special case of the last relation is therefore

$$(\sigma PSU(M)_{n,-1})[B_2] = PSU(M)_{n+1,1}[B_2]. \tag{B.5}$$

The anomaly (B.1) implies that (perform a 0-form gauge transformation)

$$Z_{SU(M)}[B_2] = Z_{SU(M)}[B_2] e^{2\pi i \frac{-1}{M} \int_{W_4} \frac{\mathfrak{P}(B_2)}{2}}, \qquad \text{i.e.} \quad SU(M)_0[B_2] = SU(M)_{-1}[B_2]. \tag{B.6}$$

By applying $\tau$ repeatedly on both sides, we get

$$SU(M)_p[B_2] = SU(M)_{p-1}[B_2]. \tag{B.7}$$

We may now apply $\sigma$ on both sides, followed by repeated applications of $\tau$, and get

$$PSU(M)_{n,p}[B_2] = PSU(M)_{n-1,p}[B_2]. \tag{B.8}$$

By combining (B.5) and (B.8), we conclude that

$$(\sigma PSU(M)_{n,-1})[B_2] = PSU(M)_{n,1}[B_2]. \tag{B.9}$$

This can also be written as

$$(\tau^{-1}\sigma\tau^{-1}PSU(M)_{n,0})[B_2] = PSU(M)_{n,0}[B_2]. \tag{B.10}$$

We conclude that the $PSU(M)_{n,0}$ theory is invariant under the combined operation $\tau^{-1}\sigma\tau^{-1}$.

**Half-space gauging and action on lines.** As anticipated above, we want to study the global variant $PSU(M)_{0,0}$. This theory has 't Hooft lines, charged under a magnetic 1-form symmetry. Let us now use $C_2$ for the associated background field, and $D_2^{(k)}(M_2)$ for the topological defects implementing the symmetry. We can describe $D_2^{(k)}(M_2)$ explicitly if we think of $PSU(M)_{0,0}$ as originating from gauging of $SU(M)_0$,

$$Z_{PSU(M)_{0,0}}[C_2] = \sum_{b_2} Z_{SU(M)_0}[b_2] e^{2\pi i \frac{1}{M} \int_{W_4} b_2 C_2}, \qquad D_2^{(k)}(M_2) = e^{2\pi i \frac{k}{M} \int_{M_2} b_2}. \tag{B.11}$$

We observed above that $PSU(M)_{0,0}$ is invariant under $\tau^{-1}\sigma\tau^{-1}$. We can therefore perform this operation in the half-space region $x > 0$, sschematically

$$\begin{aligned} x < 0 &: \quad Z_{PSU(M)_{0,0}}[C_2], \\ x > 0 &: \quad e^{2\pi i \frac{-1}{M} \int \frac{\mathfrak{P}(C_2)}{2}} \sum_{c_2} Z_{PSU(M)_{0,0}}[c_2] e^{2\pi i \frac{-1}{M} \int \frac{\mathfrak{P}(c_2)}{2}} e^{2\pi i \frac{1}{M} \int c_2 C_2}. \end{aligned} \tag{B.12}$$

We impose Dirichlet boundary conditions for $c_2$ at $x = 0$. The locus $x = 0$ realizes the topological operators implementing the non-invertible 0-form symmetry of the $PSU(M)_{0,0}$ theory. Next, let $\mathbf{H}(\gamma)$ denote a 't Hooft line of minimal charge, supported on a contractible loop $\gamma$ bounded by a disk $D$. In the region $x < 0$, $\mathbf{H}(\gamma)$ is a genuine line operator, but it is not invariant under gauge transformations of the magnetic 1-form symmetry background $C_2$. The gauge invariant combination is

$$x < 0 : \quad \mathbf{H}(\gamma) e^{-2\pi i \frac{1}{M} \int_D C_2}. \tag{B.13}$$

The analog of this quantity in the $x > 0$ region is written with $c_2$, as opposed to $C_2$,

$$x > 0 : \quad \mathbf{H}(\gamma) e^{-2\pi i \frac{1}{M} \int_D c_2}. \tag{B.14}$$

Let us recast the theory in the region $x > 0$ in terms of $SU(M)_0$,

$$x > 0 : \quad \sum_{b_2, c_2} Z_{SU(M)_0}[b_2] e^{2\pi i \frac{1}{M} \int \left[ b_2 c_2 - \frac{\mathfrak{P}(c_2)}{2} + c_2 C_2 - \frac{\mathfrak{P}(c_2)}{2} \right]}. \tag{B.15}$$

Upon varying $c_2$ in the exponent, we get the following on-shell relation in the $x > 0$ region,

$$c_2 = b_2 + C_2. \tag{B.16}$$

If we use this in (B.35), we obtain

$$x > 0 : \quad \mathbf{H}(\gamma) e^{-2\pi i \frac{1}{M} \int_D b_2} e^{-2\pi i \frac{1}{M} \int_D C_2} = \mathbf{H}(\gamma) D_2^{(-1)}(D) e^{-2\pi i \frac{1}{M} \int_D C_2}. \tag{B.17}$$

We conclude that the non-invertible defects of the $PSU(M)_{0,0}$ theory act on the minimal-charge 't Hooft line by attaching a 1-form symmetry surface defect to the line. The additional $C_2$ contribution is a $c$-number that drops away if we turn off the $C_2$ background field.

We can also rephrase the argument above in the continuum formulation. The continuum counterpart of (B.15) contains the following topological action,

$$\int \mathcal{D}b_2 \mathcal{D}\beta_1 \mathcal{D}c_2 \mathcal{D}\gamma_1 \exp 2\pi i \int_{W_4} \left[ M b_2 d\beta_1 + M c_2 d\gamma_1 + M b_2 c_2 - \frac{M}{2} c_2 c_2 + M c_2 C_2 - \frac{M}{2} C_2 C_2 \right]. \tag{B.18}$$

The quantities $b_2, \beta_1, c_2, \gamma_1$ are $p$-form gauge fields whose field strengths have integral periods, while $C_2$ is a closed 2-form with integral periods. In the simpler case in which $C_2$ is turned off, the gauge transformations are

$$b_2' = b_2 + d\lambda_1, \quad \beta_1' = \beta_1 + d\lambda_0 + \mu_1, \quad c_2' = c_2 + d\mu_1, \quad \gamma_1' = \gamma_1 + d\mu_0 - \lambda_1 - \mu_1. \tag{B.19}$$

The BF pair $b_2, \beta_1$ couples to the $SU(M)_0$ theory, while $c_2$ and $\gamma_1$ only enter via the topological terms spelled out above. The equations of motion for $\gamma_1, c_2$ read

$$M dc_2 = 0, \qquad M c_2 = M(d\gamma_1 + b_2 + C_2). \tag{B.20}$$

In the normalization relevant for the continuum formulation, the gauge invariant combination in the $x > 0$ region is

$$\mathbf{H}(\gamma) e^{-2\pi i \int_D c_2}, \tag{B.21}$$

while the topological defect implementing the magnetic 1-form symmetry of $PSU(M)_{0,0}$ is

$$D_2^{(k)}(M_2) = e^{2\pi i k \int_{M_2} b_2} = e^{2\pi i k \int_{M_2} (b_2 + d\gamma_1)}. \tag{B.22}$$

In the second step we have observed that $d\gamma_1$ is a globally defined 2-form with integral periods. We can thus add it in the exponent without affecting the result. We thus see that the continuum formulation confirms (B.17).

## B.2 4d pure YM with gauge algebra $\mathfrak{so}(4n)$

We are interested in studying the non-invertible 0-form symmetry of the global variant $Sc(4n)$. We find it convenient to adopt the $SO(4n)$ variant as our starting point. It has a $\mathbb{Z}_2$ 0-form symmetry and a $\mathbb{Z}_2 \times \mathbb{Z}_2$ 1-form symmetry. We denote the corresponding background fields as $A_1 \in H^1(W_4; \mathbb{Z}_2)$ and $B_2, C_2 \in H^2(W_4; \mathbb{Z}_2)$. The theory has the mixed anomaly

$$\mathcal{A} = \exp 2\pi i \frac{1}{2} \int_{W_5} A_1 \cup B_2 \cup C_2. \tag{B.23}$$

The anomaly implies the following relation,

$$Z_{SO(4n)}[B_2, C_2] = Z_{SO(4n)}[B_2, C_2] e^{2\pi i \frac{1}{2} \int_{W_4} B_2 C_2}. \tag{B.24}$$

The theory $Sc(4n)$ is obtained by gauging both $B_2$ and $C_2$.

Given any theory $\mathcal{T}$ coupled to two background fields $\widehat{B}_2, \widehat{C}_2 \in H^2(W_4; \mathbb{Z}_2)$, we define the following three operations:

$$Z_{\tau \mathcal{T}}[\widehat{B}_2, \widehat{C}_2] = Z_{\mathcal{T}}[\widehat{B}_2, \widehat{C}_2] e^{2\pi i \frac{1}{2} \int_{W_4} \widehat{B}_2 \widehat{C}_2},$$

$$Z_{\sigma \mathcal{T}}[\widehat{B}_2, \widehat{C}_2] = \sum_{\widetilde{B}_2, \widetilde{C}_2} Z_{\mathcal{T}}[\widetilde{B}_2, \widetilde{C}_2] e^{2\pi i \frac{1}{2} \int_{W_4} (\widetilde{B}_2 \widehat{B}_2 + \widetilde{C}_2 \widehat{C}_2)},$$

$$Z_{K\mathcal{T}}[\widehat{B}_2, \widehat{C}_2] = Z_{\mathcal{T}}[\widehat{C}_2, \widehat{B}_2]. \tag{B.25}$$

Making use of the anomaly relation (B.24), one may then verify the identity

$$(K\tau\sigma\tau Sc(4n))[\widehat{B}_2, \widehat{C}_2] = (Sc(4n))[\widehat{B}_2, \widehat{C}_2]. \tag{B.26}$$

Indeed, we have (always up to prefactors and gravitational counterterms, working on a Spin manifold)

$$
\begin{aligned}
Z_{K\tau\sigma\tau Sc(4n)}[\widehat{B}_2, \widehat{C}_2] &= Z_{\tau\sigma\tau Sc(4n)}[\widehat{C}_2, \widehat{B}_2] = Z_{\sigma\tau Sc(4n)}[\widehat{C}_2, \widehat{B}_2] e^{2\pi i \frac{1}{2}\int_{W_4} \widehat{B}_2 \widehat{C}_2} \\
&= \sum_{\widetilde{B}_2, \widetilde{C}_2} Z_{\tau Sc(4n)}[\widetilde{B}_2, \widetilde{C}_2] e^{2\pi i \frac{1}{2}\int_{W_4}(\widetilde{B}_2 \widehat{C}_2 + \widetilde{C}_2 \widehat{B}_2)} e^{2\pi i \frac{1}{2}\int_{W_4} \widehat{B}_2 \widehat{C}_2} \\
&= \sum_{\widetilde{B}_2, \widetilde{C}_2} Z_{Sc(4n)}[\widetilde{B}_2, \widetilde{C}_2] e^{2\pi i \frac{1}{2}\int_{W_4}(\widetilde{B}_2 \widetilde{C}_2 + \widetilde{B}_2 \widehat{C}_2 + \widetilde{C}_2 \widehat{B}_2 + \widehat{B}_2 \widehat{C}_2)} \\
&= \sum_{\widetilde{B}_2, \widetilde{C}_2, B_2, C_2} Z_{SO(4n)}[B_2, C_2] e^{2\pi i \frac{1}{2}\int_{W_4}(B_2 \widetilde{B}_2 + C_2 \widetilde{C}_2 + \widetilde{B}_2 \widetilde{C}_2 + \widetilde{B}_2 \widehat{C}_2 + \widetilde{C}_2 \widehat{B}_2 + \widehat{B}_2 \widehat{C}_2)}.
\end{aligned} \tag{B.27}
$$

To proceed we perform the redefinitions

$$\widetilde{B}_2 \to \widetilde{B}_2 + \widehat{B}_2 + C_2, \qquad \widetilde{C}_2 \to \widetilde{C}_2 + \widehat{C}_2 + B_2. \tag{B.28}$$

We get

$$
\begin{aligned}
Z_{K\tau\sigma\tau Sc(4n)}[\widehat{B}_2, \widehat{C}_2] &= \sum_{\widetilde{B}_2, \widetilde{C}_2, B_2, C_2} Z_{SO(4n)}[B_2, C_2] e^{2\pi i \frac{1}{2}\int_{W_4}(B_2 C_2 + B_2 \widehat{B}_2 + C_2 \widehat{C}_2 + \widetilde{B}_2 \widetilde{C}_2)} \\
&= \sum_{B_2, C_2} Z_{SO(4n)}[B_2, C_2] e^{2\pi i \frac{1}{2}\int_{W_4}(B_2 C_2 + B_2 \widehat{B}_2 + C_2 \widehat{C}_2)}.
\end{aligned} \tag{B.29}
$$

Now we make use of the anomaly relation (B.24) inside the sum,

$$
\begin{aligned}
Z_{K\tau\sigma\tau Sc(4n)}[\widehat{B}_2, \widehat{C}_2] &= \sum_{B_2, C_2} Z_{SO(4n)}[B_2, C_2] e^{2\pi i \frac{1}{2}\int_{W_4}(B_2 \widehat{B}_2 + C_2 \widehat{C}_2)} \\
&= Z_{Sc(4n)}[\widehat{B}_2, \widehat{C}_2],
\end{aligned} \tag{B.30}
$$

as claimed above.

**Half-space gauging and action on lines.** Let us regard the $Sc(4n)$ theory as coming from gauging the $SO(4n)$ theory. This allows us to write the topological defects generating the 1-form symmetries of the $Sc(4n)$ in terms of discrete gauge fields. More precisely,

$$Z_{Sc(4n)}[\widehat{B}_2, \widehat{C}_2] = \sum_{B_2, C_2} Z_{SO(4n)}[B_2, C_2] e^{2\pi i \frac{1}{2}\int_{W_4}(B_2 \widehat{B}_2 + C_2 \widehat{C}_2)}, \tag{B.31}$$

where we identify

$$
\begin{aligned}
\widehat{B}_2 &\leftrightarrow D_2^{(\widehat{B})}(M_2) = e^{2\pi i \frac{1}{2}\int_{M_2} B_2}, \\
\widehat{C}_2 &\leftrightarrow D_2^{(\widehat{C})}(M_2) = e^{2\pi i \frac{1}{2}\int_{M_2} C_2}.
\end{aligned} \tag{B.32}
$$

We consider a half-space gauging configuration, in which the region $x < 0$ has the $Sc(4n)$ theory, and the region $x > 0$ the $K\tau\sigma\tau Sc(4n)$ theory,

$$
\begin{aligned}
x < 0 &: \quad Z_{Sc(4n)}[\widehat{B}_2, \widehat{C}_2], \\
x > 0 &: \quad \sum_{\widetilde{B}_2, \widetilde{C}_2} Z_{Sc(4n)}[\widetilde{B}_2, \widetilde{C}_2] e^{2\pi i \frac{1}{2}\int_{W_4}(\widetilde{B}_2 \widetilde{C}_2 + \widetilde{B}_2 \widehat{C}_2 + \widetilde{C}_2 \widehat{B}_2 + \widehat{B}_2 \widehat{C}_2)}.
\end{aligned} \tag{B.33}
$$

We impose Dirichlet boundary conditions for $\widetilde{B}_2$, $\widetilde{C}_2$ at $x = 0$. The locus $x = 0$ realizes the codimension-1 topological defects generating the non-invertible symmetry of the $Sc(4n)$ theory.

In the $Sc(4n)$ theory we have line operators of charges $(1,0)$ and $(0,1)$ under the 1-form symmetries generated by $D_2^{(\widehat{B})}(M_2)$ and $D_2^{(\widehat{C})}(M_2)$. In the region $x > 0$ we have the gauge invariant combinations

$$x > 0 : \quad \mathbf{L}^{(1,0)}(\gamma)e^{2\pi i \frac{1}{2}\int_D \widehat{B}_2}, \qquad \mathbf{L}^{(0,1)}(\gamma)e^{2\pi i \frac{1}{2}\int_D \widehat{C}_2}, \tag{B.34}$$

where $\partial D = \gamma$. These combinations in the region $x > 0$ become

$$x > 0 : \quad \mathbf{L}^{(1,0)}(\gamma)e^{2\pi i \frac{1}{2}\int_D \widetilde{B}_2}, \qquad \mathbf{L}^{(0,1)}(\gamma)e^{2\pi i \frac{1}{2}\int_D \widetilde{C}_2}. \tag{B.35}$$

To proceed, we write the theory in the region $x > 0$ in terms of the $SO(4n)$ theory,

$$x > 0 : \quad \sum_{\widetilde{B}_2, \widetilde{C}_2, B_2, C_2} Z_{SO(4n)}[B_2, C_2]e^{2\pi i \frac{1}{2}\int_{W_4}(B_2\widetilde{B}_2 + C_2\widetilde{C}_2 + \widetilde{B}_2\widetilde{C}_2 + \widetilde{B}_2\widehat{C}_2 + \widetilde{C}_2\widehat{B}_2 + \widehat{B}_2\widehat{C}_2)}. \tag{B.36}$$

Varying the exponent with respect to $\widetilde{B}_2$, $\widetilde{C}_2$ yields

$$\widetilde{B}_2 = \widehat{B}_2 + C_2, \qquad \widetilde{C}_2 = \widehat{C}_2 + B_2. \tag{B.37}$$

We can ignore signs because these are $\mathbb{Z}_2$ classes. Using these relations in (B.35) and recalling (B.31), we find

$$x > 0 : \quad \mathbf{L}^{(1,0)}(\gamma)D_2^{(\widehat{C})}(D)e^{2\pi i \frac{1}{2}\int_D \widehat{B}_2}, \qquad \mathbf{L}^{(0,1)}(\gamma)D_2^{(\widehat{B})}(D)e^{2\pi i \frac{1}{2}\int_D \widehat{C}_2}. \tag{B.38}$$

We thus learn that, if the line $\mathbf{L}^{(1,0)}$ passes through the non-invertible defect, it emerges attached to a $D_2^{(\widehat{C})}$ surface, and analogously for the other line.

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
