# Peer review of "Aspects of Categorical Symmetries from Branes: SymTFTs and Generalized Charges"

_SciPost Physics, doi:SciPost Phys. 17, 025 (2024)_

## Round 1 · Referee Report · Anonymous (Referee 1) · 2024-3-11

Strengths

1-The paper provides a systematic correspondence between the concepts in generalized symmetries (SymTFT, generalized charges, condensation defects 't Hooft anomaly) and string theory (Topological actions, D-branes, Hanany-Witten move).
2-The paper contains many concrete examples, including 4d N=4 so(4n) and su(n) SYM, 4d N=2 and N=1 models. The discussions are very explicit.
3-The paper includes detailed tables and summary of topological actions in the appendix, which are useful for researchers who intend to do follow-ups.

Weaknesses

1-The notations in formula are not completely consistent.

Report

This paper provided detailed realizations of the novel concepts in generalized symmetries, such as SymTFT, condensation defects and generalized charges, in the geometric/brane setups in string theory. The significance of this paper is twofold: (1) it provides a new perspective to interpret these generalized symmetry concepts; (2) it inspires researchers to think about string theory in a categorical framework.

I think this paper definitely meets the standard of SciPost Physics, and should be published after a minor revision.

Requested changes

1-In the actions with differential forms throughout this paper, sometimes a wedge product ^ is used and sometimes not. Please make the notations more consistent.
2-The SymTFT actions used throughout the paper, starting from (2.3), differs from the usual conventions by a factor of $2\pi$, please comment on this point.
3-In Table 3, the symmetry $\mathbb{Z}_{2,v}$ has no background gauge field, please write a short comment on this.
4-In the formula before and after (4.38): $M=M_{p_1,q_1}\dots$, some terms are with comma and some are not, please make it more consistent.

---

## Round 1 · Referee Report · Anonymous (Referee 2) · 2024-4-22

Strengths

1- The paper advances the program of connecting the categorical description of symmetries with considerations of holography/geometric engineering.

2- The discussion is clear and many examples are given.

Weaknesses

1- Some points of the formalism are not clearly explained.

Report

This is an interesting paper on symmetries and holography/geometric engineering. The main motivation of the paper is to reinforce the connection between branes and symmetries.

This is a connection that had already been pointed out in a number of papers before (including by some of the authors of the current paper), but the current paper makes some new progress, particularly in the analysis of the connection between the Hanany-Witten effect and some aspect of categorical symmetries.

Overall this is a good paper, and I recommend publication after the minor comments below are addressed.

Requested changes

1- I don't think I understand how the linking pairings are to be computed in practice. As far as I understand, we are instructed to find a non-closed form such that it equals $\ell \Phi$, and then compute the integral of a wedge product involving this non-closed form. I have no issue with the discussion abstractly, but in practice I wouldn't know how to do this calculation. Perhaps the authors could work out an example or two explicitly, to show the interested reader how this works in detail? (Some simple geometry like $S^3/\mathbf{Z}_n$ and/or perhaps eq. (3.115) for $\mathbb{RP}^5$ would be ideal, if possible.) I apologize in advance in the likely case that this is explained in some of the references they cite, but even in this case a short summary and an example or two would greatly benefit the reader.

2- There's a typo above (3.126), where it says "souces".

3- At the beginning of section 3.6, where condensation defects are studied, the authors say "For definiteness we work in type II, but similar remarks apply to M-theory". I was a bit surprised by this, since to my knowledge brane-anti-brane annihilation is poorly understood in M-theory. Could the authors perhaps elaborate a bit on what they meant here?

Recommendation

Ask for minor revision

---

## Round 2 · Referee Report · Anonymous (Referee 3) · 2024-6-7

Report

The authors have made corrections according to the previous report, and it is recommended to be published.

Recommendation

Publish (surpasses expectations and criteria for this Journal; among top 10%)

---

## Round 2 · Referee Report · Anonymous (Referee 4) · 2024-6-10

Report

The authors have addressed my comments. I recommend this paper for publication.

Recommendation

Publish (surpasses expectations and criteria for this Journal; among top 10%)

---

## Round 2 · Referee Report · Anonymous (Referee 5) · 2024-6-20

Strengths

1- The authors give a general and unified recipe of the construction of SymTFT from either geometric engineering or holography.
2- The democratic formulation summarized in section 3 is illuminating and is a very useful tool in the construction of SymTFT.
3- The different sources of BF-couplings and anomalies have been classified and examplified in a very clear way.
4- The interplay between (generalized) symmetries and charges as incarnation of Hanany-Witten effects between branes is accounted for in a very clear way.

Weaknesses

1- In textbooks (Becker&Becker&Schwarz and Polchinski) the fields that satisfy Bianchi identities are denoted by \tilde{F} while F = dC. The authors did not try to make such notational distinction, which I understand that it greatly reduces clustering symbols, but could be a source of confusion.

Report

This work is an important step in understanding the construction of SymTFT from either geometric engineering or holography. The authors proposed a useful toolbox that enables both the construction of a large class of SymTFTs and the analysis of the various interplays between the symmetries and the charges. The mathematical tools (mostly differential forms) used in and the physical intuition (Hanany-Witten effects) behind this paper are surprisingly simple (in a good way), yet the consequences they lead to are profound. The topic, and the quality of paper, are well-suited for SciPost Phys, and I recommend for publication once a few typos are corrected.

Requested changes

1- Typo: RHS of (2.11), the superscript might be (i)?
2- Typo: (2.13), wrong place of \hat

Recommendation

Publish (easily meets expectations and criteria for this Journal; among top 50%)

---

## Round 2 · Author Response

We thank both referees for their insightful comments.

Referree 1. In response to the comment about the wedge product notation, we have fixed these inconsistencies throughout the draft and added comments where the wedge symbols are suppressed in the interest of clarity. In response to the comment about 2π normalizations, we have added a short comment below (2.3). In response to the comment about the vector symmetry background field, we have added a comment in the caption of Table 3. In response to the comment about Mp,q notation we have fixed the typos above and below (4.38).

Referee 2. We thank the reviewer for this comment. We have added a clarifying footnote on page 28. We have fixed this typo. In response to this comment we have added an explicit computation around equation (3.52) and a footnote.

---

## Editorial Decision

published